# Data treatment and corrections for estimating $H_2O$ and $CO_2$ isotope fluxes from high-frequency observations

Robbert P.J. Moonen[1], Getachew A. Adnew[1], Oscar K. Hartogensis[2], Jordi Vilà-Guerau de Arellano[2], David J. Bonell Fontas[1], and Thomas Röckmann[1]

[1]Institute for Marine and Atmospheric Research, Utrecht University, Heidelberglaan 8, 3584 CS, The Netherlands
[2]Meteorology and Air Quality Group, Wageningen University, Droevendaalsesteeg 4, 6708 PB, The Netherlands

**Correspondence:** Robbert Moonen (r.p.j.moonen@uu.nl)

**Abstract.** Current understanding of land-atmosphere exchange fluxes is limited by the fact that available observational techniques mainly quantify net fluxes, which are the sum of generally larger, bi-directional fluxes that partially cancel out. As a consequence, validation of gas exchange fluxes applied in models is challenging due to the lack of ecosystem-scale exchange flux measurements partitioned into soil, plant, and atmospheric components. One promising experimental method to partition measured turbulent fluxes uses the exchange-process-dependent isotopic fractionation of molecules like $CO_2$ and $H_2O$. When applying this method at a field scale, an isotope flux ($\delta$-flux) needs to be measured. Here, we present and discuss observations made during the LIAISE 2021 field campaign using an Eddy Covariance (EC) system coupled to two laser spectrometers for high frequency measurement of the isotopic composition of $H_2O$ and $CO_2$. This campaign took place in the summer of 2021 in the irrigated Ebro River basin near Mollerussa, Spain, embedded in a semi-arid region.

We present a systematic procedure to scrutinise and analyse measurements of the $\delta$-flux variable, which plays a central role in flux partitioning. Our experimental data indicated a larger relative signal loss in the $\delta$-fluxes of $H_2O$ compared to the net ecosystem flux of $H_2O$, while this was not true for $CO_2$. Furthermore, we find that mole fractions and isotope ratios measured with the same instrument can be offset in time by more than a minute for the $H_2O$ isotopologues due to the isotopic memory effect. We discuss how such artifacts can be detected and how they impact flux partitioning. We argue that these effects are likely due to condensation of water on a cellulose filter in our inlet system. Furthermore, we show that these artifacts can be resolved using physically sound corrections for inlet delays and high frequency loss. Only after such corrections and verification's are made, ecosystem scale fluxes can be partitioned using isotopic fluxes as constraints, which in turn allows for conceptual land-atmosphere exchange models to be validated.

## 1 Introduction

Net ecosystem flux measurements of EvapoTranspiration (ET) and Net Ecosystem Exchange (NEE) are used at many sites worldwide to study exchange of water and $CO_2$ between the biosphere and atmosphere. These net fluxes are the sum of partial flux components which are often larger than the net flux and compensate each other. Each of these gross flux components has unique sources and dependencies on environmental variables. The Gross Primary Production (GPP) is dependent on variables

like Photosynthetic Active Radiation (PAR) and the Vapour Pressure Deficit (VPD) (van Diepen et al., 2022). On the other hand, ecosystem Respiration ($R_{eco}$), has strong links to soil and leaf temperatures and water contents. Importantly, such environmental dependencies are used in atmospheric models to predict the evolution of gross and net exchange fluxes in a future changing climate. Thus, to properly validate model parameterization, we need ecosystem scale flux measurements of the gross components.

One promising method that allows for flux partitioning uses the stable isotopic composition of the exchanged molecules. A trace gas has multiple stable isotopologues, or molecules with a given isotopic configuration, which undergo exchange processes at slightly different rates. As the various exchange fluxes are caused by different physio-chemical processes, the isotopic fractionation differs between them. The combined effect of all fractionation processes can be measured on atmospheric molecules and used to partition net exchange fluxes. On the global scale, the atmospheric $CO_2$ budget constrained by measurements of the mean $\delta^{18}O$-$CO_2$ isotopic compositions has been used to separate annual NEE into GPP and $R_{eco}$ (Prentice et al., 2001). On smaller spatiotemporal scales (hourly, local), a different isotopic budget approach can be used to split NEE into GPP and $R_{eco}$, and ET into Evaporation (E) and Transpiration (T) (Lee et al., 2009; Vilà-Guerau de Arellano et al., 2020). Doing so allows us to better understand, and consequently model, the drivers of each flux component, including non-linear short term (diurnal, sub-diurnal) effects (Vilà-Guerau de Arellano et al., 2023). In 1958, Keeling introduced the well-known "Keeling plot" budgeting approach where atmospheric isotopic composition measurements can be used to infer bulk source isotopic compositions. The technique can be used for flux partitioning as well, when combined with analysis and/or modelling of specific source isotopic compositions (Good et al., 2012; Yakir and Wang, 1996). The limitation of Keeling's approach is a lack of insight into the turbulent processes underlying the exchange, and an ill-defined footprint. More recent approaches use micrometeorological measurements to derive the isotopic composition of the exchanging gas flux (iso-flux) and attempt to use it for partitioning. In that case, a mathematical framework relating ecosystem gas exchange fluxes to isotopic compositions needs to be used. Oikawa et al. (2017) present such a framework and clarify that accurate iso-flux measurements are key to reliably partition fluxes.

Two decades ago, isotopic compositions measured with laser spectrometers got precise enough to allow for gradient based iso-flux methods (Griffis et al., 2004, 2007; Welp et al., 2012; Wei et al., 2015). Some years later, it was shown that high sample throughput and precision could be achieved to perform direct flux measurements by combining stable isotope measurements with Eddy Covariance (EC) (Sturm et al., 2012; Griffis, 2013). Since then, measurement of iso-fluxes have been made with instruments that measure isotopic compositions at high temporal resolution (faster than 1Hz). Various research groups have contributed to advancing this combined technique (Wehr and Saleska, 2015; Oikawa et al., 2017; Wahl et al., 2021; Sturm et al., 2012). Still, there is much to be learned about iso-fluxes, the challenges in measuring them and how to correct sub-optimal data (Oikawa et al., 2017). Motivated by the goal of deriving flux partitioning using a method that incorporates turbulence and has a field scale footprint, we focus on the measurements and corrections of turbulent iso-fluxes derived using EC. In this manuscript, we describe the methods and measurement setup we used for making iso-flux measurements of both $H_2O$ and $CO_2$ in the field. We present data from the LIAISE field campaign which took place in July 2021 in the Ebro river basin in the northeast of Spain. In this study we focus on the challenges associated with the setup, and evaluation of the experimental

isotope (flux) data. We present measurement artifacts, the most likely causes, and correction methods we applied. Finally, we share our outlook on flux partitioning and highlight the potential for minute scale iso-flux measurements.

## 2 Theory

$H_2O$ and $CO_2$ molecules in the natural environment consist of all possible combinations of the light and heavy hydrogen, oxygen and carbon isotopes. Since the abundance of the heavy isotopes is very low (D/H = $1.6e^{-4}$, $^{17}O/^{16}O = 3.8e^{-4}$, $^{18}O/^{16}O = 2.0e^{-3}$, $^{13}C/^{12}C = 1.1e^{-2}$), heavy isotopologues are much less abundant than the light isotopologues ($H_2^{16}O$ and $^{12}C^{16}O_2$), and are therefore much more difficult to measure at high precision. An additional difficulty is that the natural variations in isotopic compositions, for example due to fractionation during gas exchange in plants leaves or soil, are small. For that reason, isotope ratios are reported in $\delta$ notation, that is, as a deviation of the heavy-to-light isotope ratio compared to that ratio in a reference sample (Mook and Geyh, 2000).

$$\delta^h X = \frac{{}^h R_{spl}}{{}^h R_{ref}} - 1 = \frac{\left[\frac{{}^h X}{{}^l X}\right]_{spl}}{\left[\frac{{}^h X}{{}^l X}\right]_{ref}} - 1 \tag{1}$$

Here, $X$ represents a molecule, $h$ represents a heavy isotopologue of that molecule, $l$ the abundant (i.e. light) isotopologue, "$_{spl}$" the measured sample, and "$_{ref}$" the reference. For hydrogen and oxygen atoms, the Vienna Standard Mean Ocean Water (VSMOW) is the common reference (D/H = $1.5576e^{-4}$, $^{18}O/^{16}O = 2.00520e^{-3}$), while for carbon this is Vienna Pee Dee Belemnite (VPDB, $^{13}C/^{12}C=1.1180e^{-2}$) (IAEA, 2017; Craig, 1957).

### 2.1 Isotopic fractionation effects associated with land-atmosphere exchange

Different processes that facilitate exchange of $H_2O$ and $CO_2$ between the earth's surface, plants, and the atmosphere, are associated with isotopic fractionation. One example is the phase transition from liquid water to vapour, where light isotopologues evaporate preferentially compared to heavy isotopologues. This is due to the comparatively higher saturation vapour pressure of light $H_2O$ compared to heavy $H_2O$ (Horita and Wesolowski, 1994). Another example is the diffusion of gases through small openings such as stomata, where the light isotopologues enter and leave the stomata at a slightly faster rate than the heavy isotopologues. This is caused by the higher velocity of lighter isotopologues (Mook and Geyh, 2000). Also, photosynthesis itself causes fractionation as $^{12}C$ is preferentially taken up by RuBisCO, resulting in a slight enrichment in the $\delta^{13}C$ of $CO_2$ remaining in the atmosphere (Farquhar et al., 1989; Adnew et al., 2020).

The preferential uptake or emission of light or heavy isotopologues at the land-atmosphere interface, combined with turbulent transport in the boundary layer, leads to vertical gradients of the different isotopologues, and thus the $\delta$-values, in the boundary layer. These idealized gradients are displayed in Figure 1 to conceptually visualize the processes. Note that no gradient-based methods were used. For $H_2O$, plant transpiration is generally enriched in $^{18}O$ compared to the atmospheric reservoir (Yakir et al., 1994). This leads to negative atmospheric gradients ($\frac{\Delta \delta^{18}O}{\Delta z}$) of $\delta^{18}O$ which will result in a positive (upward) $\delta$-flux

(Sect. 2.2). For $CO_2$, the main driver of the gradients is photosynthetic uptake. $CO_2$ near the surface gets enriched in $^{13}C$-$CO_2$, again leading to upward transport of $\delta^{13}C$, which is in this case opposite to the flux of $CO_2$ itself.

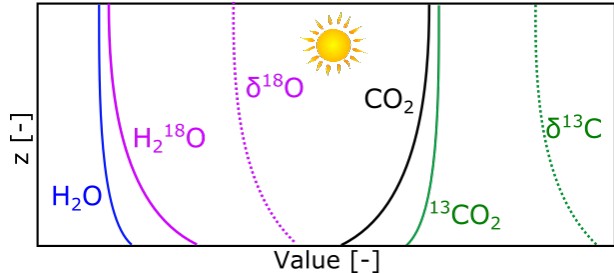

**Figure 1.** Conceptual view of the idealized gradients of mole fractions and $\delta$-values in the atmospheric surface layer (ASL) during midday ($z/L_{ob} < 0$) over a vegetated area.

## 2.2 $\delta$-flux definitions

Turbulent vertical mixing of air in the boundary layer results in a reduction of the concentration and isotope gradients that are illustrated in Fig 1. For the isotopic compositions, expressed as $\delta$-values, this results in a $\delta$-flux.

$$F_\delta = \overline{w' \delta'} \tag{2}$$

Here, $F_\delta$ is the $\delta$-flux in $\text{‰ m s}^{-1}$, $w'$ represents the perturbations in the vertical wind speed in $\text{m s}^{-1}$, and $\delta'$ the perturbations in the $\delta$-values of the molecule in question. Lee et al. (2012) and peers, refer to the $\delta$-flux variable as an iso-forcing since it links the isotopic composition of the flux source to a pertubation in the atmospheric isotopic composition. From an experimental perspective, naming it a $\delta$-flux best expresses that we are referring to a flux of measured $\delta$-values.

Next to the $\delta$-flux, we also derive the isotopic composition of the flux itself ($\delta^h X_F$, Eq. 3), which can be interpreted as the isotopic composition at the exchange interface (Griffis et al., 2004; Lee et al., 2012). Together with the atmospheric isotopic composition, this variable allows us to describe the strength and sign of the isotopic gradient (in $\text{‰ z}^{-1}$) in the atmospheric surface layer. See Fig. 1 for an example of such a gradient. The isotopic composition of the flux can be derived using either the volume flux ratios of the major and minor isotopologues or by combining the $\delta$-flux, $\delta^h X_{atm}$, the net exchange flux, and the atmospheric concentration of the molecule.

$$\delta^h X_F = \frac{^h R_F}{^h R_{ref}} - 1 = \delta^h X_{atm} + F_\delta \frac{C_x}{F_x} \tag{3}$$

$$^h R_F = \frac{\overline{w' q'_{h_x}}}{\overline{w' q'_{l_x}}} \tag{4}$$

Here, $\delta^h X_F$ is the isotopic composition of the flux, $^h R_F$ is a ratio of isotopologue fluxes, $q'_{h_x}$ represents the deviations in the mass fraction of an isotopologue with respect to the 30 minute mean. $\delta^h X_{atm}$ is the atmospheric isotopic composition of the compound, $C_x$ is the total concentration of the molecule in $\mathrm{mol\,m^{-3}}$, and $F_x$ is the flux of the molecule in $\mathrm{mol\,m^{-2}\,s^{-1}}$. When only one process (e.g. photosynthesis) with a given fractionation influences the atmospheric composition, $\delta^h X_F$ - $\delta^h X_{atm}$ should be equal to the magnitude of that fractionation (Fig. 1). In more complex environments, $\delta^h X_F$ - $\delta^h X_{atm}$ is still good indicator of which process is dominant, given that the respective fractionation effects are known. Together, $F_\delta$ and $\delta^h X_F$ are variables that complement the more common $\delta^h X_{atm}$ and net gas exchange measurements by linking them to the physio-chemical processes in the flux footprint.

## 3 LIAISE field campaign

### 3.1 Site description

We performed $\delta$-flux measurements during the LIAISE (Land surface Interactions with the Atmosphere over the Iberian Semi-arid Environment) field campaign in the summer of 2021 (Boone et al., 2021). This campaign took place in the irrigated Ebro basin near Mollerussa, Spain. The focus of the effort was to investigate the effects of large scale irrigation on the atmospheric boundary layer and large scale circulation (Mangan et al., 2023). Iso-flux measurements were made in the middle of a 300 x 400 m field with flood irrigated alfalfa (C3), a fast growing crop with large ET. The alfalfa grew from 50 to 65 cm above ground level during the measurement period described in this manuscript (25-30 July) and covered the entire field during that period. Importantly, the combination of large ET and GPP caused significant isotopic fractionation effects and related iso-fluxes with diurnal cycles to emerge. Flood irrigation took place once during the campaign, two days before our isotope measurement period started. During the measurement period one precipitation event occurred ($26^{th}$). Otherwise, measurement days showed a comparable diurnal weather cycle with largely clear sky conditions and some cirrus, $32^\circ$C mean peak temperatures, and 650 $\mathrm{Wm^{-2}}$ mean peak net radiation. Despite the high temperature and radiation levels, the alfalfa was not water stressed according to leaf-level measurements of stomatal conductance. Wind speeds at 2.45 m were below 2.5 $\mathrm{ms^{-1}}$ for 90% of the time, while the wind direction alternated between easterly and south westerly due to a sea breeze circulation. In Appendix A1 we included a figure which displays the behaviour of some general meteorological variables during the measurement period. The presence of comprehensive measurements of auxiliary variables (like soil moisture contents, stomatal aperture, etc.) will allow us to investigate the iso-fluxes and their main drivers environmental in later work (Mangan et al., 2023; Boone et al., 2021).

### 3.2 Setup

The Iso-flux setup consisted of an Eddy Covariance (EC) station with the addition of two laser spectrometers. The EC station was an IRGASON EC-100 (Campbell Scientific, Logan, USA), which combines a Sonic Anemometer with an Open Path Gas Analyser (OPGA) sampling at 20 Hz (Fig. 2). It was installed on a tripod at 2.45 m above ground level and faced South ($180^\circ$). We used EddyPro version 7.06 (Fratini and Mauder, 2014) from LI-COR Inc (Lincoln, U.S.A.) to output level 6 processed raw

data of the IRGASON dataset which includes corrections for raw data screening including spike removal (Vickers and Mahrt, 1997) and axis rotation with the planar-fit procedure (Wilczak et al., 2001). Processing of the high frequency data to fluxes was done using our own code to be able to include our own spectral corrections. By keeping the flux processing chain equal between OPGA and LS's we ensure that all differences were instrument related. The raw scalar data from the OPGA and LS's were detrended and subsequently filtered for outliers using an Inter-Quartile Range (IQR) filter of 2.5x the inter-quartile range deviation from the mean interval value. For the flux calculations, we applied density and sonic-T corrections. No spectral flux corrections (as described in e.g. Moore, 1986) were applied to prevent interference with our own spectral correction method described in Sect 4.3.2 (note that raw data can't be regenerated from a corrected cospectrum). Moreover, the OPGA and spectral scaling corrections we find for the $\delta$-fluxes (Sect 5.3) are an order magnitude larger compared to the 5-10% loss a spectral flux correction compensates for (Foken, 2008). A 1D flux footprint analysis performed using the Kljun et al. (2004) footprint model implemented in EddyPro indicated that 90% of the flux originated from within 85 m radius around the measurement setup, confirming that fluxes from all wind sectors originated from within the alfalfa field.

20 cm below the anemometer's center an inlet line continuously sampled atmospheric air for analysis in the laser spectrometers. The tubing and instruments downstream were kept free from dust and insects using a Whatmann cellulose thimble inlet filter (Whatmann plc, Maidstone, UK) that was placed at the air inlet. To prevent the bulky instrument enclosures from impacting the turbulence measurements, they were placed away from the EC mast and connected via a 9 m inlet line (3/8" OD, 5/16" ID tubing). This inlet was kept to a reasonably short length to prevent mixing of air samples in the inlet. A total air flow rate of approximately 30 l min$^{-1}$ was generated using the suction of the laser spectrometers (9.9 l min$^{-1}$), and an additional scroll pump with a flow rate of 20+ l min$^{-1}$. This high flow rate assured turbulent flow inlet conditions (Re > 3000). The side panel in Fig. 2 illustrates the layout of the inlet system. Lag times between the EC-system and the isotope analysers caused by the inlet tubing were corrected during post processing (Sect. 4.2). Also, we confirmed no major leaks were present in the inlet tubes by drawing a vacuum on the entire inlet system. To reduce the isotopic exchange between the air and the wall of the inlet tube, the copper inlet line was heated to a 50°C setpoint using a heating wire and tube isolation.

The inlet line separated in between the instruments, downstream of the main inlet, as shown in the flow diagram of Fig. 2. There, 0.9 l min$^{-1}$ of air was directed to a Picarro L2130-i laser spectrometer (Picarro, Santa Clara, USA) measuring H$_2$O isotopologues (H$_2$O, DHO, H$_2^{18}$O) and modified to run at higher sample flow-rates. Another 9 l min$^{-1}$ was directed to an Aerodyne TILDAS-CS laser spectrometer (Aerodyne Research, Billerica, USA) measuring CO$_2$ isotopologues (CO$_2$, $^{13}$CO$_2$, CO$^{18}$O). These instruments have measurement frequencies of 4 Hz and 10 Hz respectively, which allows for eddies of the smaller turbulent scales (cm) to be distinguished, and therefore most of the turbulent energy spectrum to be resolved (Moene and Van Dam, 2014). Both instruments were installed in weatherproof, temperature controlled enclosures which were placed on pallets to protect them from (flood irrigation) water and dirt.

For the TILDAS-CS, we used a field enclosure manufactured by Aerodyne Research Inc., which we purged with N$_2$ to prevent CO$_2$ absorption in the instruments optics (Fig. 2). Additionally, the dry N$_2$ prevents water from condensing on the Peltier thermoelectric cooler used in the setup. The temperature setpoint in the enclosure was 35°C, matching the setpoint

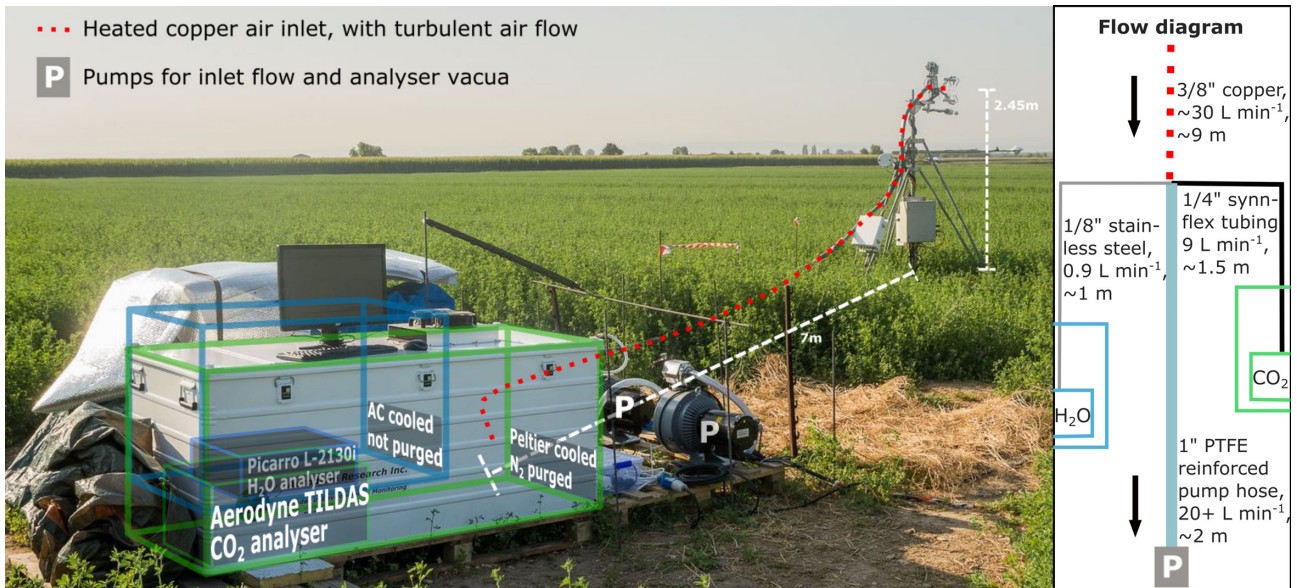

**Figure 2.** Picture of the iso-flux measurement setup (taken by Wouter Mol, wbmol@wur.nl), overlayed with a schematic overview of the measurement setup. The instrument and enclosure outlined in green indicate the $CO_2$ isotopologue setup. The instrument and enclosure outlined in blue indicate the $H_2O$ isotopologue setup, which is behind the $CO_2$ enclosure in the picture. The flow diagram in the side panel indicates how the air was distributed over the analysers and the inlet pump, and lists the outer diameter, material type, flow rate, and length of each tube. Note that most of the length of the tubing feeding air to the analyzers was inside the enclosures.

of the liquid coolant stabilizing the analysers internal temperature. Still, we observed diurnal variations in the power of the analyser's liquid chiller unit (OASIS), which affected the observed isotopic composition of atmospheric $CO_2$ (Sect. 5.2).

The Picarro $H_2O$ isotopologue monitor was placed in a custom built, insulated enclosure which was temperature controlled and dried using a compact AC unit set to 28°C (Fig. 2). Purging the enclosure with dry air was not required as the optical path
in the instrument is short which minimizes out-of-cell light absorption (Picarro, 2021). An AC unit increases the risk of water condensation compared to a Peltier element due to the on-off nature of control. While the analyser's setpoint of 80°C for the sample cell prevents water condensation internally, the inlet tubing is vulnerable to cold pulses. To reduce the external heating from solar radiation and thus the required cooling power, we installed reflective sun-shielding at 15 cm above the lid of the enclosure, allowing for ventilation.
The vacuum pumps providing the required low pressure to the analysers were placed next to the enclosures on another pallet together with the inlet pump, 4G modem, and inlet temperature controller. A roof of wetted hay provided shielding from the sun and evaporative cooling of the air passing over the pumps. During rain events, a tarp was installed instead. The enclosures and peripherals were located perpendicular to the main wind directions with respect to the EC system to prevent footprint disruptions.

 **4   Data treatment**

## 4.1   Calibrations

Laser spectrometers require regular calibrations for accurate measurements of the concentration or isotopic composition of the target species. Griffith (2018) gives an excellent overview of suitable calibration procedures stating that conceptually, two effects need to be addressed by calibration: 1) the dependence of the $\delta$-values on mole fractions and 2) the span calibration,
or, the calibration of the measured isotopic composition against a reference standard. $\delta$-flux measurements do not require high accuracy but mainly need high precision, because detrended fluctuations in $\delta$-values are used to derive the 30 minute $\delta$-flux (Eq. 2). Longer timescale instrument drift and related uncertainty in the absolute isotopic composition are thus removed from the signal (Griffis et al., 2010; Van Kesteren et al., 2013). For this reason, frequent span calibrations were not our priority and we determined the span calibrations and mole fraction dependency of the $\delta$-values in the lab before and after the campaign.
Mole fraction calibrations are key for $\delta$-flux measurements because the variations in atmospheric isotopic compositions are naturally associated with variations in mole fraction, and thus a dependence of the isotopic composition on mole fraction represents a first order interference with the target signal.

The L2130-i was calibrated with a commercial Standards Delivery Module (SDM, Picarro, Santa Clara, USA). Two liquid standards that spanned the range of values measured on site were used. One was demineralized tap water from the Netherlands
(52°N) and the other was melt water from ice-cores from Greenland (75°N). Both were linked to the VSMOW scale using reference standards from the IAEA (Tb. 1). Extensive calibration details are provided in Appendix A2.

**Table 1.** The isotopic compositions of the $H_2O$ calibration standards including the standard error of the 1-min binned data during cross-calibrations of the references, and the average atmospheric isotopic composition during the measurement period including its approximate range

|  | $\delta^{18}O$ | $\delta D$ |
|---|---|---|
| **NL tap water** | -6.98 ± 0.02‰ | -47.12 ± 0.04‰ |
| **GL icecore** | -30.80 ± 0.02‰ | -240.86 ± 0.04‰ |
| **LIAISE atm** | -13.4 ± 1‰ | -94 ± 8‰ |

Differences in the span calibrations performed before and after the campaign were ±0.4‰ for $\delta^{18}O$ and ±0.3‰ for $\delta D$ at atmospheric isotopic compositions. We suspect instrument drift is the cause given the small uncertainty in the (re)calibrations (0.02‰ for both $\delta^{13}C$ and $\delta^{18}O$). When ignoring drift inducing events like instrument rebooting and transportation, the inter-
polated drift during the measurement period is still below 0.1‰ for both species.

The consistency in the mole fraction calibrations is indicated in Fig. 3. Note that the mole fraction calibration curves reveal a cross dependency on the isotopic composition. The origin could not be precisely identified but we suspect that a small leak of ambient air during calibrations could cause this issue. In that case, the ratio of ambient to calibration vapour differed dependent on $H_2O$ concentration, and thus affected the measured isotopic composition.

To reduce the effect of this contamination on the coefficients of the calibration fit, we interpolated the coefficients linearly between the two standards to approximate the isotopic composition of the atmosphere. A 3 : 1 mixture of "NL tap water" and 1 "GL icecore" has a similar isotopic composition as the atmospheric water vapour and was thus used to derive the coefficients applied to the campaign data. We suggest that the similarity in the Weighted Avg (yellow) calibration coefficients "Before" and "After" the campaign in Fig. 3 is no coincidence, but a feature of an ambient air leak of variable magnitude.

In Appendix A4, we give an example where the calibration coefficients in Fig. 3 are fitted to the measurements. We also show simulations where we assume a small (counter) leak (of 0.3%) which is able to explain the observed mole fraction dependencies. An instrument related cross dependency of the isotopic composition on the mole fraction dependence, as described by Weng et al. (2020), is not expected to average out like this. Ultimately, the dependencies were eliminated using $15000\ \mu mol\,mol^{-1}$ as the reference $H_2O$ mole fraction.

The TILDAS-CS was calibrated using a GASMIX AIOLOS 2 (AlyTech, Juvisy-sur-Orge, France). For the span calibrations we used two standards with known isotopic compositions. Next to that, a $8000\ \mu mol\,mol^{-1}$ $CO_2$ canister was diluted with synthetic air ($N_2$, $O_2$, Ar) using a mixing scheme to derive the mole fraction dependence (Tb. 2). Calibration details are provided in Appendix A2. During the campaign we observed slow $5‰$ variations in the $\delta$-values with a diurnal cycle related to instrument housekeeping variables. Consequently, we have little confidence in the measured absolute atmospheric isotopic composition.

However, as discussed before, isotope flux measurements do not require long term accuracy but short term precision. Important to note is that later experiments with similar housekeeping-related drift revealed that mole fraction dependencies remain unaffected (Appendix A4).

**Table 2.** The dependence of isotopic composition measurements ($\delta$-values) on the mole fraction of the respective molecule, $H_2O$ or $CO_2$, expressed in $\mu mol\,mol^{-1}$. The $CO_2$ dependencies were derived after the campaign while the $H_2O$ dependencies represent the weighted average coefficients during the campaign (see Fig. 3).

| Mole fraction dependence | $\delta^{18}O$ - $CO_2$ | $\delta^{13}C$ - $CO_2$ | $\delta D$ - $H_2O$ | $\delta^{18}O$ - $H_2O$ |
|---|---|---|---|---|
| **Linear** | -1.27e$^{-2}$ | -3.64e$^{-2}$ | -7.71e$^{-5}$ | -8.54e$^{-5}$ |
| **Quadratic** | 1.80e$^{-6}$ | 1.16e$^{-5}$ | -1.78e$^{-9}$ | -2.60e$^{-9}$ |

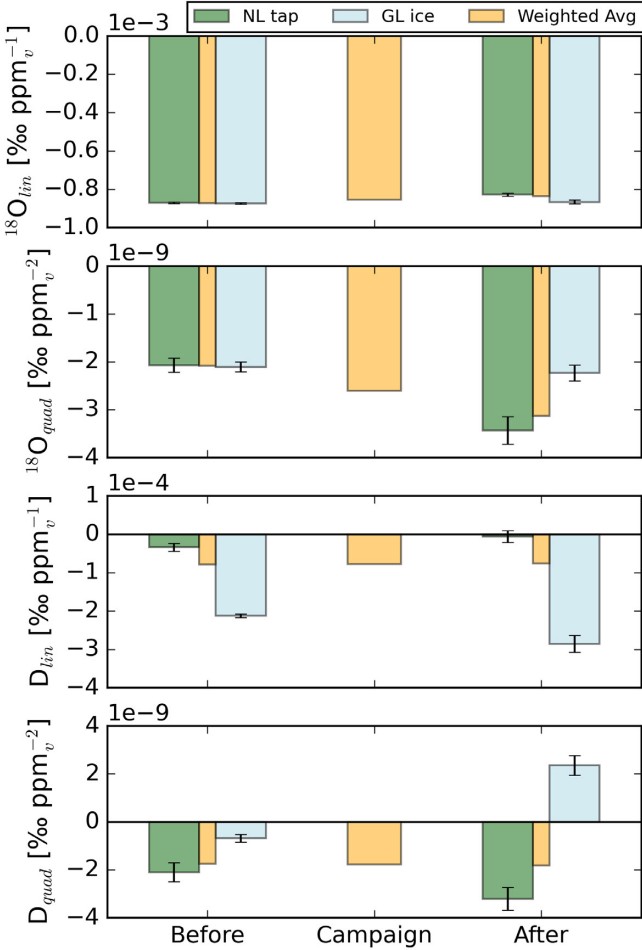

**Figure 3.** Calibration coefficients for solving the mole fraction dependence ($\delta D_{mol} = D_{quad} * H_2O_{ppm_v}^2 + D_{lin} * H_2O_{ppm_v}$), derived before and after the LIAISE campaign for the L2130-i $H_2O$ isotopologue analyser. The yellow bars at the "Before" and "After" moments indicate the virtual calibration coefficients, i.e., what the coefficients would have been when the water standard would have had an isotopic composition similar to the atmospheric isotopic composition. As this minimized inlet contamination effects these coefficients are the ones we worked with. "Campaign" represents the value these virtual coefficients would have had during the measurement campaign in case of linear drift, and are the coefficients we used to correct our measurements with.

## 4.2 Time shift corrections

A practical issue when combining high frequency data from two separate instruments, in our case laser spectrometers and EC, are non-synchronous and drifting data logger clocks in addition to inlet time lag. Since both instrument types measured the mole fraction of $CO_2$ and/or $H_2O$, these mole fraction measurements were used to time align the high frequency time series in post processing using a correlation coefficient (r) based alignment scheme that works as follows (similar to Fan et al., 1990).

After the two time series are coarsely aligned using known clock offsets, the longer time series was divided into short data intervals. We choose to use 10-minute intervals instead of the 30-minute intervals used for flux calculations to increase sensitivity to data series which drift fast and irregularly with respect to each other. One of the data series is subsequently cropped by a minute on each side which makes it possible to shift it in time with respect to the other data series within this (two) minute window. Here, the minimal time shift is the measurement interval of the highest frequency time series. For each of these unique time shifts a correlation coefficient (r) can be calculated between the time series. Note that for deriving r, the high frequency time series should be sub-sampled to the frequency of the low frequency time series. We used masking (or indexing / slicing) for sub-sampling to drop the data points in the high frequency time series that do not align with the low frequency time series. The maximum value of r indicates the time shift of optimal correlation and thus the most probable offset between the measured time series.

We sped up the time alignment by first deriving the r for a subset of time shifts to find the approximate optimum. Subsequently, we filled in the missing r values but only around the approximate optimum. Fig. 4 shows an example of the derived time drift between the $H_2O$ signal of a laser spectrometer and an EC station over a day at two hour intervals. Note that the data logger clocks drifted near linearly in this example. By adjusting the time for each 10-minute section and sub-sampling the data from the higher frequency sensors we constructed one data set of uniform frequency, unaffected by data logger clock drift or inlet line related time lags.

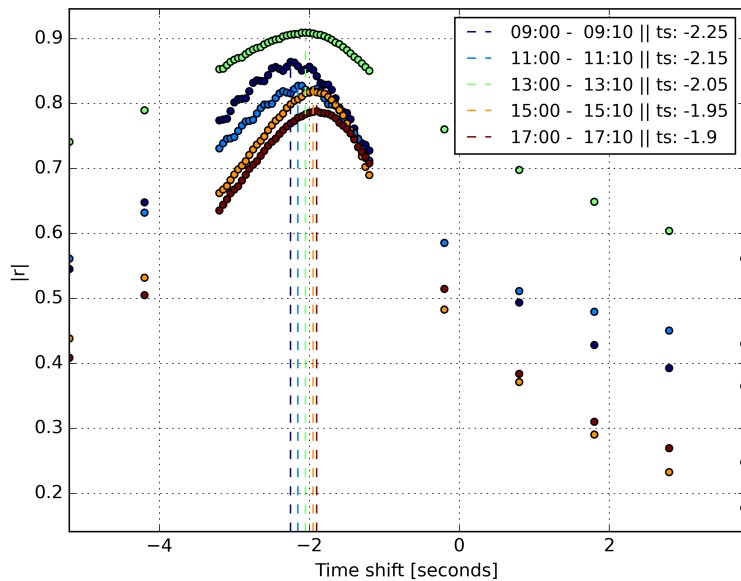

**Figure 4.** Example of the correlation coefficient (r) based time shifts algorithm. In this case the shift between the EC station and the Picarro L2130-i is determined using the $H_2O$ mole fraction measurements of both instruments. The different colors each represent a 10-minute section of data on July $25^{th}$, as indicated in the legend. Each dashed vertical line indicates the optimal time shift of the two 10-minute data sections. The derived time shift values are provided in the legend. A negative time shift indicates that the laser spectrometer is delayed compared to the EC system. Note that during this period the instrument clocks were drifting approximately linearly with respect to each other.

## 4.3 Spectral corrections

Open path gas analysers are known to capture high frequency contributions to gas exchange fluxes better compared to closed path instruments. This is because the smallest spatiotemporal eddy scales are missed by closed path instruments due to inlet line signal attenuation and sample cell retention times (Spank and Bernhofer, 2008). The same is true for the closed path laser spectrometers we use for isotopologue measurements. In this section we detail how we corrected for the lost high frequency signal of both the mass fluxes and delta fluxes using the mole fraction signal from the EC system.

### 4.3.1 Net flux correction


To compare the mass fluxes derived using the EC and the isotopologue instruments, we used cospectra of the fluctuations in $w$ and $CO_2$ or $H_2O$. Such spectra are based on a Fast Fourier Transform (FFT) of a 30 minute data interval and express the contribution to the covariance between two signals as a function of frequencies. In Fig. 5, cospectra of the vertical wind speed (w [$\mathrm{m\,s^{-1}}$]) with specific humidity and $CO_2$ (q [$\mathrm{kg\,kg^{-1}}$]) are shown.

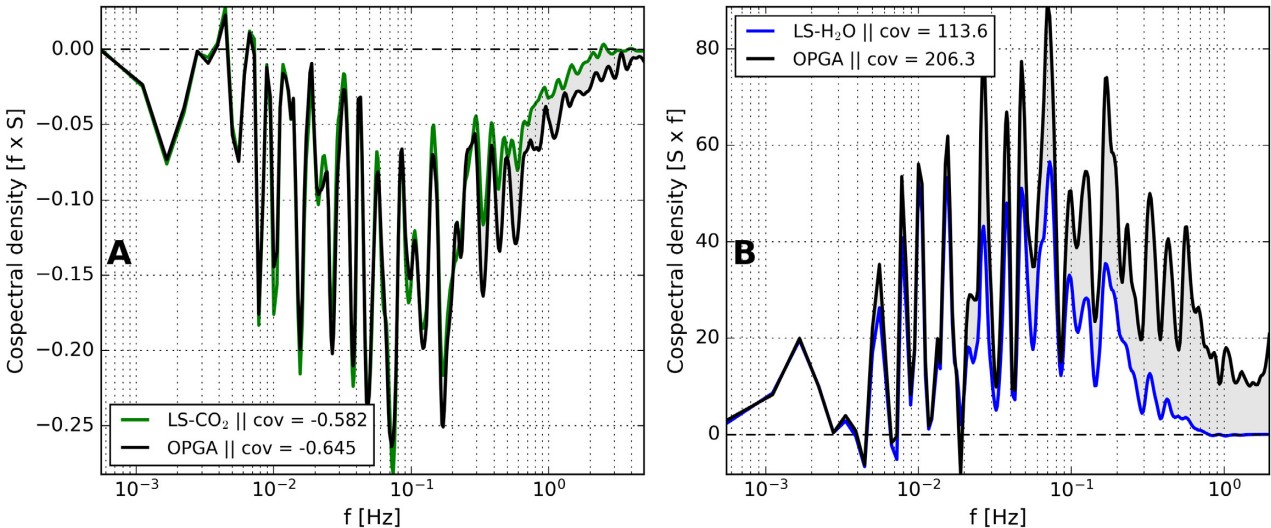

**Figure 5.** Comparison of the cospectra of w′ and q′ from both the closed path laser spectrometers and the OPGA of the Eddy Covariance station. Panel A represents the $CO_2$-w cospectra, and panel B the $H_2O$-w cospectra. The spectra are based on a 30-minute data interval taken during the measurement period starting at 12:00 on July $25^{th}$ 2021.

It is apparent from Fig. 5b that a significant part of the total exchange flux, mostly at higher frequencies, was not captured by the Picarro L2130-i. The TILDAS-CS $CO_2$ isotopologue analyser suffered to a much smaller extent from such high frequency signal loss and captured cospectra similar to the ones derived using the OPGA. When iso-fluxes are measured with OPGA measurements no high frequency signal is lost, so correcting net flux spectra is not required. In case only closed path measurements are used, some correction is needed. One such correction is explained in Spank and Bernhofer (2008) and works

by deriving a transfer function for the closed path instrument to correct for the reduced high frequency signal. However, the observed high frequency loss can also affect single isotopologue flux cospectra and $\delta$-flux cospectra, and therefore needs to be taken into account in some way, optionally through using the signal loss in the net flux measurement.

#### 4.3.2   $\delta$-flux correction

As suggested in the previous section, the fraction of the missed mole fraction signal is one way to estimate how much high

frequency signal is likely lacking from isotopologue and $\delta$-fluxes. This approach has been applied in several other studies (e.g. Wahl et al. (2021); Oikawa et al. (2017); Wehr et al. (2013)), for example through the use of cumulative cospectra (Ogives) of the $w'q'$ between an OPGA and a laser spectrometer. By correcting the high frequency loss of each isotopologue for the loss in net flux, the $\delta$-flux is implicitly increased by the same loss factor (see Eq. 3 & 4).

$$CF_{opga}^x = \frac{cov[w'q'_x]_{opga}}{cov[w'q'_x]_{cpga}} \equiv \frac{\int S_{opga}^{w'q'}(f)\,df}{\int S_{cpga}^{w'q'}(f)\,df}. \tag{5}$$

Here, $CF_{opga}^{x}$ is the correction factor based on the flux measured using a closed path or open path analyser, and $S$ represents the frequency dependent cospectral density (as plotted on the y-axis in e.g. Fig. 5).

Investigation of the cospectra of $\delta$-fluxes and net exchange fluxes revealed that the shapes of both are related at all frequencies for our $CO_2$ isotope measurements (Fig. 6A). Strikingly, this is not true for the $H_2O$ isotope measurements. Fig. 6B indicates that the cospectra of $\delta^{18}O$ and $H_2O$ with respect to $w'$, e.g. the purple and the black lines, are similar only on timescales >100

s, while at faster timescales there are strong differences. The general trend, sign, and small features around $3\times10^{-3}$ Hz and $8\times10^{-3}$ Hz match well. However, at shorter timescales, the $\delta^{18}O$ spectrum progressively diminishes to noise. In contrast, the net $H_2O$ exchange spectrum remains positive with significant contributions to the total exchange flux in this high frequency region.

We propose that the reason for this missing $\delta$-flux signal is a sub-optimal setup rather then a true land-atmosphere exchange

phenomenon (Sect. 6). Given the overlap in spectral shapes at the low frequency side, we designed a spectral scaling approach for finding a $\delta$-flux correction for the signal loss at the high frequency side. The key reasoning behind our approach is that the cospectral shape of a $\delta$-flux should be identical to the shape of the total exchange flux. This implies that eddies of any size which transport isotopically modified $H_2O$ or $CO_2$ have proportionally altered mixing ratios and $\delta$-values. In other words, we assume that each exchange process with its unique fractionation effects contributes to the total $\delta$-flux at all eddy sizes (Sect. 6).

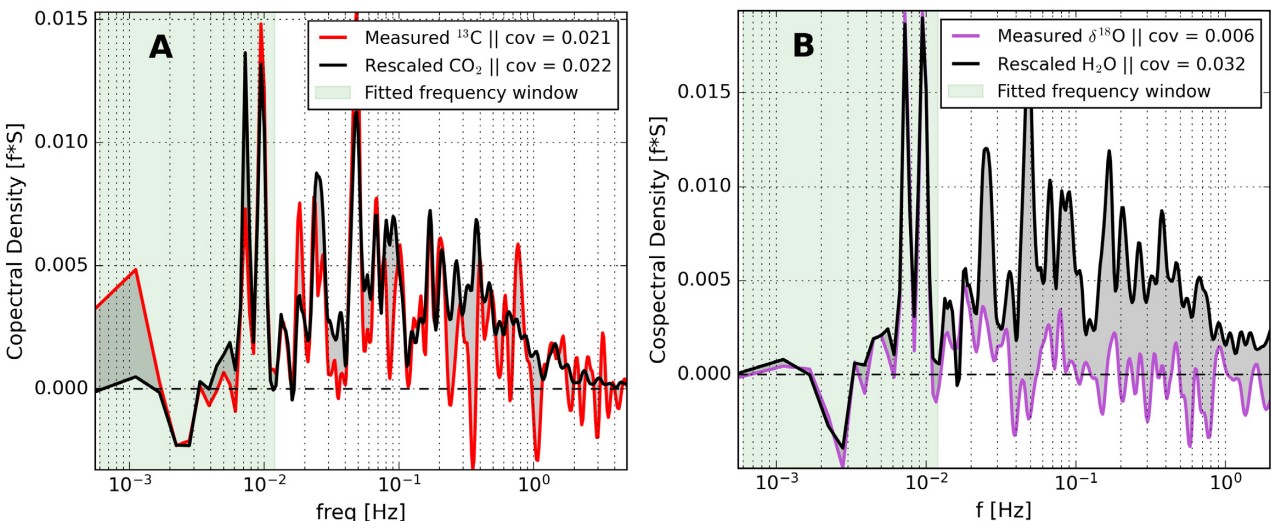

**Figure 6.** Cospectral flux comparison of $\delta^{18}O$ (of $H_2O$) with $H_2O$ (from EC) in panel A, and of $\delta^{13}C$ with $CO_2$ (from EC) in panel B. All cospectra were calculated with respect to w'. The $CO_2$ and $H_2O$ spectra were re-scaled to match the isotopic exchange spectra at timescales longer then 100 seconds. The spectra are based on a 30 minute data interval taken during the measurement period starting at 14:00 on July $25^{th}$ 2021.

To correct for this loss in $\delta^{18}O$ signal we use the $H_2O$ covariance, re-scaled using the measured low frequency contributions from the measured $\delta^{18}O$ flux covariance. We determine the scaling factor (SF) which scales down the $H_2O$ cospectral density

to the $\delta^{18}O$ cospectral density by fitting the spectral powers to Eq. (6) for each 30 minute flux period. The only free variable is choosing a reasonable Low Pass Filter (LPF) that indicates until which frequency the spectra overlap reliably. We chose a LPF of 0.012 Hz based on visual inspection of various co spectra, leaving 19 data points for making a fit. We note that the addition or subtraction of some data points near the filter boundary doesn't change the fit much, suggesting limited sensitivity to the exact value of the LPF.

$$S_{\delta 18O}(f) = SF * S_{H2O}(f) + 0 \qquad where \qquad \begin{cases} f < LPF_{empirical} \\ and \\ sign(S_{\delta 18O}(f)) \equiv sign(S_{H2O}(f)) \end{cases} \tag{6}$$

Here, $S_{\delta^{18}O}(f)$ is the cospectral power of the raw $\delta^{18}O$ $\delta$-flux, $SF$ is the scaling factor, and $LPF_{empirical}$ is the empirically derived Low Pass Filter below which the two co spectra still overlap (see Fig. 6B). Note that the equations above can be applied to any $\delta$-flux but that the condition of the signs being equal is not universal. For example, during photosynthesis the $CO_2$ flux and $\delta^{13}C$ $\delta$-flux are of opposite signs. The condition should thus be adjusted from $\equiv$ to $\neq$. Fitting Eq. 6 is done using a zero-intercept linear regression that relates the spectral densities of $\delta^{18}O$ and $H_2O$, where the slope coefficient equates to the $SF$. Now, $S^*_{\delta 18O}(f)$ and $\overline{w'\delta^{18}O'^*}$, where $*$ denotes the corrected variable, can be defined as follows.

$$\overline{w'\delta^{18}O'^*} = \int S^*_{\delta 18O}(f)\, df = SF \int S_{H2O}(f)\, df = SF\overline{w'q'_x} \tag{7}$$

Additionally, we define the correction factors based on this spectral scaling technique.

$$CF^{\delta 18O}_{spec} = \frac{\overline{w'\delta 18O'^*}}{\overline{w'\delta 18O'}} \tag{8}$$

The benefit of this $\delta$-flux correction is that it is physically sound whether the cause of the missing high frequency signal is lacking sample throughput or isotopic attenuation in the inlet line. Also, the $\delta$-fluxes, which are essential for net ecosystem flux partitioning, do not need need to be calculated indirectly from corrected isotopologue fluxes, but are themselves the target variable. For our data set, we found that the $\delta$-fluxes of $H_2O$ were poorly resolved and that in most cases $CF^{\delta 18O}_{spec}$ & $CF^{\delta D}_{spec}$ » $CF^{H2O}_{opga}$. At the same time, the $\delta$-fluxes of $CO_2$ were well resolved and $CF^{CO2}_{opga} \approx CF^{\delta 13C}_{spec} \approx CF^{\delta 18O}_{spec} \approx 1$ (see Sect. 5.2 and 5.3). The importance of this discrepancy between the $H_2O$ and $CO_2$ signals and between $CF_{spec}$ and $CF^{H2O}_{opga}$ is discussed in Sect. 6.

## 4.4 Quality control and error quantification

In our analysis we minimize systematic errors in the isotope flux dataset caused by outliers in the scalar data by applying a series of data filters. First, we applied a bounds filter to the isotopic data based on their nominal diurnal range during the measurement campaign. Second, instances were operators worked on the instruments and/or opened the enclosures were

flagged and removed, eliminating signal fluctuations due to temperature instabilities. Third, as part of the flux calculations, all detrended variables going into flux calculations, from both the EC-system and the isotope analysers, were filtered for outliers using an Inter-Quartile Range (IQR) filter with a bound of +/- 2.5x the inter-quartile range. Last, flux intervals where less than 75% of the raw data was retained after these steps were excluded from the analysis.

In section. 4.3.2 we investigated the systematic errors caused by lacking high frequency flux contributions, and in the previous paragraph we indicated how we prevented systematic errors caused by signal instabilities and outliers. On top of this, we quantified a random error based on the uncertainty in the slope, and thus the scaling factor, between the net exchange flux and $\delta$-flux (see Sect 4.3.2). The 68% confidence interval of the slope fit was propagated to indicate the error in the spectrally corrected delta fluxes. Data points where the propagated slope errors were above the limits stated in Tab 3 where rejected. Note that drift in the $CO_2$ instrument on long timescales occasionally caused outliers in our spectral correction without a large slope error. The corresponding outliers in the corrected $\delta$-flux were filtered out with a bounds filter (Tab. 3).

The uncertainty in the source compositions was determined by propagating the errors in the $\delta$-fluxes, and the errors in the net exchange flux. For $CO_2$, where we have more confidence in the uncorrected $\delta$-fluxes compared to the rescaled $\delta$-fluxes, we derived the error for the raw source compositions. In that case, we assume the relative error in the $CO_2$ flux to be a good approximation for the relative error in the uncorrected delta flux. For both the net $CO_2$ and $H_2O$ fluxes, relative errors were based on random error estimates from EddyPro after Finkelstein and Sims (2001). A major cause for the scatter in the isotopic source compositions is related to its definition given in Equation 3. When the relative errors in the fluxes become large, especially during sign transitions where the net flux values cross zero, the uncertainty in the source compositions increases too. Like for the delta fluxes, data points where the propagated errors were above the limits stated in Tab. 3 were filtered out. After these filtering steps, 81% of the 30-min flux intervals of $CO_2$, and 80% of the 30-min intervals of $H_2O$ were still complete.

**Table 3.** Error limits and bounds applied to the $\delta$-fluxes and isotopic source compositions. Sect. 4.4 describes how the errors were calculated. The source compositions for $CO_2$ are calculated relative to the atmospheric isotopic composition, similar to Figure 8 ($\Delta\delta^h X = \delta^h X_F - \delta^h X_{atm}$)

| | $\delta$-flux/source species | Error limit (one-sided) | Lower bound | Upper bound |
|---|---|---|---|---|
| **H$_2$O** | $F_\delta$-$\delta D$ | 0.07 ‰ m s-1 | N/A | N/A |
| | $F_\delta$-$\delta^{18}O$ | 0.07 / 8 ‰ m s-1 | N/A | N/A |
| | $\delta D_F$ | 9 ‰ | -120 ‰ | -40 ‰ |
| | $\delta^{18}O_F$ | 9 / 8 ‰ | -17 ‰ | -3 ‰ |
| **CO$_2$** | $F_\delta$-$\delta^{13}C$ | 0.03 ‰ m s-1 | -0.02 ‰ m s-1 | 0.05 ‰ m s-1 |
| | $F_\delta$-$\delta^{18}O$ | 0.01 ‰ m s-1 | -0.02 ‰ m s-1 | 0.02 ‰ m s-1 |
| | $\Delta\delta^{13}C$ | 7 ‰ | -30 ‰ | 5 ‰ |
| | $\Delta\delta^{18}O$ | 4 ‰ | -20 ‰ | 10 ‰ |

# 5 Results

## 5.1 Time lag in isotopic signal

We used the same time alignment strategy described in Sect. 4.2 for synchronizing the laser spectrometers with the Eddy Co-variance system to investigate the lag time of isotopic signals with respect to the mole fractions of the molecules. In theory, eddies with isotopically modified $H_2O$ or $CO_2$ are expected to have deviations in mole fractions and $\delta$-values that are proportional. When measuring such air parcels with the same instrument, at the same time, the $\delta$-values and mole fractions should co-vary perfectly. Still, we found a time lag between $H_2O$ and its isotopic signals of tens of seconds that had a strong diurnal

cycle. Fig. 7 shows the pattern of the time lag between $\delta D$ and $H_2O$ over the measurement period. $\delta^{18}O$ displays near identical behaviour with approximately 20% smaller lag times. During night time, time lags were largest and often not resolved as no valid time lag could be identified. Note that the $H_2O$ signal displayed was measured by the laser spectrometer and not by the OPGA.

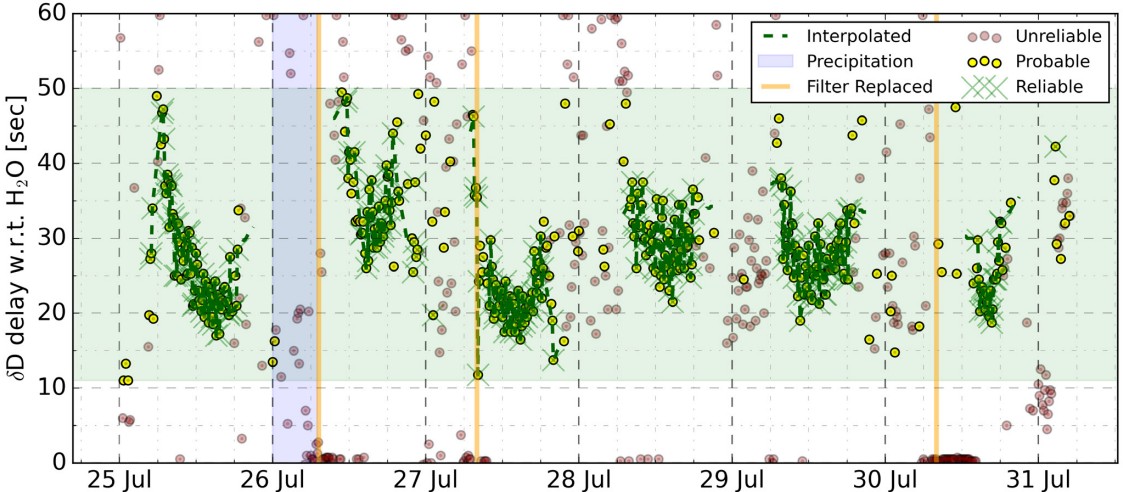

**Figure 7.** Time shifts between the $\delta D$ and $H_2O$ signals of the L2130-i analyzer. All symbols represent a 10 minute interval for which a time shift was derived. The red symbols indicate those intervals where the time shift was based on a low correlation coefficient ($r < 0.25$) and/or unlikely magnitude. The yellow symbols indicate that the time shift has a magnitude fitting within the general pattern of time lags (empirically set light green window) and an $r > 0.25$ (See Fig. 4). The green symbols indicate the data points which are part of a sequence of reliable "yellow" points. The dashed green line fitted through the green symbols was used to align the $\delta$ and $H_2O$ signals.

In Fig. 7, the green dots are those time shifts which fall into the empirically derived range of time shifts, have sufficiently

high correlation coefficients ($r > 0.25$), and are part of a sequence of successive points which all comply with the first two rules. Consequently, the green dashed line connecting the green dots indicates the likely pattern in the time lag between the isotopic composition and the $H_2O$ mole fraction. There are significant differences when inter-comparing days, but in the late afternoon - when temperatures are highest - the time lag is generally smallest. Red dots indicate that the time shifts had low

$R^2$ values or that the time shifts deviated too strongly from its neighbours. We see clusters of red dots on the mornings of the $26^{th}$, $27^{th}$ and $30^{th}$, likely due to replacement of the inlet filter. In the discussion (Sect. 6), we will clarify why and how the time lag is related to this inlet filter.

The $CO_2$ signal and its $\delta$-values do not suffer from a time offset in their signals. For $\delta^{13}C$, the lag time is 0 for the entire data set, with few outliers. $\delta^{18}O$ has time lags of zero, 90% of the time. It has more outliers and occasional midday time lags of only 1-2 seconds, which we chose not to correct for (Appendix A5).

Evidently, the lag times we find are problematic for $\delta$-flux calculations as they create an offset between w' and $\delta$' ( Eq. 2). To prevent this, we shifted the $\delta$ signals in time to match $H_2O$ like explained in Sect. 4.2. On top of this, we discarded nighttime data as no reliable time shift could be found due to limited natural signal variability during stable atmospheric conditions and large lag times associated with high relative humidity levels (Appendix A1). Note that the spectral corrections explained in Sect. 4.3.2 were applied after these time adjustments were made.

## 5.2   $CO_2$ Fluxes and isotopic signals

Using the time shift and spectral corrections, we generated a final output data set containing the various net ecosystem and $\delta$-fluxes. Fig. 8 shows the $CO_2$ exchange fluxes between the fast growing alfalfa crop and the atmosphere.

Clear diurnal trends are visible in the $CO_2$ uptake and the $\delta^{13}C$ exchange flux. The sign of the net ecosystem exchange of $CO_2$ is as expected, demonstrating the dominance of photosynthesis during daytime and respiration at night. The total $CO_2$ flux signal derived from the closed cell $CO_2$ laser spectrometer matches the OPGA well. A linear regression between both results in a slope of 0.91, and a $R^2$ of 0.98. Similarly, the raw $\delta^{13}C$ and $\delta^{18}O$ $\delta$-fluxes are comparable to the spectrally corrected variants. This gives confidence in our ability to resolve the $\delta$-fluxes for $CO_2$ without needing to rely on corrections (Sect. 4.3.2). Note that the instability of the $CO_2$ analyser on longer timescales, as mentioned in Sect. 4.1, causes increased uncertainty in the spectrally scaled $\delta$-flux signals compared to the raw $\delta$-flux signals. Therefore, we show the spectrally corrected $\delta$-fluxes in Fig. 8 in grey for completeness but proceed in the further analysis with the raw $\delta$-flux signals. The signs of the $\delta$-flux indicates that during daytime photosynthetic uptake there is an upward transport of air parcels with higher $\delta^{13}C$. This is in line with the fact that plants preferentially take up $^{12}C$-$CO_2$, and follows the conceptual logic presented in Fig. 1. Panel D and E show the difference between the isotopic signature of the vertical flux and the ambient reservoir ($\Delta\delta^h X = \delta^h X_F - \delta^h X_{atm}$). For $\delta^{13}C$, we observe that the surface reservoir is $\approx$ -15‰ more depleted in $^{13}C$-$CO_2$ than the atmosphere. Given an atmospheric value of $\approx$-8‰, the vertical flux (or source) signature is -23‰, which is a typical value for the $\delta^{13}C$ composition of C3 plants (Kohn, 2010). When individual fluxes are close to zero, which happens during morning and afternoon sign transitions, unrealistic values occur in the derived source composition calculated according to Eq. 3. Most of these data points were filtered out due to the error limits set in Sect. 4.4. Still, some unlikely values persist, especially around zero for $\Delta\delta^{13}C$, where error propagation results in small absolute errors. Also note that the errors in the $\delta$-fluxes are somewhat smaller from the $28^{th}$ onwards due to a software change that reduced the temperature and consequent isotopic variability in the LS-$CO_2$.

$^{18}O$-$CO_2$ $\delta$-fluxes vary around zero, indicating small and variable atmospheric gradients caused by small differences between atmospheric and source compositions. Instrument instability also affects $F_\delta$-$\delta^{18}O$ and given the small 'real' flux signal,

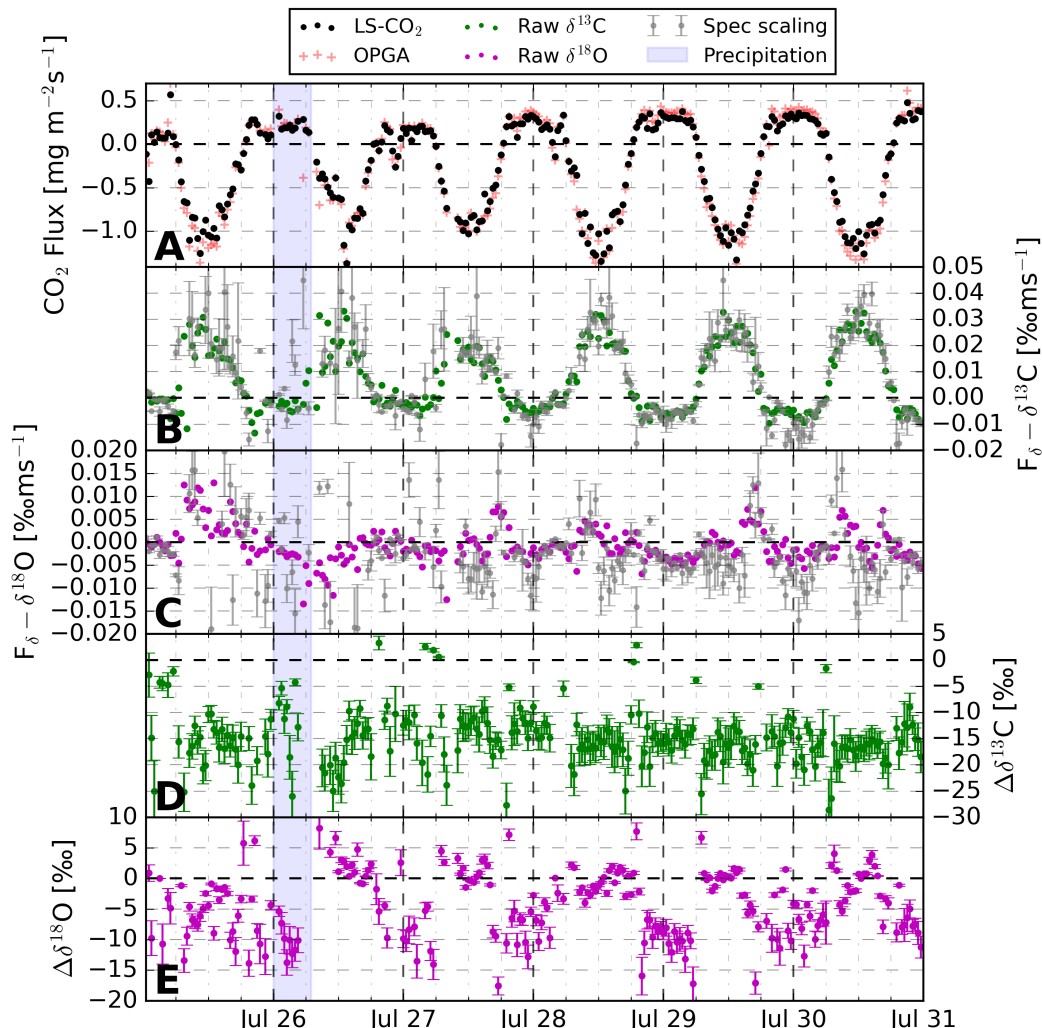

**Figure 8.** 30 minute $CO_2$ exchange fluxes over the 6 day measurement period above an alfalfa crop field near Mollerussa, Spain. Panel B and C display the $\delta$-flux derived using Eq. (2), including the uncertainty estimate propagated from a 68% confidence interval in fitting the spectral scaling method (Sect. 4.4). Panel D and E show the difference in isotopic composition between the vertical exchange flux and atmosphere ($\delta^h X_F$ - $\delta^h X_{atm}$), including the propagated $1\sigma$ uncertainty. The difference is plotted instead of both variables separately to eliminate the effect of instrument drift (see Sect. 4.1 on calibrations).

the instability likely dominates the noisy spectrally scaled $F_\delta$-$\delta^{18}O$ signal. However, CF's are regularly larger then 1, which is likely an artifact of the low frequency instrument instability, but could also be an indication of some missed high frequency

flux signal. In line with a small $\delta$-flux, our daytime measurements show no notable difference between the isotopic composition of the vertical flux and the atmosphere ($\Delta\delta^{18}O \approx 0$, Fig. 8). This is not in line with the normative enrichment in the $\delta^{18}O$-$CO_2$ signature near the earths surface over vegetated areas described in literature (Clog et al., 2015). Key processes

impacting the $\delta^{18}O$ of $CO_2$ are generally the oxygen isotopic exchange in soil and leaf water, and diffusive fractionation of $^{18}O\text{-}CO_2$ during plant assimilation (Rothfuss et al., 2013; Adnew et al., 2023). Given the large stomatal conductance of the alfalfa crop in the footprint, diffusive fractionation effects will have been small (Adnew et al., 2020).

We assume that the water fed to the vegetation - and thus the soil - being largely melt water from the Pyrenees is the major cause of the small $\Delta\delta^{18}O$. This water will be more depleted compared to rainwater because of the increased altitude at which droplets formed (Gat et al., 2001, p. 53-55). When this isotopically depleted water is used by the plants and exchanges isotopes with $CO_2$ in the mesophyll, this will lead to relatively $^{18}O\text{-}CO_2$ depleted $\delta$-fluxes during daytime (Yakir et al., 1994). Moreover, frequent irrigation events prevent strong enrichment of the liquid water in the top soil. Therefore, soil contributions to the respiration, and thus the $\delta^{18}O_F$, which get equilibrated with soil water under influence of Carbonic Anhydrase, will be more similar to $\delta^{18}O_{atm}$. The invasion of atmospheric $CO_2$ into and out of the soil, where oxygen isotopic equibrilation takes place, will contribute in a similar way (Wingate et al., 2009). Finally, oxygen exchange of $CO_2$ with $H_2O$ in the plants mesophyll, where the water isotopic composition is linked to root-zone water from the Pyrenees, equally supports relatively depleted $\delta^{18}O_F$ (Yakir et al., 1994).

Another interesting feature visible in Fig. 8 is the effect of the precipitation event on $\delta^{18}O$. The $\delta$-fluxes become amplified and the source-atmosphere difference increases, most notably just after the precipitation event. It is not clear whether the $\delta^{18}O$ equibrilation of $CO_2$ with $H_2O$, responsible for changing $F_\delta\text{-}\delta^{18}O$ ($CO_2$), predominantly takes place in the atmosphere, the soil, or in the mesophyll of plants.

## 5.3 H₂O fluxes and isotopic signals

In the previous section we indicated that the $CO_2$ isotope fluxes we measured were well resolved and therefore did not require corrections. This was not true for our $H_2O$ isotope fluxes as large time shifts were required to realign the data set, and corrections for the signal loss at high frequency needed to be applied.

Fig. 9 shows an overview of the net ecosystem and isotopic exchange of water vapour over the 6 day measurement period. Clear diurnal patterns are visible in all variables, including the isotopic compositions of the flux displayed in panels D and E. The precipitation event on the $26^{th}$ reduces the magnitude of the Latent Heat Flux ($L_vE$) and deforms its diurnal pattern, likely due to cloud shading. The $L_vE$ signal also reveals large differences between the gas exchange measured by the closed path instrument and the OPGA. A linear regression between both variables results in a slope of 0.59 and an $R^2$ of 0.94. In Sect. 4.3.1 we have shown that this is largely caused by missed high frequency variations in the $H_2O$ signal. This ratio between the OPGA and the closed path instrument is not constant and is largest on the $25^{th}$ and the $30^{th}$, and is equal to the magnitude of the OPGA scaling applied in panels B and C to the $F_\delta$ of $\delta D$ and $\delta^{18}O$. While the OPGA correction is significant, panel B and C reveal that the magnitude of the spectral scaling correction on the $\delta$-fluxes is even larger (Sect. 6) which is in line with the loss of high frequency contributions to the $\delta$-flux in Fig. 6. The CF's for $\delta D$ were about 10% larger then the CF's of $\delta^{18}O$, which matches with the larger lag times for $\delta D$ shown in Sect. 5.1.

In terms of processes, transpiration dominated, and the $\delta$-values in the stomata increased strongly during daytime due to the persistent preferential evaporation of light isotopologues from the small water volume (Sect. 2.1). As a consequence, transpired

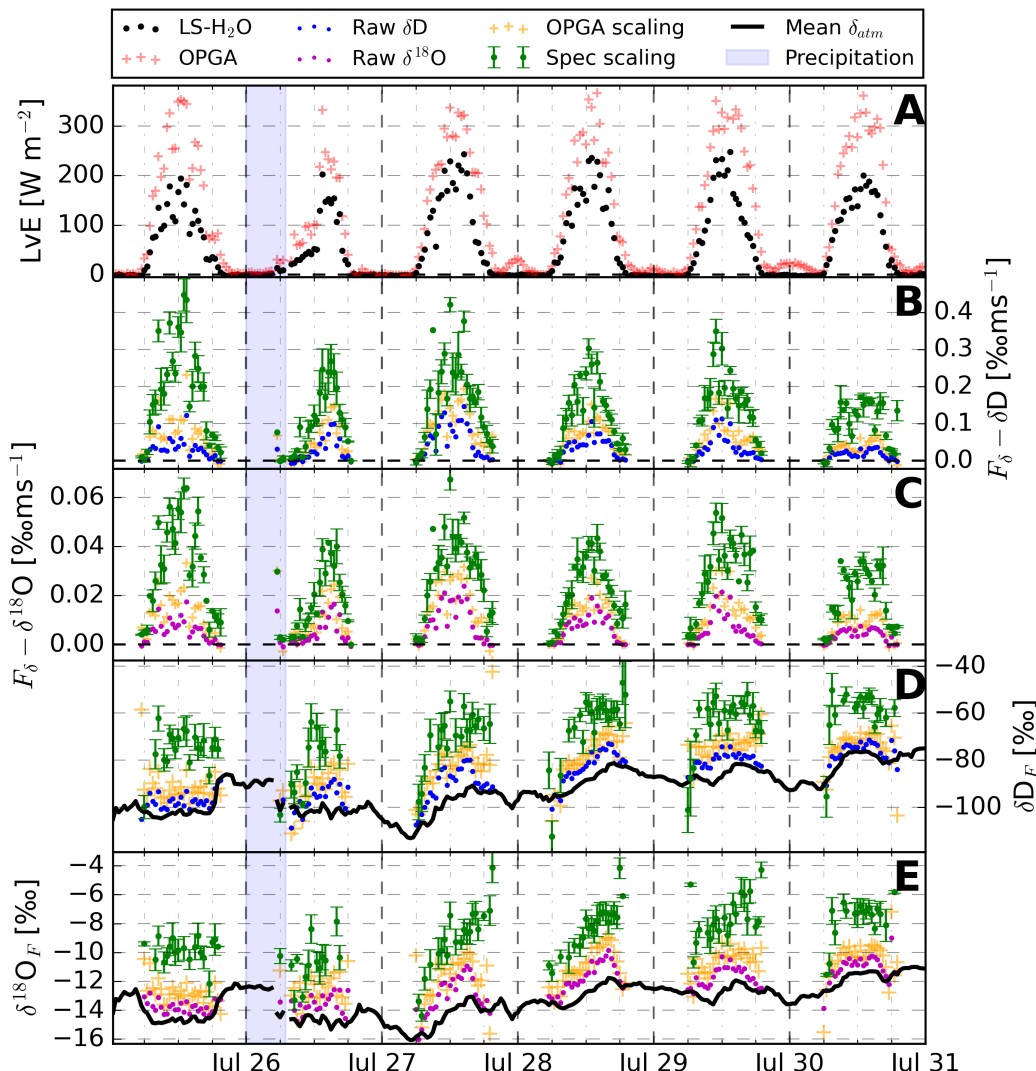

**Figure 9.** 30-minute $H_2O$ exchange fluxes over the 6-day measurement period in Spain near Mollerussa. Panel A shows the evolution of $L_vE$ measured by the LS-$H_2O$ and OPGA. Panel B and C display the $\delta$-flux derived using Eq. (2), including the uncertainty estimate propagated from a 68% confidence interval in fitting the spectral scaling method (Sect. 4.4). Panel D and E indicate the isotopic composition of the vertical flux ($\delta^h X_F$) and the atmospheric background ($\delta^h X_{atm}$, black line), including the propagated $1\sigma$ errors in the fluxes. Panels B through E all contain three different symbols. The full circles represent the raw $\delta$-fluxes or $\delta^h X_F$, while the yellow and green markings indicate the corrected ones. These corrections are based on OPGA scaling like is common in isotope flux research (yellow), or on the Spectral scaling technique we detail in Sect. 4.3.2 (green).

water vapour is enriched in D and $^{18}O$ compared to the atmospheric background, which leads to isotope enrichment of air near the surface and ultimately to a negative vertical gradient (Fig. 1). In turn, the $\delta$-fluxes are positive, transporting enriched water

vapour up into the mixed layer. The precipitation water - which is more enriched compared to the irrigation water – does not have a notable impact on the $\delta$-flux or source composition on short timescales. This suggests that the alfalfa, which has deep roots, takes up water from deeper soil layers where the water sources are mixed or buffered. In line with the positive $\delta$-fluxes, the isotopic composition of the vertical flux ($\delta^h X_F$) starts at near atmospheric values in the early morning, and becomes enriched in the course of the day when transpiration persists and intensifies. This process and its effect on the isotopic signals is similar for $\delta D$ and $\delta^{18}O$.

## 6 Discussion

We have shown in our analysis that isotope flux measurements can add process information to state of the art $H_2O$ or $CO_2$ net ecosystem exchanges flux measurements and atmospheric isotopic composition measurements, that can be used in future studies to partition fluxes. Still, obstacles like unreliable measurements and expensive instrumentation limit further implementation of the technique. Also, the scarce and possibly biased $\delta$-flux data prevent further development of the partitioning method. To allow for more widespread implementation, corrections that ensure reliable $\delta$-flux data are key. As mentioned above, $\delta$-fluxes themselves are generally corrected using the signal loss in the net flux measured by the closed path laser spectrometer compared to the signal loss of the same net flux measured using an open path gas analyser (Wahl et al., 2021; Oikawa et al., 2017; Wehr et al., 2013). We presented a different correction method based on spectral scaling, which suggests qualitatively different corrections. What are its consequences?

First, recall that in our $H_2O$ laser spectrometer, mole fraction signals arrived before isotopologue signals with a time offset of tens of seconds that varied over the day. We know, for example from simple lab tests like breathing highly humid air into the inlet tube, that such time-offsets can be caused by condensation in the inlet. This is most probably caused by heavy isotopologues being more strongly bound to the liquid phase, and thus having a longer residence time in condensation droplets compared to light isotopologues. We expected that our preventive measures of using an actively heated inlet and high enclosure temperatures would have eliminated such condensation effects. However, in our setup during LIAISE the cellulose inlet filter was not heated. Likely, the hydrophilic nature of the material allowed for a small liquid water reservoir to form, which caused the isotopic attenuation and consequent lag times that we observed in our data (Reishofer et al., 2022). The diurnal cycle in lag times would suggest a variable reservoir size, positively related to the atmospheric relative humidity which is generally lower during daytime compared to nighttime.

Despite these complications, we are confident in the reliability and accuracy of our measurements for the following reasons. First, we observe well defined cospectra for $\overline{w'q'}$ using the mole fractions measured by the $H_2O$ laser spectrometer, which are to a great degree similar to those made with OPGA data. Secondly, after correcting for the lag times, we find $\delta$-fluxes with cospectral signal, mainly at the lower frequencies. Finally, the $\delta$-fluxes we resolve are of the sign we expect and follow logical diurnal cycles. Likely, the dampened isotopic signals have the same cause as the mole fraction - $\delta$-value lag times: Isotopic exchange in the inlet. Missing cospectral signal in $\delta$-fluxes and mole fraction fluxes is the rule rather than the exception in iso-flux measurements (Oikawa et al., 2017; Wahl et al., 2021; Wehr et al., 2013). The cause of missing high frequency signal is

lacking time resolution in the measurements. This can even occur when using the appropriate high frequency sensors when the re-flushing of the sample cell with new sample air takes longer then the analysis of a sample. However, we show that besides lacking sample separation due to limited flow rates or sampling frequency, isotopic attenuation in the inlet line can also be a cause. We suggest using the lag time between mole fractions and $\delta$-values as a diagnostic tool to identify inlet attenuation, at least for $H_2O$. If such exchange is found to be present, spectra will certainly be affected and should be corrected appropriately.

We presented two options for correcting the spectra of isotopologue fluxes that differ significantly in their outcome. The OPGA based scaling method has been applied in previous studies and uses the lack in magnitude of the net exchange flux measured by an isotopologue analyser compared to an OPGA, to derive a correction factor with which all individual isotopologue fluxes are corrected. According to Eq. (3 & 4) this implies that the loss in $\delta$-flux is as large as the loss in net exchange flux. We show that in cases where there is inlet exchange and attenuation, nonuniform time offsets between the isotopologues causes a greater loss of covariance with w' for $\delta$-values compared to the mole fractions of a molecule. These nonuniform time offsets will naturally apply to eddies of shorter timescales more strongly. Therefore, our spectral scaling correction method only uses the signal from the very biggest and "longest" eddies. Conceptually, the covariance between two signals that are time offset by 10 seconds is not affected much at timescales > 100 seconds. One implication is that even if the goal is to measure mass fluxes of individual isotopologues, it is better to correct the $\delta$-flux spectrum and use Eq. (3 & 4) to retrieve the corrected isotopologue concentrations.

Zooming in on the spectral scaling principle there is one fundamental assumption that requires further discussion. Namely, the assumption that eddies with altered concentrations of a mole fraction as a consequence of land atmosphere exchange have proportionally altered $\delta$-values dependent on the process of exchange. In other words, the exchange cospectra of $\overline{w'CO_2'}$ and e.g. $\overline{w'\delta^{13}C'}$ should have identical shapes, irrespective of the sign. For the footprint of an EC station above a low crop, small eddies arising from specific leaves or soil sections will have undergone exchange and will now contain modified concentrations of $CO_2$ and $H_2O$. The added or removed molecules leave their specific isotopic fingerprints. Turbulence will organize eddies and mix this source signal into the Kolmogorov cascade of eddy scales to be detected by the EC station (Kolmogorov, 1941). All eddies, big or small, will then pick up part of the source signal. Essentially, the mole fraction and isotope effects of the surface source or sink will stay coupled and will thus be observable in a similar way throughout the scales of eddies.

To prove this hypothesis we can investigate the cospectra of $\delta$-fluxes and mole fraction fluxes measured with a setup in which all turbulent scales are well represented. In Fig. 6 we show that the cospectral density for the $\delta^{13}C$ and $CO_2$ observations is generally very similar, supporting the validity of our hypothesis. While comparing the corrected and non-corrected d-fluxes of both $\delta^{13}C$ and $\delta^{18}O$ in Fig.8 we see the same pattern. The corrected d-fluxes are mostly similar to the non-corrected d-fluxes, even though all high frequency fluctuations were eliminated. Note that the noise and large uncertainties that are present in the corrected delta fluxes can be partially attributed to the instrument drift in the LS-$CO_2$ on long timescales and a relatively low signal to noise ratio in the $\delta^{18}O$-fluxes. More precise experimental measurements of the net ecosystem exchange and net isotopic exchange of trace gasses affected by various fractionation processes should increase confidence in this hypothesis.

The spectral scaling technique that arises generates correction factors for the $\delta$ fluxes which deviate, in our case strongly, from the correction factors for the net exchange flux. This is of major importance as it will lead to a different flux partitioning.

In this study, we do not expand on the consequences for partitioning much as we do not have strong constraints on all isotopic signatures that are needed for partitioning. However, isotopic ecosystem flux partitioning is impacted by our findings.

Be aware that in contrast to most correction approaches, where the order of magnitude of the correction is 10%, the spectral scaling approach leads to correction factors of the order of 100%. This means that most of the signal in the corrected $\delta$-flux does not originate from the actual measurements of the $\delta$-flux, but from the correction method, which is not desirable. Consequently, the errors of the corrected delta fluxes (see Sec. 4.4) can best be based on the uncertainty in fit of the correction, and not on the uncertainty of the measured $\delta$-flux. As long as errors are properly quantified by propagating this fit error to the flux, we do believe that using the spectral correction approach is valuable for deriving $\delta$-fluxes of the correct magnitude.

The implications of the spectral scaling principle are broader than using it to find adequate corrections. If the hypotheses are correct, high frequency isotope measurements may not be required for 30 min average iso-flux determinations as we can infer the high frequency iso-flux contributions from the OPGA scaling. If we do not need to resolve the smallest eddies, setups can be simplified significantly by reducing inlet flow, increasing inlet line length, and setting up the analyser further away from the EC station. Note that in stable atmospheric conditions, the largest eddy scales do not contribute to the flux. However, ample flux signal is needed in the limited fitting window to make a reliable correction, and this poses a limit on how slow the low frequency isotopic composition measurements can be, or at what frequency a LPF can be placed (Sec. 4.3.2). Even when low frequency flux contributions are present, there will be more uncertainty in the determined $\delta$-flux due to the error in fitting the spectra (Sec. 4.4). This can potentially be compensated by increased measurement precision through instrument development instead of increased measurement frequency. It opens the door to use cheaper, low flow rate isotope analysers to measure $\delta$-fluxes at more ecosystem flux measurement sites.

## 6.1 Outlook

The isotope-dependent, loss of high frequency information in the co-spectra reported here is an essential feature that needs to be addressed in future, isotope-based flux partitioning. Ideally, a field experiment designed to determine the full isotopic $\delta$-flux cospectra needs to be performed that applies the best practices we outline in Sect. 7. A high frequency data set that does not suffer from the artifacts we encountered could then be used to corroborate our proposed spectral correction method by downsampling to a lower frequency data set. In addition, a lower measurement frequency limit can be established which ensures that adequate low frequency $\delta$-flux contributions are available to make a fit, under a variety of meteorological conditions. We also recommend the exploration of our spectral correction method for isotope analysers that are not necessarily designed for fast turbulence measurements, for instance the laser spectrometers with high flow rates used to measure vertical $\delta$-gradients (Griffis et al., 2007; Wei et al., 2015). By adding sonic anemometers close to the inlets, spectral corrected $\delta$-fluxes can be derived using the EC method and compared with the $\delta$-fluxes derived from the vertical profiles. Good agreement would give confidence in the spectral scaling method. Further investigation into the lag times between delta-values and mole fractions would be necessary if the isotopic memory effects presented here cannot be excluded by avoiding the use of hydroscopic materials.

Besides flux partitioning at 30-minute intervals, there are other open questions in land-atmosphere exchange that require high frequency iso-flux measurements. For example; how do intermittent cloud patterns impact the partial fluxes of $H_2O$ and $CO_2$ at

the second to minute scale? We know from previous investigations that land-atmosphere exchange behaves strongly non-linearly in such situations (Vilà-Guerau de Arellano et al., 2020). For example, continuous half shade results in much different exchange rates than alternating full shade and clear skies. In future work we aim to get to the core of such non-linearity's by making high time-resolution iso-flux measurements using a combination of high frequency isotopologue measurements and laser scintillometry (Van Kesteren et al., 2013; Vilà-Guerau de Arellano et al., 2019). With properly constrained auxiliary variables such measurements may allow us to derive partitioned minute scale fluxes of $H_2O$ and $CO_2$.

A first quantification of how isotope exchange behaves at short timescales is presented in Fig. 10 in the form of a quadrant analysis (Shaw et al. (1983)). Here, the co-varying perturbations constituting the fluxes are plotted. This allows for dynamics and background patterns of the exchange to be investigated, such as the contributions of specific types of eddies. Fig. 10 gives an example of such a quadrant plot during midday on the $25^{th}$ during the LIAISE campaign.

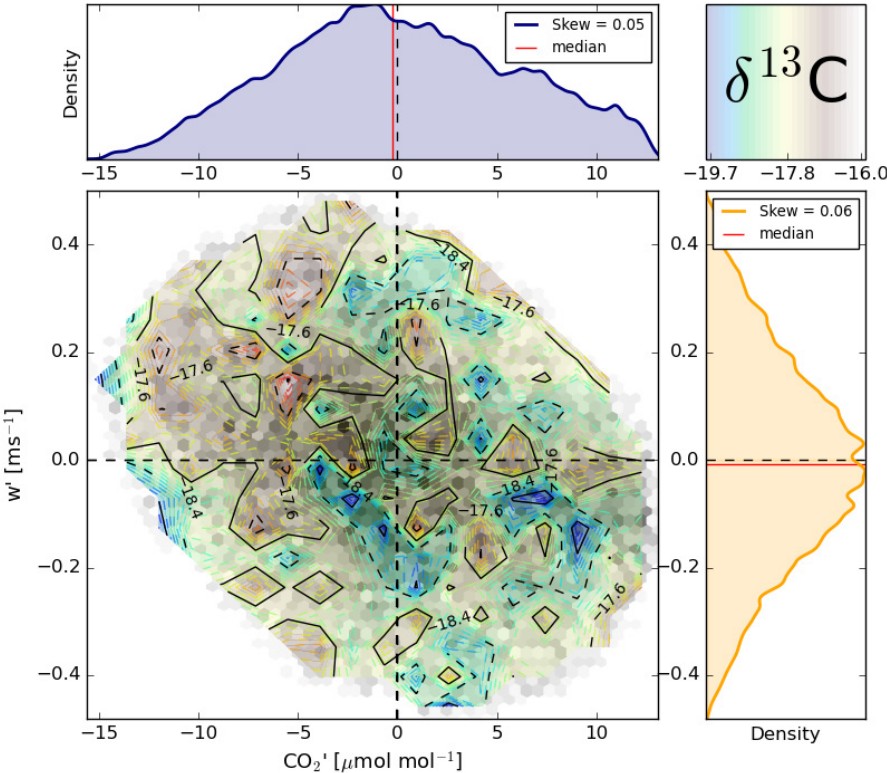

**Figure 10.** The land-atmosphere exchange of $CO_2$ within one 30 minute interval including the effect on $\delta^{13}C$. The interval was taken from July $25^{th}$ at 14:30 UTC. The isotopic composition is indicated as a coloured contour plot over the w' and $CO_2$' parameter space which itself is plotted in black. Note that only the $90^{th}$ percentile contour of the quadrant figure is shown to allow for increased detail around the relevant dense center of the plot and prevent outliers in isotopic composition, $CO_2$ concentration and vertical wind speed to become dominant. Additionally, note that the absolute value of $\delta^{13}C$ is off.

The covariance $(\overline{w'CO_2'})$ of the detrended w' and $CO_2$' signals laid out in Fig. 10 is the main component of to the net $CO_2$ flux $(\rho\,\overline{w'CO_2'})$. Clearly visible is the dynamic nature of the eddies within the 30 minute flux interval. See for example the blue blob at $0.15\ \mathrm{m\,s^{-1}}$ and $5\ \mathrm{\mu mol\,mol^{-1}}$. It is relatively depleted in $\delta^{13}C$ indicating that this air parcel has not been enriched in $\delta^{13}C$ through photosynthetic $^{12}CO_2$ uptake. In line with this, the $CO_2$ concentration of the air parcel is higher than the average at the altitude of the EC station. The air parcel however is moving up vertically towards the mixed layer, which is opposed to the flux direction. Still, on average, we find a clear pattern of depleted eddies with high $CO_2$ concentrations being carried towards the plants, and enriched eddies with reduced $CO_2$ contents being transported into the mixed layer. Individual air parcels moving upwards with reduced concentrations of $CO_2$ should indeed generally be enriched in $\delta^{13}C$. This signal is in line with the photosynthetic fractionation process we described before.

Apart from visualizing the quasi random land-atmosphere exchange, a useful feature of figures like Fig. 10 is that residual layer air entrainment signals can be recognized and separated from surface influences (Efstathiou et al., 2020). In our case, the bottom right quadrant shows some pockets of air with high $CO_2$ concentrations and low $\delta^{13}C$ values which must originate from higher up in the atmosphere. An intelligent algorithm could be designed to extract isotopic compositions of entraining air, which might be used as boundary conditions in model simulations (Lee et al., 2012; Vilà-Guerau de Arellano et al., 2019).

# 7 Best practices

Based on our experience measuring high frequency iso-fluxes using the eddy covariance technique, we provide best practice recommendations for setting up isotope flux experiments.

- Laser instruments should be installed in weatherproof environments, which are temperature stabilized to increase instrument stability. In harsh environmental conditions, climatization for the inlet and laser spectrometer pumps needs to be considered. Finally, the combined power consumption of the analysers, enclosures and pumps exceeds 4 kW for our setup, requiring special attention in remote places.

- Mole fraction calibrations are crucial, and more important than thorough span calibrations for isotope flux measurements. Mole fraction dependencies need to be corrected for to prevent artificial iso-flux contributions. The range for which the mole fraction dependencies are established should match range of mole fractions in ambient air.

- Isotopic compositions should be measured at sub second time scales and the sample flow rate must be sufficiently fast to keep the residence time of air in the sample cell short enough that the air in the cell is replaced in between measurements.

- To maximize the high frequency response of the iso-flux measurements and enable effective time lag corrections, we recommend placing the air inlet in proximity ($\sim$20 cm) of the anemometer centre, minimizing the length of the inlet tube, and minimizing air sample mixing in the inlet with short residence times and turbulent flow conditions.

■ Install an OPGA near the sonic anemometers center that measures the same molecule as the high frequency isotope analyser measures. This makes it possible to check how good the frequency responds of the isotope analysers was, and is required for correcting the isotope exchange cospectra in post-processing.

■ Liquid water formation through condensation on hydroscopic materials leads to isotopic attenuation which severely affects the results. We recommend using inlet tubing which has a smooth inner surface inner and actively heating to
575 minimize potential condensation and exchange effects. In addition, hydrophilic materials in the inlet system, such as our cellulose inlet filter, should not be used.

Additionally, we provide the following best practice recommendations for post-processing the isotope flux measurements.

■ Apply outlier filtering to the detrended scalar data before flux calculations, especially for the measured isotopic compositions. This prevents errors in the flux estimates due to fluctuations that were not induced by turbulence. We used an
580 Inter-Quantile Range (IQR) filter for outlier filtering, as described in Sec. 4.4.

■ Time alignment of the high frequency isotope analyser(s) and the high frequency anemometer measurements is a prerequisite for calculating reliable iso-fluxes. The alignment strategy we used is described in Sec. 4.2 and uses the mole fraction signals of the target molecule, measured by both the OPGA and high frequency isotope analyser, as a reference. It corrects for both the effects of instrument clock drift and inlet delays.

■ Test if there is a lag between the mole fraction and $\delta$-value signals of the isotope analyser, especially when measuring $H_2O$ isotope fluxes. This can also be done using the alignment strategy described in Sec. 4.2. If the lag is not zero, isotopic inlet attenuation likely occurred and exchange spectra are probably affected.

■ Compare the cospectra of the net exchange flux with the cospectra of the isotope exchange flux to identify if high frequency isotope flux signal was missed, as is shown in Fig. 5. If flux signal was missed, the spectral scaling method
can be used to correct for the missing high frequency signal.

■ Quantify the uncertainty inherent to the correction method in case a major contribution to the $\delta$-fluxes comes from the spectral scaling correction. In Sec. 4.4 we indicate how this uncertainty can be derived, and propagates to the $\delta$-fluxes and isotopic source compositions.

## 8 Conclusions

We have presented a methodological approach for measuring iso-fluxes during the LIAISE 2021 field campaign. The measurements encompassed six days and were supported by comprehensive auxiliary data which will support future modelling studies of biosphere-atmosphere exchange, integrating the associated isotope effects. Our analysis has introduced two key new concepts in terms of data processing. First, we have identified and quantified time lags between mole fractions and $\delta$-values, have shown how to correct for those, and have interpreted these lag times as a marker for isotopic inlet line attenuation and water condensation artefacts. We have documented large time shifts between the $H_2O$ isotopologues we measured, which has not been reported previously. The attenuation of the isotopic signal in the inlet line was likely caused by a small liquid water reservoir on the inlet filter. We demonstrate that when a time shift is detected, $\delta$-flux spectra most probably lack more high frequency contributions compared to net exchange spectra. This cannot be taken into account by previously used data processing approaches, which posit that signal loss is identical for net fluxes and $\delta$-fluxes. The second new concept is an alternative data processing approach that uses re-scaling of covariance spectra and which does allow the asymmetric signal loss to be corrected for. Finally, we have illustrated how this new spectral scaling technique would impact flux partitioning, and how it could help to simplify isotopic flux measurement setups in the future.

*Data availability.* All 30-minute interval iso-flux data and EC-flux data is publicly available at https://doi.org/10.6084/m9.figshare.23828514.v2. Furthermore, we are pleased to share the high-frequency data sets we generated. We invite all interested researchers to contact us using the correspondence email-address (r.p.j.moonen@uu.nl).

*Author contributions.* Robbert Moonen was responsible for the data aquisition, data processing, data analysis, and writing. Advanced technical support was provided by David Bonell Fontas, Getachew Adnew, and Oscar Hartogensis. Advanced scientific support was provided by Thomas Röckmann, Oscar Hartogensis, Jordi Vilà-Guerau de Arellano, and Getachew Adnew. All authors contributed to the finalization of the text.

*Competing interests.* At least one of the (co-)authors is a member of the editorial board of Atmospheric Measurement Techniques.

*Acknowledgements.* We would like to thank the NWO (OCENW.KLEIN.407) for providing funding for the CloudRoots Project (www.cloudroots.wur.nl). Also, we like to thanks the technical support and advice provided by Marcel Portanger, Paul Smeets, Henk Snelle, Roy Meinen, and Giorgio Cover. Finally, we would like to thank the staff of Aerodyne and Picarro for their support in working with their instruments.

# Appendix A

 **A1    Meteorological conditions during the measurement period**

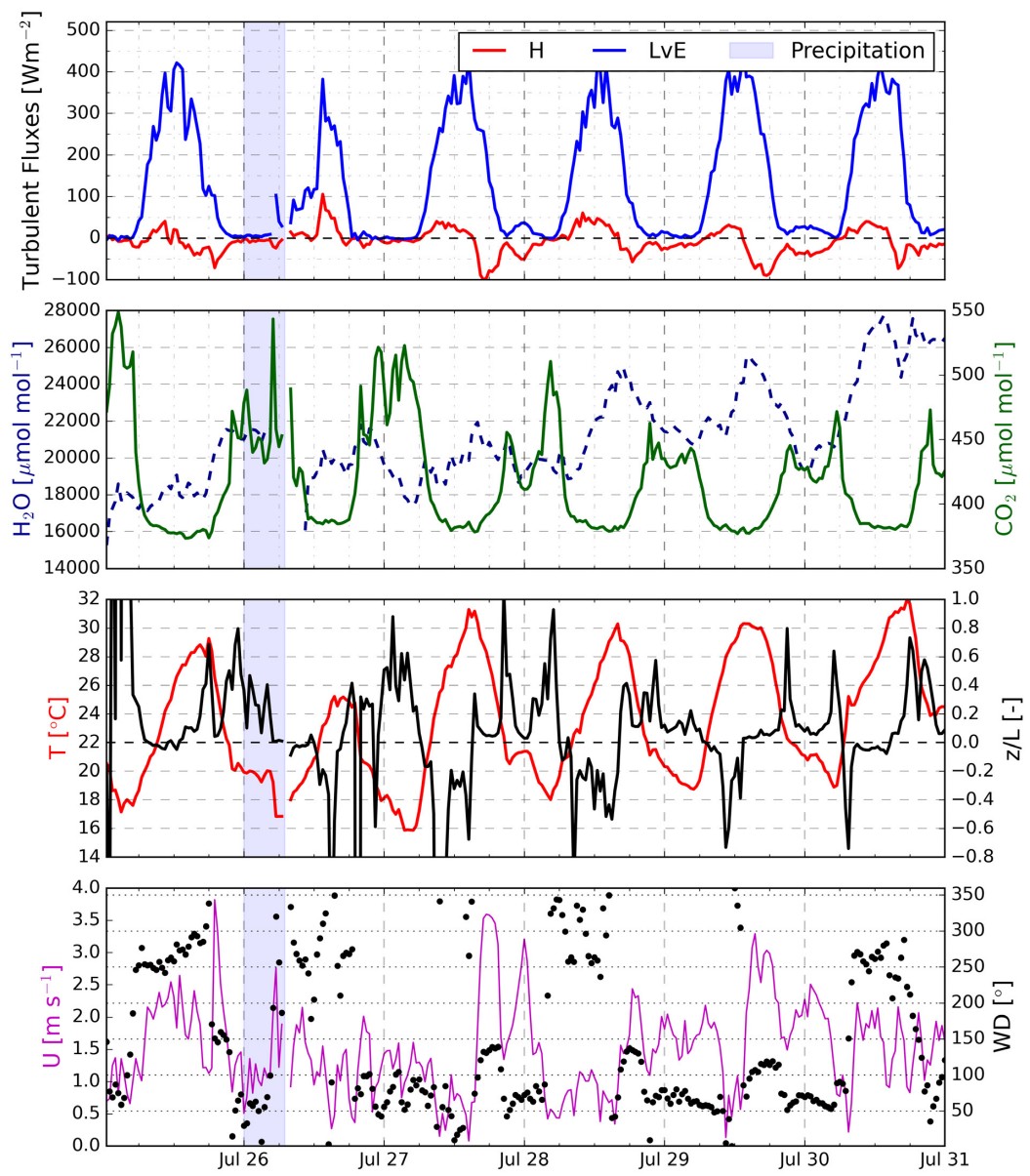

**Figure A1.** Basic meteorological variables measured using the EC-system and the OPGA. Except for the precipitation event on the $26^{th}$, relatively steady flux, stability and wind regimes can be seen. The $H_2O$ mole fraction is least stable, and increases steadily from 18000 to 26000 µmol mol$^{-1}$

.

## A2    Calibration details for both $H_2O$ and $CO_2$ laser spectrometers

**Table A1.** Detailed overview of the laser spectrometer calibrations. The $CO_2$ TILDAS-CS was calibrated using a GASMIX AIOLOS 2 (AlyTech, Juvisy-sur-Orge, France), the $H_2O$ L2130-i was calibrated with a commercial Standards Delivery Module (SDM, Picarro, Santa Clara, USA).

| | LS-H$_2$O | LS-CO$_2$ |
|---|---|---|
| **Mole fraction calibration** | | |
| **Pulse duration** | 30 min | 30 min |
| **Stable duration** | ~20min, visual inspection | = 20 min |
| **Range** | $7000 - 24000\ \mu mol\,mol^{-1}$ | $290 - 600\ \mu mol\,mol^{-1}$ |
| **Steps in mole fractions** | 3 | 4 |
| **Avg mole frac stability ($\sigma$)** | $69.2\ \mu mol\,mol^{-1}$ | $0.24\ \mu mol\,mol^{-1}$ |
| **Accepted variability ($\sigma$)** | $< 200\ \mu mol\,mol^{-1}$ | NA, stable calibrations |
| **Repeats** | 6 before campaign, 4 after | 7 after campaign |
| **Span calibration** | | |
| **Pulse duration** | 30 min | 50 min |
| **Stable duration** | ~20min, visual inspection | = 25 min |
| **Target mole fraction** | $15000\ \mu mol\,mol^{-1}$ | $400\ \mu mol\,mol^{-1}$ |
| **Steps in isotopic composition** | 2 | 2 |
| **Avg mole frac stability ($\sigma$)** | $97.9\ \mu mol\,mol^{-1}$ | $0.24\ \mu mol\,mol^{-1}$ |
| **Max variability ($\sigma$)** | $161\ \mu mol\,mol^{-1}$ | NA, stable calibrations |
| **Repeats** | 4 before campaign, 2 after | 1 after campaign |
| **Standard 1** | NL tap water (err. = se)<br>$\delta^{18}O$: -6.98 $\pm$ 0.02<br>$\delta D$: -47.12 $\pm$ 0.04 | Cylinder 1 (err. = se)<br>$\delta^{13}C$: -10.80 $\pm$ 0.02<br>$\delta^{18}O$: 39.60 $\pm$ 0.02 |
| **Standard 2** | GL icecore (err. = se)<br>$\delta^{18}O$: -30.80 $\pm$ 0.02<br>$\delta D$: -240.86 $\pm$ 0.04 | Cylinder 2 (err. = se)<br>$\delta^{13}C$: -7.21 $\pm$ 0.02<br>$\delta^{18}O$: 31.01 $\pm$ 0.02 |

## A3 Example of fitted mole fraction dependencies for the LS-H$_2$O, including a simulated leak

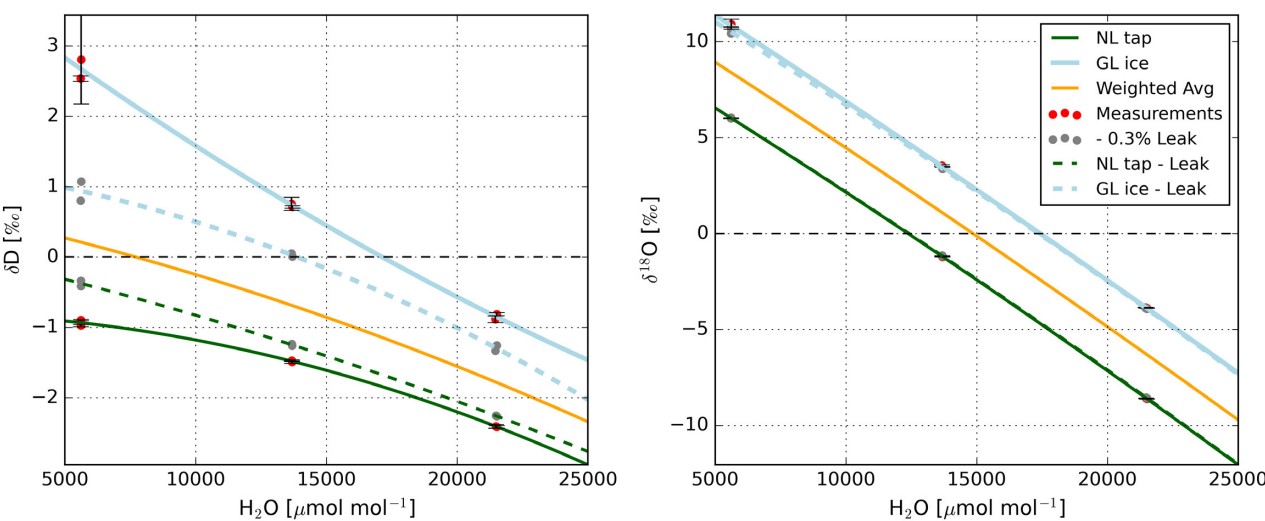

**Figure A2.** Mole fraction dependency fitted to standard measurements taken after the campaign period (see Fig.3). Note that the measurements are offset to be approximately zero on the y-axis to be able to display both standards in one figure. The solid green and blue lines indicate the fitted dependencies of each standard. To show that the discrepancy between both lines, especially for $\delta$D, is likely caused by a leak of ambient air, the dashed lines show a simulated mole fraction dependency when an ambient air leak of 0.3% is subtracted from the calibration gas flow. In this simulation, we assumed an ambient H$_2$O mole fraction of 20000 µmol mol$^{-1}$, and used the isotopic compositions of the standards and the atmospheric air given in Tab. 1. The magnitude of the ambient air leak was chosen so that it compensates for the suspected real ambient air leak and results in similar mole fraction dependencies for either calibration standard. Note that these corrected dependencies are similar to the Weighted Avg dependency we derived. The larger mole fraction dependence in $\delta^{18}$O, combined with a smaller absolute difference between the delta values of the calibration standards and ambient air (see Tab. 1), results in a much less pronounced effect of the leak in $\delta^{18}$O-H$_2$O

## A4    Temporal evolution of $\delta^{13}$C-CO$_2$ mole fraction calibrations

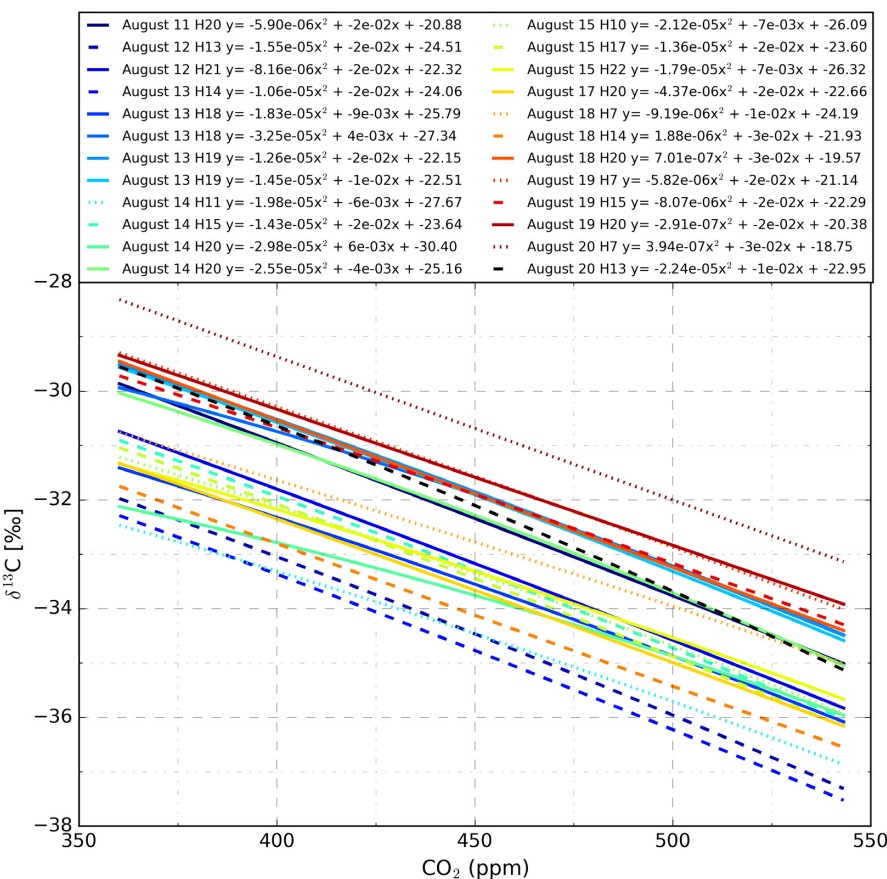

**Figure A3.** Mole fraction dependence of the CO$_2$ laser spectrometer during operation at another measurement site. As during the measurement period described in this document the isotopic composition measurements were unstable at timescales of tens-of-minutes due to instrument instability. Even though the absolute isotopic composition varied strongly over hourly timescales, mole fraction dependencies remained relatively constant during the 9 day period. The calibrations shown were made using a system that traps a calibration sample in the instruments sample cell. After flushing the sample cell with calibration gas 4 times, the final gas sample was trapped and measured for 5 minutes. Data from the last 3 minutes was averaged and used to fit calibration lines. The 3-minute plateaus were all stable below $\sigma$-CO$_2$ of 1 µmol mol$^{-1}$. This process was repeated at 4 CO$_2$ concentration levels to derive one of the plotted calibration lines.

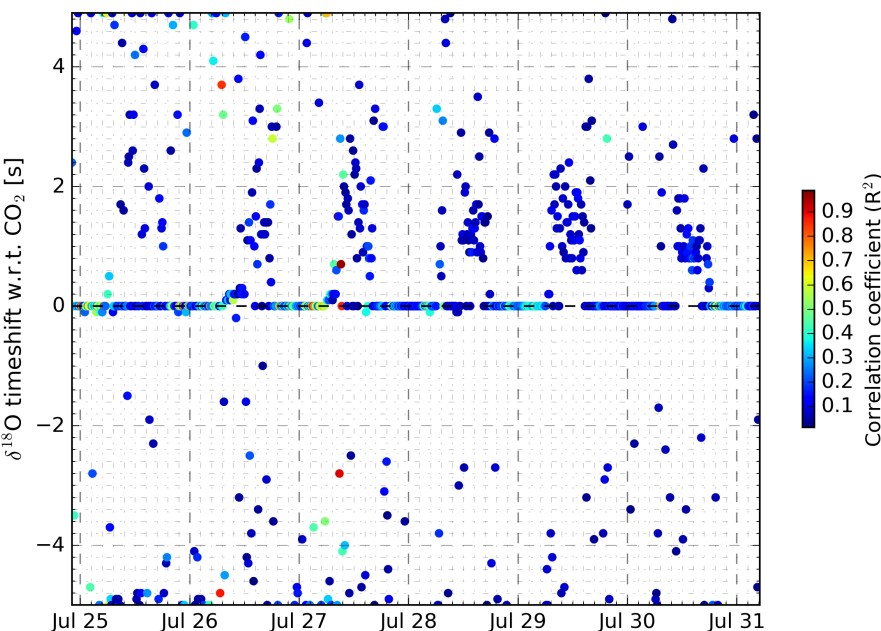

**Figure A4.** Time offset between CO$_2$ and its $\delta^{18}$O isotope ratio derived using the method described in Sect. 4.2. The colors indicate the value of the correlation coefficient. Its value is generally low due to high frequency noise in the $\delta^{18}$O signal.

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
