# Peer review of "Data treatment and corrections for estimating $H_2O$ and $CO_2$ isotope fluxes from high-frequency observations"

_EGUsphere, 2023_

## Referee Comment (RC1)

Review for „Data treatment and corrections for estimating H2O and CO2 isotope fluxes from high-frequency observations" by Robbert P.J. Moonen et al.

**General comments:**

The authors present H2O and CO2 flux data from a 6-day long field campaign in North Eastern Spain above a crop field with special focus on measuring the isotopic signal of the fluxes with the aim to gain information about ecosystem scale flux partitioning. The authors discuss the feasibility and complications arising from the combination of a high frequency eddy-covariance (EC) system with slower laser spectrometer measurements in their set-up. By addressing the isotopic composition of the fluxes, the authors tackle an underexplored way to separate the different contributors to an "easily measurable" net flux signal which are essential to validate earth system models and to identify potential model bias sources. The general reason for the current lag of establishing this method in the wider agricultural, ecological, and hydrological scientific community is the lack of high-frequency isotope measurements and the consequential need for post-processing methods to correct for the signal loss in the high-frequency domain. Adding on to a few already existing published methods, the authors present a spectral post-processing method that uses only the information stored in the lowest frequencies to correct for the loss of information in the high frequencies. This results in very high correction factors. The authors also mention some potential general set-up errors that have led to sub-optimal data quality during their campaign. The manuscript is well written and the presented data is nicely visualized. A data availability statement is missing.

With their paper the authors contribute to developing a state-of-the art method for measuring isotope fluxes which is timely and interesting for scientists from different communities. However, I am missing a general discussion on the limitations of the presented method and an analysis of a performance difference to previously published methods, an analysis on the correction factors including magnitude and dependency on environmental conditions, an evaluation of the associated uncertainty, a recommendation for "best practices" when measuring isofluxes and the mentioned but never shown supporting meteorological data; all of which would help other scientists to decide whether to use this method for their campaigns. Therefore, further analysis is needed which I will detail in the specific comments and list the few technical corrections.

**Specific comments:**

L. 12-13: The discussed "offset" in isotope and humidity measurements is commonly referred to as "memory" in isotopic studies, which is stronger for $H_2O$ than $CO_2$ measurements due to the structure of the molecules. It is also more prominent for D than for $^{18}O$ and has to be considered when designing the set-up. Please discuss what could have been done better to minimize the memory effects as much as possible (no/different filter, shorter tubes, less surface)

L. 17: As mentioned above, for stating that the method works "reliably" I am missing an uncertainty evaluation as well as a CF analysis and/or a different quality control procedure.

L. 26: Are the presented data actually interpretable as "ecosystem scale flux measurements"? More details on fetch, and the field site are needed to evaluate this.

L. 37&38: Please describe what "framework" you are referring to and it is not clear at this point which parameters are needed for measuring the "turbulent iso-flux" (what does "iso-flux" mean here)

L. 80: In section 2.2 or in the Introduction you should add more literature on the different methods used for measuring iso-fluxes and discuss their limitations and give reasons why you chose your set-up. Literature missing: Yakir & Wang 1996, Keeling 1958, Welp 2008, Wei 2015, and specifically Good et al. 2012 and 2014.

L. 110: For a general understanding of the environmental conditions during the field campaign and the discussion of the limitations of the methodology it is essential that you show the meteorological data, if not in the main text you can decide to do so in the appendix. As a minimum I would like to see the magnitude of surface fluxes, the humidity and $CO_2$ level, the atmospheric stability and a wind rose. All these parameters influence the measurements and I assume, also the data quality.

L. 125: Setup: Have you performed a dedicated lag time test while in operation to see how long it takes from the inlet until you see a signal in the spectrometers that is purley due to the pump rate end tube length. A detailed setup scheme or diagram would also be of help to identify the reason for the comparably prominent memory effects. How have you tested your setup for leaks? At a pump rate of 30 l/min the risk for leaks is high and could also explain why EC mole fraction and spectrometer data did not always align.

L. 130: A close up picture of the actual inlet setup would be beneficial for understanding the location of the filter and the inlet compared to the EC system.

L. 132: specify inner diameter and material of inlet tubing

L. 133: Was the air flow rate of 30L/min measured or just estimated? Specify where the lines were separated. A diagram might help here.

L. 161: Data treatment: The processing steps of the isotope fluxes are detailed in the manuscript but there is no mentioning of corrections, problems, data quality or similar of the original EC IRGASON data which in itself requires careful consideration. See e.g. Mauder et al. 2006. What software was used to compute the net humidity and $CO_2$ fluxes, how did you downsample the original 20Hz EC wind data to match to the spectrometer dataset?, What despiking technique was used?, Was data excluded due to unfavourable wind conditions?, How did you rotate the wind data?, What percentage of the full EC dataset can be regarded as high-quality data?

L. 162: Calibrations: There is essential information missing in the description of the calibration procedure. Specifically, you should detail what criteria were used to generate the dataset used for the calibration: In what humidity/CO2 range were you calibrating, how long were the intervals averaged for the individual calibration pulses, what was the maximum accepted mole fraction fluctuation within a pulse, how many datapoints were used for the calibrations, what is the uncertainty on these pulses, what was the uncertainty associated with the standards you used… (can go in appendix)

L. 170-173: The "however" is irritating in this sentence structure. Also, the humidity-dependency or mole fraction dependency is critical especially at low humidity levels. Therefore, it is essential that you show the humidity level during your field campaign.

Table 1 & 2: Consider merging the two tables and importantly giving all the information for both $CO_2$ and $H_2O$ calibrations, i.e. standard values used, associated uncertainty, differences in calibrations before and after campaign

L. 182 & L. 189,190: What makes you think the cross-dependency between mole fraction dependency and isotopic signal is NOT what Weng et al. 2020 observed but a leak in the system?

Figure 3: Consider taking out this figure and adding the information in the Table. It does not contain a lot of information but uses a lot of space. Instead, presenting the fit to the mole fraction data and the residuals of that fit (before and after the campaign) contains more information.

L. 198: Consider restructuring the manuscript in a way so that both "time shift corrections" (one due to clock offset and one due to memory effects) are described in the same paragraph since the same (?) method was applied to identify the time lags and it is otherwise a bit confusing to introduce the second time shift correction step later in the results section.

L. 201: When correlating two timeseries, one usually calculates the correlation coefficient r [-1,1] while R2 is a statistical parameter within linear regression that defines the goodness-of-fit. If you calculated the correlation coefficient, the adequate symbol is "r".

L. 202: "after the two timeseries are coarsely aligned". Please specify which observed atmospheric parameters you have correlated to find the specific lag times.

L. 206: Generally, how have you sub sampled? Averaged or sliced?

L. 212: Why did you choose to correlated and shift 10min data sections when you later compute the fluxes for 30min averaging periods? Does that lead to inconsistencies?

L. 198-214 & Figure 4: The whole section including Figure 4 on time shift corrections is too detailed. I assume that "lag-time correlations" is an established enough method to identify time offsets that does not require such in-detail description. Instead of showing the concept of a lag-time correlation in Figure 4, you could show the identified lag times over the period of the measurement campaign, which would visualize the "hour of day" dependency you later refer to and the linear drift.

L.224: The w timeseries for the cospectra with the slower sensors was downsampled for this. Mention this specifically and detail how?

Fig. 5: Consider introducing subplot labels instead of "left" and "right".

L. 228: As said earlier, also the OPGA needs corrections due to path averaging, detrending, wind rotations..

L. 236 ff: I encourage the authors to separately outline the different spectral correction methods used previously in the literature that they cite here and how their approach is different.

L. 243: I encourage the authors to show the figure Appendix A3 in the main manuscript next to the Figure 6 since these figures visualize the methodology explained in this manuscript. I therefore think A3 is significantly more relevant in the main article compared to e.g. the time lag figure

L. 246: What is the highest frequency your system can resolve for $CO_2$ and $H_2O$ respectively? Is this different for different isotopologues?

L. 255: the phrase "instead of using the measured d18O covariance" is confusing here since you ARE using the measured d18O covariance for the fitting procedure. Reword please.

L. 265: How was the fitting done? By minimizing what function?

L. 258: Considering that the atmospheric state influences which are the dominating eddy sizes one could think that a varying LPF frequency is beneficial to increase the

"fittable" part of the spectra whenever the atmospheric conditions are favourable for the system.

From a very practical approach, how many datapoints were available for fitting with this LPF and an integration time of 30min?

Figure 6: Please add a line at the LPF frequency to visualize in which range you are fitting and please add the unscaled wH2O co-spectrum for comparison

Eq. 8 and Line 269: For supporting your statement that the correction is "physically sound" you should present analysis and statistics on the computed correction factors and identify potential physical reasons for variations in the CF magnitude. Further, as explained earlier, an error or uncertainty estimate is needed.

L. 273: How did the CF between d18O and dD for H2O compare? Were there strong differences? Also, judging from Fig 8, panel 3, the CF for d18O in CO2 is often >1. Show analysis of CF for both CO2 and H2O systems

L. 283: How did you calculate the time lag between isotopes and mole fraction timeseries? Was there a difference between the different isotopes? This would directly influences your cospectra calculations

L. 283 & L. 290: Can you give a reason for the diurnal cycle pattern in the time lag / memory? Does it have to do with the absolute humidity level? Why is lag smallest when temperatures are high?

L. 295: But did you correct for these found offsets or not?

L. 298: What is the reason you were not able to match the timeseries during the night? Too low variability in the signal? Too high lag times? What was the humidity level and the atmospheric stratification during the night times?

Fig. 8:

  -Panel labels are missing,

  -I notice some few positive Delta d13C values in panel D. Do you trust these values?

  -What quality control screening have you implemented for the CO2 isotope dataset?

  -Top panel: Could you show agreement between Aerodyne and OPGA as scatter plot and give correlation coefficient? What software did you use to calculate the net fluxes shown in this panel?

  - panel 2 and 3: since the re-scaled data points are your final data product, consider swapping the markers for raw and re-scaled, since the light gray crosses are very subtle

  - What is the uncertainty on the data you present here?

  - There is significantly more noise visible on the signal in panel 3 compared to 2, and scaling factor and thus CF seem to be >1.

L. 302: So you did spectrally correct the CO2 isotope fluxes? This sentence is in disagreement with L. 309 and L. 337? Please clarify.

L 323-327: Have you analyzed a sample of the irrigation water for its isotopic composition?

L 236: How do the results form the H2O isotopes support this hypothesis? If the difference in Irrigation vs Precipitation isotopic composition is indeed the reason, you would expect to see a prominent difference in the ET isotopic composition after the Precipitation event compared to other days.

Fig 9:

- Top panel: a scatter plot between OPGA and Picarro LvE would be nice to see.
- Top panel: Do you have an explanation for the "W" shape of the OPGA signal when looking at one day from noon-noon?
- Also L. 345-348: Which formula/method was used to calculate OPGA scaling? Could you compare the performance of the different methods and analyze differences and explain them with meteorological conditions – the bias seems to be quite significant → maybe do this in discussion

L. 360: The paper would benefit from a clear best-practice recommendation from the authors concerning the system set-up for future field campaigns to allow for more widespread implementation.

L. 370-373: I would suggest to either show the results of these phase spectra or remove the sentences relating to it.

L 375: To judge "reliability and accuracy" an uncertainty or error analysis is missing.

L. 387: I encourage the authors to expand on the discussion between the two different correction methods, present detailed analysis and discuss causes for the differences.

L 405: In other words, you assume that turbulence is not fractionating.

L. 413: The authors have to show an analysis of these correction factors in magnitude, variability and timing and discuss possible explanations for it.

L. 417-425: This is an interesting thought but I would assume the performance of this method is highly dependent on atmospheric stability and the nature of the turbulent transport (i.e. better suitable for unstable atmospheric stratification). Please elaborate on the limitations of the presented method concerning different atmospheric and general field site conditions. If the atmospheric stability requires shorter flux integration periods (very stable atmospheres might require integration times of a few minutes only) and using the constant LPF, there could be cases where essentially no data points are available for the fitting procedure of your presented method.

Additionally: Could you somehow manipulate an existing timeseries to represent a timelag and smoothing impacted timeseries and therefore show the feasibility of your method?, i.e. no signal loss even if you only use frequencies up to the LPF.

Section 6.1: I find the presentation on the quadrant analyses very interesting and would encourage an additional figure (maybe included in Figure 10) with a complementary time window (e.g. night time) to see the differences in the iso flux behavior and the added information of such an analysis technique.

L. 458: I would argue that a time lag correlation is no new concept, please remove

Conclusions section: The whole section is missing impact and the highlighting of why this method is superior to other methods. Further, an uncertainty statement and a best practice recommendation should be given for any reader that would like to implement a similar technique in future field studies.

Figure A1: instead of showing the fitting lines color coded it would be better to show the fitting parameters as timeseries to see drift and/or variability. Details on how the data points that are the basis for these fits were generated are missing. (accepted variability, length of calibration pulses,…)

Figure A2: From line 337 I got the impression that the $CO_2$ flux data did not need additional time lag corrections? How does this compare to a lag analysis of d13C – $CO_2$ mole data?

**Technical corrections:**

L. 10: rewrite ".. analyse the measured values of central $\partial$-flux variable". What does central mean here?

L. 19: ET and NEE need to be introduced first

L. 44 and other places: The "Wahl et al." reference is missing details. Update reference to the published version https://doi.org/10.1029/2020JD034400

L. 44: Delete the Griffis 2013 citation here.

Eq. 1.: Usually, the R is defined with the isotopologue ratio instead of the isotope ratio (Mook 2000) which is also what the laser spectrometer instruments measure.

L. 66: Evaporation should not be used as exemplary process to explain "equilibrium fractionation" since it is NOT an equilibrium process alone and kinetic fractionation contributes to total fractionation. Rewrite that paragraph.

L. 119: $26^{th}$

L. 127: mention sampling frequency of Irgason

L. 131: correct "inst"

L. 135: correct grammar

L. 138: delete "m"

L. 314: insert what delta values you are giving here? D18O? of CO2 or H2O?

L. 316: Specify everywhere in the script whether you talk about d18O of CO2 or H2O.

Fig. 9 caption: The panel labels are mixed up.

L. 350: what is meant by "evaporative fractionation related to the transpiration" -reword

L. 381: Exchange "A possible cause" with "The cause"

Figure A3: Which time period is presented?, use "re-scaled CO2" for consistency in legend

**Literature:**

Good, Stephen P., Keir Soderberg, Lixin Wang, and Kelly K. Caylor. "Uncertainties in the Assessment of the Isotopic Composition of Surface Fluxes: A Direct Comparison of Techniques Using Laser-Based Water Vapor Isotope Analyzers." Journal of Geophysical Research: Atmospheres 117, no. D15 (2012). https://doi.org/10.1029/2011JD017168.

Good, Stephen P., Keir Soderberg, Kaiyu Guan, Elizabeth G. King, Todd M. Scanlon, and Kelly K. Caylor. "δ 2 H Isotopic Flux Partitioning of Evapotranspiration over a Grass Field Following a Water Pulse and Subsequent Dry Down." Water Resources Research 50, no. 2 (2014): 1410–32. https://doi.org/10.1002/2013WR014333.

Keeling, Charles D. "The Concentration and Isotopic Abundances of Atmospheric Carbon Dioxide in Rural Areas." Geochimica et Cosmochimica Acta 13 (1958): 322–34.

Mauder, Matthias, Claudia Liebethal, Mathias Göckede, Jens-Peter Leps, Frank Beyrich, and Thomas Foken. "Processing and Quality Control of Flux Data during LITFASS-2003." Boundary-Layer Meteorology 121, no. 1 (2006): 67–88. https://doi.org/10.1007/s10546-006-9094-0.

Wahl, S., H. C. Steen-Larsen, J. Reuder, and M. Hörhold. "Quantifying the Stable Water Isotopologue Exchange Between the Snow Surface and Lower Atmosphere by Direct Flux Measurements." Journal of Geophysical Research: Atmospheres 126, no. 13 (July 16, 2021): 1–24. https://doi.org/10.1029/2020JD034400.

Welp, Lisa R., Xuhui Lee, Kyounghee Kim, Timothy J. GRriffis, Kaycie A. Billmark, and John M. Baker. "δ 18 O of Water Vapour, Evapotranspiration and the Sites of Leaf Water Evaporation in a Soybean Canopy." Plant, Cell \& Environment 31, no. 9 (2008): 1214–28. https://doi.org/10.1111/j.1365-3040.2008.01826.x.

Wei, Zhongwang, Kei Yoshimura, Atsushi Okazaki, Wonsik Kim, Zhongfang Liu, and Masaharu Yokoi. "Partitioning of Evapotranspiration Using High-Frequency Water Vapor Isotopic Measurement over a Rice Paddy Field." Water Resources Research 51, no. 5 (2015): 3716–29. https://doi.org/10.1002/2014WR016737.

Yakir, Dan, and Xue Feng Wang. "Fluxes of CO2 and Water between Terrestrial Vegetation and the Atmosphere Estimated from Isotope Measurements." Nature 380, no. 6574 (April 11, 1996): 515–17. https://doi.org/10.1038/380515a0.

---

## Author Comment (AC1)

**Rebuttal**

*Data treatment and corrections for estimating $H_2O$ and $CO_2$ isotope fluxes from high frequency observations.*

Robbert P.J. Moonen, Getachew A. Adnew, Oscar K. Hartogensis, Jordi Vilà -Guerau de Arellano, David J. Bonell Fontas, and Thomas Röckmann

We thank the referees and the editor for their insightful comments and encouraging the resubmission. To give an organized response to all comments, we grouped the remarks of all referees by topic. The remaining points will be dealt with separately. After each point, we specify how we modified the manuscript. Our author replies appear in blue, changes to the final manuscript appear in red colour. Figures in the rebuttal use alphabetic numbering, in contrast to the numerical numbering in the manuscript.

**General evaluations**

**Summary referee #1**

The authors present $H_2O$ and $CO_2$ flux data from a 6-day long field campaign in North Eastern Spain above a crop field with special focus on measuring the isotopic signal of the fluxes with the aim to gain information about ecosystem scale flux partitioning. The authors discuss the feasibility and complications arising from the combination of a high frequency eddy-covariance (EC) system with slower laser spectrometer measurements in their set-up. By addressing the isotopic composition of the fluxes, the authors tackle an underexplored way to separate the different contributors to an "easily measurable" net flux signal which are essential to validate earth system models and to identify potential model bias sources. The general reason for the current lag of establishing this method in the wider agricultural, ecological, and hydrological scientific community is the lack of high-frequency isotope measurements and the consequential need for post-processing methods to correct for the signal loss in the high-frequency domain. Adding on to a few already existing published methods, the authors present a spectral post-processing method that uses only the information stored in the lowest frequencies to correct for the loss of information in the high frequencies. This results in very high correction factors. The authors also mention some potential general setup errors that have led to sub-optimal data quality during their campaign. The manuscript is well written and the presented data is nicely visualized. A data availability statement is missing. With their paper the authors contribute to developing a state-of-the art method for measuring isotope fluxes which is timely and interesting for scientists from different communities. However, I am missing a general discussion on the limitations of the presented method and an analysis of a performance difference to previously published methods, an analysis on the correction factors including magnitude and dependency on environmental conditions, an evaluation of the associated uncertainty, a recommendation for "best practices" when measuring isofluxes and the mentioned but never shown supporting meteorological data; all of which would help other scientists to decide whether to use this method for their campaigns. Therefore, further analysis is needed which I will detail in the specific comments and list the few technical corrections.

**Summary referee #2**

Based on combined eddy-covariance and fast isotope measurements (with 2 different instruments) over flood irrigated alfalfa, the authors report case findings and methodological step innovations (especially a revision of an empirical, required response correction for the isotope timeseries).

General comments

The manuscript well matches the scope of AMT and contributes to an important task, i.e. bringing forward combined isotope and EC measurements towards the point where they might eventually enable robust, continuously operational flux source partitioning. As far as the reviewer (who is more familiar with EC and micrometeorology than isotopes) can judge, the study design and analysis were thorough and state-of-the-art. The text and especially the figures are elaborated with care and mostly clear. A mixture of unfortunate

small surprises (e.g. the suspected ambient air leak) and possibly oversights (if not overlooked, no actual partitioned fluxes are shown, which in turn might be due to the fact that no reference measurements of the isotopic composition of e.g. transpiration and evaporation water are mentioned) seems to have prevented the study from being a full success story about isotope-based EC flux partitioning and thus leave it somewhat behind the "cutting edge" of this area. It would be helpful for the readership to learn more clearly whether or not a complete source partitioning was the original aim and if so, what where the main obstacles. However, since mixed outcomes like this still appear to be the rule (the publication of which should help the community to learn about the prospects and pitfalls of the methodology) rather than the exception, and since the authors share interesting findings and innovations, it is definitely worthwhile publishing in AMT. That said, some issues mostly with the presentation require at least minor revisions. The style of documenting the methodology and offering explanations for findings has a slight tendency to be too "anecdotal", e.g. when mentioning a "reasonably short" inlet, claiming the insensitivity of results to the low-pass filter or using in the discussion's line of reasoning something that was looked at during data analysis but does not seem to be presented in the manuscript. More details on these examples and a few more are given below, but I would like to ask the authors to thoroughly screen the whole manuscript for such style, and keep it to a minimum - i.e. remove unnecessary undefended statements, defend (e.g. with data, calculations or appropriate citations) the necessary ones, and clarify the speculative status of the remaining ones.

Thanks again for the positive words and valuable suggestions. The point on data availability does not come back in the specific comments, so we note here that we have added a data availability statement. All other points mentioned in the general comments are answered in the specific comments below.

**Time lag**

> **(Ref #1)** L. 198: Consider restructuring the manuscript in a way so that both "time shift corrections" (one due to clock offset and one due to memory effects) are described in the same paragraph since the same (?) method was applied to identify the time lags and it is otherwise a bit confusing to introduce the second time shift correction step later in the results section.

> **(Ref #1)** L. 198-214 & Figure 4: The whole section including Figure 4 on time shift corrections is too detailed. I assume that "lag-time correlations" is an established enough method to identify time offsets that does not require such in-detail description. Instead of showing the concept of a lag-time correlation in Figure 4, you could show the identified lag times over the period of the measurement campaign, which would visualize the "hour of day" dependency you later refer to and the linear drift.

Although lag time corrections have been used in various previous studies, we explained the method in detail for two reasons. First, the lag time between the isotopic signals and the related mole fractions - indeed caused by a memory effect - is a new and interesting finding, which requires a description on how the time delays were determined. As we think the time lag should be treated as a *result*, we had to split the topic over the methods and results sections. Secondly, we time-aligned instruments with different measurement frequencies, which as far as we are aware is not common. From line 205 onward, we specify our approach to deal with this frequency discrepancy, where the correlation coefficient can only be derived after subsampling the EC data to the same frequency as the laser instruments for measuring isotopic compositions (4 & 10Hz).

> **(Ref #1)** L. 12-13: The discussed "offset" in isotope and humidity measurements is commonly referred to as "memory" in isotopic studies, which is stronger for $H_2O$ than $CO_2$ measurements due to the structure of the molecules. It is also more prominent for D than for $^{18}O$ and has to be considered when designing the set-up. Please discuss what could have been done better to minimize the memory effects as much as possible (no/different filter, shorter tubes, less surface)

Thanks for the suggestion. We have added that the lag time is caused by the isotopic memory effect. In a new *best practices* section that we added to the manuscript, we recommend how memory effects can best be avoided (see below). In addition, we discuss the likely cause for the memory effect in the discussion section.

We have applied the following changes to the text:
Furthermore, we find that mole fractions and isotope ratios measured with the same instrument can be offset in time by more than a minute for the $H_2O$ isotopologues due to an isotopic memory effect.

*(Ref #1) L. 201: When correlating two timeseries, one usually calculates the correlation coefficient r [-1,1] while R2 is a statistical parameter within linear regression that defines the goodness-of-fit. If you calculated the correlation coefficient, the adequate symbol is "r".*

Thanks for spotting this. We have changed the symbol from 'R2' to 'r' in both the text and the figure.

[Figure]

*Figure A (Figure 4 in manuscript): Example of the time shift algorithm that optimizes the correlation coefficient r. In this case the shift between the EC station and the L2130-i laser spectrometer is determined using the $H_2O$ mole fraction measurements of both instruments. The different colors each represent a 10-minute section of data collected on July 25th, as indicated in the legend. Each dashed vertical line indicates the optimal time shift between the two 10-minute data sections. The derived time shift values are provided in the legend. A negative time shift indicates that the laser spectrometer is delayed compared to the EC system. Note that during this period the instrument clocks were drifting approximately linearly with respect to each other.*

*(Ref #1) L. 202: "after the two timeseries are coarsely aligned". Please specify which observed atmospheric parameters you have correlated to find the specific lag times.*

The mole fraction of the respective $CO_2$ or $H_2O$ laser spectrometer was correlated to the mole fraction signal from the OPGA. To keep the text general, i.e. applicable to both $CO_2$ or $H_2O$, we did not specify the variable being correlated. We realize that it is useful to specify the information and in the revised version we have made it explicit that for aligning the isotope analysers, the respective mole fraction signals were used. For the "isotope – mole fraction" shift described in section 5.1, the correlated variables are explicitly mentioned.

We have applied the following changes to the text:

Old: *In order to synchronize two high frequency time series of related atmospheric variables in post processing, we used a correlation coefficient (R2) based alignment scheme that works as follows (similar to Fan et al. (1990)).*

New: Since both instrument types measured the mole fraction of $CO_2$ or $H_2O$, these mole fraction measurements were used to time align the high frequency time series in post processing using a correlation coefficient (r) based alignment scheme that works as follows (similar to Fan et al., 1990).
* * *
**(Ref #1)** *L. 206: Generally, how have you sub sampled? Averaged or sliced?*
* * *
We say "sub-sampled" as in we took individual samples than align with the lower frequency isotope measurements. If we would have averaged, we would have called it a (moving) average operation.

We have added the following information to clarify our subsampling approach:
Note that for deriving r, the high frequency time series should be sub-sampled to the frequency of the low frequency time series. We used masking (or indexing / slicing) for sub-sampling to drop the data points in the high frequency time series that do not align with the low frequency time series.
* * *
**(Ref #1)** *L. 212: Why did you choose to correlated and shift 10min data sections when you later compute the fluxes for 30min averaging periods? Does that lead to inconsistencies?*
* * *
Since the lag times can vary considerably over the day, the 30 min periods lead in some cases to considerable time lag jumps from one half hour to the next, e.g. 10+ second for the mole fraction – isotope lag time. The 10-minute sections are a compromise between having enough data to establish reliable time lags based on the 'r', and preventing large time lag jumps.
For periods where lag times are more stable over time, calculating half hourly lag times would be sufficient, but 10-minute lag times give similar results, so we choose 10 min sections to stay consistent with the lag time determination.
After applying the time shifts, some duplicate values may be present in the 30 min flux interval data. While that can affect the highest frequency flux contributions, we believe it leads to smaller inconsistencies compared to shifting the 30 minute blocks, which would result in correlating "stretched or shrunk" time series.
* * *
**(Ref #1)** *L. 246: What is the highest frequency your system can resolve for $CO_2$ and $H_2O$ respectively? Is this different for different isotopologues?*
* * *
We mentioned the specifications of the isotope enabled laser spectrometers, including measurement frequencies (4 and 10 Hz for $H_2O$ and $CO_2$ respectively) in section 3.2. Mole fractions and isotopic compositions are measured at the same frequency by the laser spectrometers. We added the EC and OPGA measurement frequencies (20 Hz) to section 3.2. Note that the sample throughput rates match and/or exceed the measurement frequencies.

We have applied the following changes to the text:
", which combines a Sonic Anemometer with an Open Path Gas Analyser (OPGA) sampling at 20 Hz (Fig. 2)."
* * *
**(Ref #1)** *L. 283: How did you calculate the time lag between isotopes and mole fraction timeseries? Was there a difference between the different isotopes? This would directly influence your cospectra calculations.*
* * *
Thanks for pointing out that this was not clearly explained, we have adapted the text as follow:

We used the same time alignment strategy described in Sect. 4.2 for synchronizing the laser spectrometers with the Eddy Covariance system, to investigate the lag time of isotopic signals with respect to the mole fractions.

Fig. 7 shows the pattern of the time lag between δD and $H_2O$ over the measurement period. δ18O displays a near identical behaviour with approximately 20% smaller lag times.

> *(Ref #1) L. 283 & L. 290: Can you give a reason for the diurnal cycle pattern in the time lag / memory? Does it have to do with the absolute humidity level? Why is lag smallest when temperatures are high?*

We discuss the causes of the lag times in section 6. Following the referee suggestion, we have now explicitly mentioned relative humidity. Note that we use relative humidity in our argument rather than absolute humidity as it is more related to the problem at hand, i.e. the condensation of water.

We have applied the following changes to the text:
Likely, the hydrophilic nature of the material allowed for a small liquid water reservoir to form, which caused the isotopic exchange and consequent lag times that we observed in our data (Reishofer et al., 2022). The diurnal cycle in lag times would suggest a variable reservoir size, positively related to the atmospheric relative humidity which is generally lower during daytime compared to night-time.

> *(Ref #1) L. 295: But did you correct for these found offsets or not?*
>
> *(Ref #1) Figure A2: From line 337 I got the impression that the $CO_2$ flux data did not need additional time lag corrections? How does this compare to a lag analysis of d13C – $CO_2$ mole data?*

Yes, we corrected the large offsets found in the $H_2O$ isotopic signal as stated in L. 298. Also, we mentioned that we applied this correction before the flux or spectral calculations. $\delta^{18}O$-$CO_2$ time lags were zero 90% of the time, with occasional positive midday time lags 20 times smaller compared to the $H_2O$ time lags, which is why we did not correct for it.

We made the following changes to the text:
The $CO_2$ signal and its δ-values do not suffer from a time offset in their signals. For $\delta^{13}C$, the lag time is 0 for the entire data-set, with few outliers. $\delta^{18}O$ has time lags of zero 90% of the time, with more outliers and occasional midday time lags of only 1-2 seconds, which we chose not to correct for (Appendix A2).

> *(Ref #1) L. 298: What is the reason you were not able to match the timeseries during the night? Too low variability in the signal? Too high lag times? What was the humidity level and the atmospheric stratification during the night times?*

Lag times were indeed longer during night-time, likely due to the increased relative humidity. On top of that, stable conditions reduced the variability of $H_2O$ and its isotopologues in the atmosphere, mostly at longer timescales. These meteorological conditions are shown in Figure B of the rebuttal and have been added to Appendix A1 of the manuscript as described in the Eddy Covariance section of the rebuttal. Smaller, short timescale fluctuations with larger than normal lag times resulted in small correlation coefficients and unrealistic differences between subsequent time lags.

We adjusted the text to include the reasons for discarding the night-time data and refer to the described meteorological conditions in the appendix.
L. 298; On top of this, we discarded nighttime data as no reliable time shift could be found due to limited natural signal variability during stable atmospheric conditions and large lag times associated with high relative humidity levels (Appendix A1).

> *(Ref #1) L.224: The w timeseries for the cospectra with the slower sensors was down sampled for this. Mention this specifically and detail how?*

We have adjusted the text as follows:
L. 213: By adjusting the time for each 10-minute section and sub-sampling the data for the higher frequency sensors we constructed one data set of uniform frequency, unaffected by data logger clock drift or inlet line related time lags.

**Calibrations**

> **(Ref #1)** *L. 162: Calibrations: There is essential information missing in the description of the calibration procedure. Specifically, you should detail what criteria were used to generate the dataset used for the calibration: In what humidity/CO₂ range were you calibrating, how long were the intervals averaged for the individual calibration pulses, what was the maximum accepted mole fraction fluctuation within a pulse, how many datapoints were used for the calibrations, what is the uncertainty on these pulses, what was the uncertainty associated with the standards you used… (can go in appendix)*

We have added the following table (Table A1) to the Appendix with the requested calibration details and refer to it in section 4.1, which deals with the calibrations. We note that due to several delays in delivery, the instrument housings arrived (CO₂) or were completed (H₂O) only days before we had to leave for the LIAISE campaign which by then had already started. Therefore, there was no time to build systems in the enclosures for in-field calibrations, and the calibrations were only performed before and after the campaign. Having said that, the absolute calibration of the isotope values is less important for the derivation of the iso-fluxes, where only the fluctuations around the mean of the flux interval (30 min) is used.

**Table A1.** Detailed overview of the laser spectrometer calibrations. The $CO_2$ TILDAS-CS was calibrated using a GASMIX AIOLOS 2 (AlyTech, Juvisy-sur-Orge, France), the $H_2O$ L2130-i was calibrated with a commercial Standards Delivery Module (SDM, Picarro, Santa Clara, USA).

| | LS-H₂O | LS-CO₂ |
|---|---|---|
| **Mole fraction calibration** | | |
| **Pulse duration** | 30 min | 30 min |
| **Stable duration** | ~20min, visual inspection | = 20 min |
| **Range** | $7000 - 24000\ \mu mol\ mol^{-1}$ | $290 - 600\ \mu mol\ mol^{-1}$ |
| **Steps in mole fractions** | 3 | 4 |
| **Avg mole frac stability ($\sigma$)** | $69.2\ \mu mol\ mol^{-1}$ | $0.24\ \mu mol\ mol^{-1}$ |
| **Accepted variability ($\sigma$)** | $< 200\ \mu mol\ mol^{-1}$ | NA, stable calibrations |
| **Repeats** | 6 before campaign, 4 after | 7 after campaign |
| **Span calibration** | | |
| **Pulse duration** | 30 min | 50 min |
| **Stable duration** | ~20min, visual inspection | = 25 min |
| **Target mole fraction** | $15000\ \mu mol\ mol^{-1}$ | $400\ \mu mol\ mol^{-1}$ |
| **Steps in isotopic composition** | 2 | 2 |
| **Avg mole frac stability ($\sigma$)** | $97.9\ \mu mol\ mol^{-1}$ | $0.24\ \mu mol\ mol^{-1}$ |
| **Max variability ($\sigma$)** | $161\ \mu mol\ mol^{-1}$ | NA, stable calibrations |
| **Repeats** | 4 before campaign, 2 after | 1 after campaign |
| **Standard 1** | NL tap water (err. = se) | Cylinder 1 (err. = se) |
| | $\delta^{18}O$: -6.98 ± 0.008 | $\delta^{13}C$: -10.80 ± 0.0046 |
| | $\delta D$: -47.12 ± 0.018 | $\delta^{18}O$: 39.60 ± 0.0044 |
| **Standard 2** | GL icecore (err. = se) | Cylinder 2 (err. = se) |
| | $\delta^{18}O$: -30.80 ± 0.008 | $\delta^{13}C$: -7.21 ± 0.0052 |
| | $\delta D$: -240.86 ± 0.018 | $\delta^{18}O$: 31.01 ± 0.0050 |

> *(Ref #1) L. 182 & L. 189,190: What makes you think the cross-dependency between mole fraction dependency and isotopic signal is NOT what Weng et al. 2020 observed but a leak in the system?*

> *Figure 3: Consider taking out this figure and adding the information in the Table. It does not contain a lot of information but uses a lot of space. Instead, presenting the fit to the mole fraction data and the residuals of that fit (before and after the campaign) contains more information.*

> *(Ref #2) L186: Unclear statement (especially the "is described by") - do you want to say that assuming a 3 to 1 ratio approximately optimized the ability of a NL-tap-water to GL-icecore "virtual mixture" to mimick the atm measurement from Table 1?*

Figure 3 is intended to clarify why we believe that the cross dependency is the consequence of a leak and not of the effect Weng et al. (2020) described. We believe that a table cannot convey that same message. During mole fraction calibrations, synthetic air was mixed with water vapour in various ratios. We found that, especially for Deuterium, calibration curves varied dependent on the standard we used. This could be due to either the effect described by Weng et al. (2020) or a leak in the system during calibrations. We attribute the effect to a leak for the following reason. A leak adds a constant fraction of atmospheric air with its respective water content and isotopic composition. In that case, the dryer the calibration gas stream, the stronger the influence of the ambient air, generating an artificial mole fraction dependence. This artificial dependence should disappear in a situation where the isotopic composition of the reference gas stream and atmospheric water are identical. The virtual calibration coefficients in fact reproduced that scenario. Importantly, the virtual coefficients are near identical before and after the campaign, while the measured coefficients were significantly different, (Fig 3). If the effect Weng et al. describe were dominant, this similarity would be coincidental, while if a leak were the cause, the similarity is exactly what would be expected. We therefore believe that the result supports the theory of a leak.

In the text we describe this as follows:
The origin could not be precisely identified but we suspect that a small leak of ambient air during calibrations could cause this issue. In case of a leak, the ratio of ambient to calibration vapour will differ dependent on $H_2O$ concentration, and thus affect the measured isotopic composition. To reduce the effect of this contamination on the coefficients of the calibration fit, we interpolated the coefficients linearly between the two standards to approximate the isotopic composition of the atmosphere. A 3 : 1 mixture of "NL tap water" and  1 "GL icecore" has a similar isotopic composition as the atmospheric water vapour and was thus used to derive the coefficients applied to the campaign data. We suggest that the similarity in the "Weighted Avg" (yellow) calibration coefficients "Before" and "After" the campaign in Fig. 3 is no coincidence, but a feature of an ambient air leak of variable magnitude. An instrument related cross dependency of the isotopic composition on the mole fraction dependence, as described by Weng et al. (2020), would not cause such a similarity.

> *(Ref #1) L. 170-173: The "however" is irritating in this sentence structure. Also, the humidity dependency or mole fraction dependency is critical especially at low humidity levels. Therefore, it is essential that you show the humidity level during your field campaign. Table 1 & 2: Consider merging the two tables and importantly giving all the information for both $CO_2$ and $H_2O$ calibrations, i.e. standard values used, associated uncertainty, differences in calibrations before and after campaign.*

We have deleted "however"
in the lab before and after the campaign. , Mole fraction calibrations are key for

In addition, we added the information on the meteorological situation to the appendix (Figure C). Details on the calibrations, including the humidity range over which the instrument was calibrated are shown in the table above which is included in the appendix of the revised manuscript.

**Eddy covariance**

> *(Ref #1) L. 26: Are the presented data actually interpretable as "ecosystem scale flux measurements"? More details on fetch, and the field site are needed to evaluate this.*

> *(**Ref #1**) L. 110: For a general understanding of the environmental conditions during the field campaign and the discussion of the limitations of the methodology it is essential that you show the meteorological data, if not in the main text you can decide to do so in the appendix. As a minimum I would like to see the magnitude of surface fluxes, the humidity and $CO_2$ level, the atmospheric stability and a wind rose. All these parameters influence the measurements and I assume, also the data quality.*

> *(**Ref #1**) L. 161: Data treatment: The processing steps of the isotope fluxes are detailed in the manuscript but there is no mentioning of corrections, problems, data quality or similar of the original EC IRGASON data which in itself requires careful consideration. See e.g. Mauder et al. 2006. What software was used to compute the net humidity and $CO_2$ fluxes, how did you downsample the original 20Hz EC wind data to match to the spectrometer dataset?, What despiking technique was used?, Was data excluded due to unfavourable wind conditions?, How did you rotate the wind data?, What percentage of the full EC dataset can be regarded as high-quality data?*

> *(**Ref #1**) L. 228: As said earlier, also the OPGA needs corrections due to path averaging, detrending, wind rotations..*

> *(**Ref #2**) L361-364: For example, given this interesting question it would have been interesting to quantify the effect of the different correction methods on partitioned fluxes (e.g. on evaporation vs. transpiration). Open-path gas analyzer measurements are also still subject to spectral loss that requires correction (e.g. path averaging effect), is this applied or considered in any way (see comment on L141)?*

Figure B was added to the appendix of the manuscript to display the general meteorological conditions. It shows an early onset of negative sensible heat fluxes in the afternoon, providing energy to the still positive evapotranspiration flux. Furthermore, we observed high humidity levels (close to 30.000 ppm) associated with irrigation in the surroundings. $CO_2$ concentrations followed an expected diurnal cycle with a photosynthesis induced low during midday. Atmospheric stability levels, indicated by z/L, show a diurnal cycle with stable conditions during night-time, and unstable conditions during the morning to early afternoon. As in H, the afternoon transition to a stable temperature stratification is visible. Finally, the wind regime that changes direction diurnally is displayed (Figure B).

[Figure]

*Figure B (Appendix A1 in manuscript): Basic meteorological variables measured using the EC-system and the OPGA. Except for the precipitation event on the 26th, fluxes, stability and wind regimes were relatively stable. The H₂O shows a gradual increase on top of the diurnal variability from 18000 to 26000 μmol mol⁻¹.*

A Wind Rose (Figure C) of the 6 campaign days indicates the variable wind regime in an alternative way. Westerly winds occurred during daytime and easterly winds during night-time like we described in the site description, and display in Figure B.

[Figure]

*Figure C (Not included in manuscript): Wind rose indicating the bidirectional wind pattern from the east and west during the measurement period*

To decide if we should eliminate certain wind sectors, we performed a 1D footprint analysis using the Kljun et al. (2004) footprint model implemented in the EddyPro (LI-COR, Inc, Lincoln, U.S.A.) software (see Figure D). The analysis indicates that given the 2.45 m height of the EC-station, 90% of the flux contribution originated from within an 85-meter radius around the measurement setup during the entire campaign. Standard errors were only included when multiple footprint estimates were returned for one time interval by EddyPro. Importantly, our measurements were taken from the center of a highly homogeneous 300 x 400 m alfalfa field. As the horizontal extent of the field is much larger than the 90% footprint boundary of 85 m, we conclude that we are indeed measuring at the field scale, and no wind sectors need to be excluded based source location of the fluxes we measured.

[Figure]

*Figure D (Not in manuscript): 1D footprint analysis output of EC-station during the measurement period.*

To keep the manuscript concise, we decided not to include Figures C & D. We did add a description of their contents and we note that the last panel of our new Figure B depicts the wind data as a timeserie.

We used EddyPro version 7.06 (Fratini & Mauder, 2014) from LI-COR Inc (Lincoln, U.S.A.) to output level 6 processed raw data of the IRGASON dataset which includes corrections for raw data screening including spike removal (Vickers & Mahrt, 1997) and axis rotation with the planar-fit procedure (Wilczak et al., 2001). Processing of the high frequency data to fluxes was done using our own code to be able to include our own spectral corrections. By keeping the flux processing chain equal between OPGA and LS's we ensure that all differences were instrument related. The raw scalar data from the OPGA and LS's were detrended and subsequently filtered for outliers using an InterQuartile Range (IQR) filter of 2.5 the interquartile range deviation from the mean interval value. For the flux calculations, we applied density and sonic T corrections. No spectral flux corrections (as described in e.g. Moore, 1986) were applied to prevent interference with our own spectral correction method described in Sect 4.3.2 (note that raw data can't be regenerated from a corrected cospectrum). Moreover, the OPGA and spectral scaling corrections we find for the δ-fluxes (Sect 5.3) are an order magnitude larger compared to the 5-10% loss a spectral flux correction compensates for (Foken, 2008).

The down sampling technique we used is now specified in the text as described in our replies to the comments on "lag time" above.

We have applied the following changes to the text:
We added Figure B to the appendix and references to it in the site description (Sect. 3.1)

L. 114: Iso-flux measurements were made in the middle of a 300 x 400 m field with flood irrigated alfalfa (C3)

L. 129: "on a tripod at 2.45 m above ground level and faced South (180◦).
We used EddyPro version 7.06 (Fratini & Mauder, 2014) from LI-COR Inc (Lincoln, U.S.A.) to output level 6 processed raw data of the IRGASON dataset which includes corrections for raw data screening including spike removal (Vickers & Mahrt, 1997) and axis rotation with the planar-fit procedure (Wilczak et al., 2001). Processing of the high frequency data to fluxes was done using our own code to be able to include our own spectral corrections. By keeping the flux processing chain equal between OPGA and LS's we ensure that all differences were instrument related. The raw scalar data from the OPGA and LS's were detrended and subsequently filtered for outliers using an InterQuartile Range (IQR) filter of 2.5 the interquartile range deviation from the mean interval value. For the flux calculations, we applied density and sonic T corrections. No spectral flux corrections (as described in e.g. Moore, 1986) were applied to prevent interference with our own spectral correction method described in Sect 4.3.2 (note that raw data can't be regenerated from a corrected cospectrum). Moreover, the OPGA and spectral scaling corrections we find for the δ-fluxes (Sect 5.3) are an order magnitude larger compared to the 5-10% loss a spectral flux correction compensates for (Foken, 2008).
A 1D flux footprint analysis performed using the Kljun et al. (2004) footprint model implemented in EddyPro indicated that 90% of the flux originated from within 85 m radius around the measurement setup, confirming that fluxes from all wind sectors originated from within the alfalfa field. 20 cm below the anemometer's center an inlet line continuously sampled atmospheric air for analysis "

**Quality control and error estimation**

| |
|---|
| *(Ref #1) What quality control screening have you implemented for the CO₂ isotope dataset?*
 We filtered for extreme outliers using a window of acceptable isotopic compositions over the entire campaign. This range was 300 : 650 for CO₂, -40 : -5 for d13C, and 5 : 40 for d18O. Note that the broad range for the isotopic compositions was due to the instability of the instrument described in the manuscript. Additionally, instances where we worked on the instruments and/or opened the enclosures were flagged and removed.
 In the revised manuscript, we added an additional InterQuartile Range (IQR) outlier filter with a bound from the mean to +/- 2.5 times the range from q1 to q3 for cleaning all detrended turbulent variables including the isotopic signals of both CO₂ and H₂O. The overall results did not change much due to this addition, while some individual flux datapoints did change magnitude significantly.
 After these filtering steps, flux intervals with less then 75% of the expected raw data present were eliminated from the analysis.
 On top of minimizing errors due to spikes in the data, we minimized the systematic errors in delta fluxes due to missed high frequency contributions with the spectral scaling detailed in section 4.3.2. |
| *(Ref #1) L. 17: As mentioned above, for stating that the method works "reliably" I am missing an uncertainty evaluation as well as a CF analysis and/or a different quality control procedure.* |
| *(Ref #1) Eq. 8 and Line 269: For supporting your statement that the correction is "physically sound" you should present analysis and statistics on the computed correction factors and identify potential physical reasons for variations in the CF magnitude. Further, as explained earlier, an error or uncertainty estimate is needed.* |
| *(Ref #1) L 375: To judge "reliability and accuracy" an uncertainty or error analysis is missing.* |
| *(Ref #1) L. 413: The authors have to show an analysis of these correction factors in magnitude, variability and timing and discuss possible explanations for it.* |
| *(Ref #1) Fig. 9 - What is the uncertainty on the data you present here?* |
| *(Ref #1) I notice some few positive Delta d13C values in Figure 8 panel D. Do you trust these values?* |

In the revised version we have addressed the remarks on error analysis in two ways:

1. **Δ-fluxes.** We included an uncertainty in the δ-flux estimates related to our spectral scaling approach, which was the main point in the referees' comments. The error is based on the uncertainty in the slope, and thus the scaling factor, between the net exchange flux and δ-flux. We added error bars to the revised figures 8 and 9 that show the propagated error in the δ-fluxes given a 68% confidence interval of the linear regression slope (see panels B and C of the figures below). Data points where the propagated slope errors were above the limits stated in the table below (also included in the newly added table 3) where rejected. Note that drift in the $CO_2$ instrument on long timescales occasionally caused outliers in our spectral correction without a large slope error. The corresponding outliers in the corrected δ-flux were filtered out with a bounds filter.

   All filters are summarized in the table below.

| | δ-flux species | Error limit (one-sided) | Bound-L | Bound-U |
|---|---|---|---|---|
| $H_2O$ | δ D | 0.07 ‰ m s-1 | N/A | N/A |
| | δ 18O | 0.07 / 8 ‰ m s-1 | N/A | N/A |
| $CO_2$ | δ 13C | 0.03 ‰ m s-1 | -0.02 ‰ m s-1 | 0.05 ‰ m s-1 |
| | δ 18O | 0.01 ‰ m s-1 | -0.02 ‰ m s-1 | 0.02‰ m s-1 |

   We realize that the uncorrected δ -fluxes also have a random error, but this error cannot be determined because of the loss of the high-frequency signal. Instead, we used the relative error in the net exchange of H2O and CO2, respectively, as a proxy for the relative random error in the δ-fluxes. This estimate of the random component of the flux errors is smaller than the errors derived from the slope uncertainty of the correction factor.

2. **Source compositions.** We computed the errors in the derived source compositions by propagating the errors in the δ-fluxes, and in the net exchange flux. For water, we use the rescaled δ-fluxes and for the CO2 the uncorrected δ-fluxes, as described above. For the net fluxes of both $CO_2$ and $H_2O$, relative errors were based on the random error estimates from EddyPro after Finkelstein & Sims, (2001). A major cause for the scatter in the isotopic source composition is related to the definition given in Equation 3. When the relative errors in the fluxes become large, especially during sign transitions where the net flux values cross zero, the uncertainty in the source compositions increases too. Like for the delta fluxes, data points where the propagated errors were above the limits stated in the table below were filtered out.

| | Source comp | Error limit (one-sided) | Bound-L | Bound-U |
|---|---|---|---|---|
| $H_2O$ | δ D | 9 ‰ | -120 ‰ | -40 ‰ |
| | δ 18O | 9 / 8 ‰ | -17 ‰ | -3 ‰ |
| $CO_2$ | δ 13C | 7 ‰ | -30 ‰ | 5 ‰ |
| | δ 18O | 4 ‰ | -20 ‰ | 10 ‰ |

   Note that nearly all data points filtered out occur during the morning or afternoon sign transitions. All remaining outliers in the source compositions, like the few positive d13C instances, also occur during sign transitions. We observed that on some occasions, especially when the isotopic composition of the source is near zero, the propagated error scales with the value of the source composition, preventing those instances from being filtered out (figure 8).

We made the following changes to the text:

First, we added a section on errors and data quality to the data processing section.
4.4 Quality control and error quantification

In our analysis we minimize systematic errors in the isotope flux dataset caused by outliers in the scalar data by applying a series of data filters. First, we applied a bounds filter to the isotopic data based on their nominal diurnal range during the measurement campaign. Second, instances were operators worked on

the instruments and/or opened the enclosures were flagged and removed, eliminating signal fluctuations due to temperature instabilities. Third, as part of the flux calculations, all detrended variables going into flux calculations, from both the EC-system and the isotope analysers, were filtered for outliers using an InterQuartile Range (IQR) filter with a bound of +/- 2.5 times the interquartile range. Last, flux intervals where less than 75% of the raw data was retained after these steps were excluded from the analysis.

In section 4.3.2 we investigated the systematic errors caused by lacking high frequency flux contributions, and in the previous paragraph we indicated how we prevented systematic errors caused by signal instabilities and outliers. On top of this, we quantified a random error based on the uncertainty in the slope, and thus the scaling factor, between the net exchange flux and δ-flux (see section 4.3.2). The 68% confidence interval of the slope fit was propagated to indicate the error in the spectrally corrected delta fluxes. Data points where the propagated slope errors were above the limits stated in table 3 where rejected. Note that drift in the $CO_2$ instrument on long timescales occasionally caused outliers in our spectral correction without a large slope error. The corresponding outliers in the corrected δ-flux were filtered out with a bounds filter (table 3).

The uncertainty in the source compositions was determined by propagating the errors in the δ-fluxes, and the errors in the net exchange flux. For $CO_2$, where we have more confidence in the uncorrected δ-fluxes compared to the rescaled δ-fluxes, we derived the error for the raw source compositions. In that case, we assume the relative error in the $CO_2$ flux to be a good approximation for the relative error in the uncorrected delta flux. For both the net $CO_2$ and $H_2O$ fluxes, relative errors were based on random error estimates from EddyPro after Finkelstein & Sims, (2001). A major cause for the scatter in the isotopic source compositions is related to its definition given in Equation 3. When the relative errors in the fluxes become large, especially during sign transitions where the net flux values cross zero, the uncertainty in the source compositions increases too. Like for the delta fluxes, data points where the propagated errors were above the limits stated in table 3 were filtered out.

We added a table indicating the error limits we accepted.

**Table 3.** Error limits and bounds applied to the δ-fluxes and isotopic source compositions. Sect. 4.4 describes how the errors were calculated. The source compositions for $CO_2$ are calculated relative to the atmospheric isotopic composition, similar to Figure 8 ($\Delta\delta^h X = \delta^h X_F - \delta^h X_{atm}$)

| | δ-flux/source species | Error limit (one-sided) | Lower bound | Upper bound |
|---|---|---|---|---|
| $H_2O$ | $F_\delta$-$\delta D$ | 0.07 ‰ m s-1 | N/A | N/A |
| | $F_\delta$-$\delta^{18}O$ | 0.07 / 8 ‰ m s-1 | N/A | N/A |
| | $\delta D_F$ | 9 ‰ | -120 ‰ | -40 ‰ |
| | $\delta^{18}O_F$ | 9 / 8 ‰ | -17 ‰ | -3 ‰ |
| $CO_2$ | $F_\delta$-$\delta^{13}C$ | 0.03 ‰ m s-1 | -0.02 ‰ m s-1 | 0.05 ‰ m s-1 |
| | $F_\delta$-$\delta^{18}O$ | 0.01 ‰ m s-1 | -0.02 ‰ m s-1 | 0.02 ‰ m s-1 |
| | $\Delta\delta^{13}C$ | 7 ‰ | -30 ‰ | 5 ‰ |
| | $\Delta\delta^{18}O$ | 4 ‰ | -20 ‰ | 10 ‰ |

Figures 8 and 9 were updated and now include error bars for the delta fluxes and source compositions.

[Figure]

*Figure E (Figure 8 in manuscript): 30-minute CO₂ exchange fluxes over the 6-day measurement period above an alfalfa crop field near Mollerussa, Spain. Panels B and C display the δ-flux derived using Eq. (2), including the uncertainty estimate propagated from a 68% confidence interval in fitting the spectral scaling method (Sect. 4.4). Panels D and E show the difference in isotopic composition between the vertical exchange flux and the atmosphere (δhXF - δhXatm), including the propagated 1σ uncertainty. The difference is plotted to eliminate the effect of instrument drift (see Sect. 4.1 on calibrations).*

[Figure]

*Figure F (Figure 9 in manuscript): 30-minute $H_2O$ exchange fluxes over the 6-day measurement period in Spain near Mollerussa. Panel A shows the evolution of Lv E measured by the LS-$H_2O$ and OPGA. Panels B and C display the δ-flux derived using Eq. (2), including the uncertainty estimate propagated from a 68% confidence interval in fitting the spectral scaling method (Sect. 4.4). Panels D and E indicate the isotopic composition of the vertical flux (δhXF ) and the atmospheric background (δhXatm, black line), including the propagated 1σ errors in the fluxes. In panels B through E, full circles represent the raw δ-fluxes or δhXF and yellow and green markings indicate the corrected ones. These corrections are based on OPGA scaling like is common in isotope flux research (yellow), or on the Spectral scaling technique we detail in Sect. 4.3.2 (green).*

We adjusted the text in the results section, incorporating information on the errors.

L. 315; which is a typical value for the isotopic composition of C3 plants (Kohn, 2010). When individual fluxes are close to zero, which happens during morning and afternoon sign transitions, unrealistic values occur in the derived source composition calculated according to Eq. 3. Most of these datapoints were filtered out due to the error limits set in Sect. 4.4. Still, some unlikely values persist, especially around zero for Dd13C, where error propagation results in small absolute errors. Also note that the errors in the δ-fluxes are somewhat smaller from July 28th onwards, due to a software change that reduced the temperature and consequent isotopic variability in the LS-CO₂.

**Measurement setup**

| |
|---|
| **(Ref #1)** *L. 133: Was the air flow rate of 30L/min measured or just estimated? Specify where the lines were separated. A diagram might help here.* |
| **(Ref #2)** *L133: The wording suggests that the authors are sure the pump flows add up arithmetically, ideally by having "closed the budget" with independent flow measurements of both components and the combination. If not sure, choose a wording that reverts to what you know for sure - e.g. how much flow the inlet scroll pump generates alone (with or without all the resistances of the complete system) and how much you measured in total for the combined system.* |

The air flow rate is the sum of the measured flow rate of the scroll pump flow through the main 9m tube, which slightly exceeded our MFM limit of 20L/min and 9.9 L/min measured flow rate of the laser spectrometers. We added them arithmetically after having calculated an insignificant pressure drop over

the inlet line of 3.5%. To better communicate the gas flow setup, we added a flow diagram to the schematic overview of figure 2 (see Figure G), including flow rates, tubing materials and tubing diameters. We adjusted the text to refer to the figure and clarify the 30L/min is based on a summation of the suction generated by all pumps in the set-up.

[Figure]

*Figure G: (Figure 2 in manuscript): Picture of the iso-flux measurement setup (taken by Wouter Mol, wbmol@wur.nl), overlayed with a schematic overview of the measurement setup. The instrument and enclosure outlined in green indicate the $CO_2$ isotopologue setup. The instrument and enclosure outlined in blue indicate the $H_2O$ isotopologue setup, which is behind the $CO_2$ enclosure in the picture. The flow diagram in the side panel indicates how the air was distributed over the analysers and the inlet pump, and lists the outer diameter, material type, flow rate, and length of each tube. Note that most of the length of the tubing feeding air to the analyzers was inside the enclosures.*

We have applied the following changes to the text:
"A total flow rate of approximately 30 l min-1 was generated by the suction of the laser spectrometers (9.9 l min-1) and an additional scroll pump with a flow rate of >20 l min−1. This high flow rate assures turbulent flow inlet conditions (Re > 3000). The side panel in Figure 2 illustrates the setup of the inlet system. The 0.9 l min−1 flow to the H2O laser spectrometer and 9 l min-1 to the CO2-LS were split off from the main inlet line in between the instrument enclosures"

> *(Ref #1) L. 125: Setup: Have you performed a dedicated lag time test while in operation to seehow long it takes from the inlet until you see a signal in the spectrometers that is purely due to the pump rate end tube length. A detailed setup scheme or diagram would also be of help to identify the reason for the comparably prominent memory effects. How have you tested your setup for leaks? At a pump rate of 30 l/min the risk for leaks is high and could also explain why EC mole fraction and spectrometer data did not always align.*

The schematic is now added to Figure 2 (see Figure G).
We did not measure the lag time directly in the way suggested and did not have the equipment for a thorough leak check in the field, however, we were aware of both potential issues, and put in dedicated efforts to avoid them:

- Two operators checked and double-checked all connections in the inlet tubing. The inlet system drew a vacuum when closed (no vacuum pressure measured), suggesting no major leaks were present. Because of the wide tubing used, the pressure drop along the tubing was small (3.5%). Thus, the pressure gradient (outside the tubing to inside the tubing) that would drive a flow through a small leak is very small. We conclude that potential small leaks should not contribute significantly to the total air volume sampled (sucking in even 1 l/min without a large pressure gradient requires a rather large orifice, which we would have detected with the pressure test). Also conceptually, we have not been able to describe a scenario where the lag times between the isotope and mole fraction signals could

be explained by a leak. Even for a mixed sample analysed in the spectrometers, mole fraction and isotopic contributions would still be related.

- Lag times were minimized by a combination of high inlet flow rates and appropriate tubing diameters. The flow velocity in the inlet tube was 10.7 m/s, which results in a lag time of 0.9 seconds of the laser spectrometers with respect to the EC system. The extra ~1m tubing (1/8") for LS-$H_2O$ and 1.5m tubing (¼") for LS-$CO_2$) from the split to the actual analyser inlet added 0.14 seconds to the LS-$H_2O$, and 0.10 seconds to the LS-$CO_2$ (Also see Figure G). Importantly, these lag times in the inlet system are constant and corrected for by time aligning the EC system and the laser spectrometers using the mole fraction signals measured by both (sect. 4.2). The drift we observed between the EC system and the laser spectrometers was small and constant during the measurement period, likely related to clock drift. Lag times between isotope signals and mole fractions (sect. 5.1) are not expected to be affected by the constant sample travel time. Note that during the sample travel time, flow in the tubing was turbulent which should minimize the mixing of samples in the tubing.

In the revised document we added the following remarks:
"Lag times between the EC-system and the isotope analysers caused by the inlet tubing were corrected during post processing (Sect. 4.2). Also, we confirmed that no major leaks were present in the inlet tubes by drawing a vacuum on the entire inlet system. To reduce the isotopic exchange between "

*(Ref #1) L. 130: A close-up picture of the actual inlet setup would be beneficial for understanding the location of the filter and the inlet compared to the EC system.*

[Figure]

*Figure H (Not included in manuscript): Eddy Covariance system and inlet of isotopologue analyzers, including the inlet filter.*

We added a schematic of the inlet setup to the overview figure of the analytical setup in the manuscript (see Figure G). The inlet filter is shown in figure H, but we believe explicitly mentioning this filter location is as informative to the reader as including Figure H.

We applied the following changes to the text:
The tubing and instruments downstream were kept free from dust and insects using a Whatmann cellulose thimble filter (Whatmann plc, Maidstone, UK) that was placed at the air inlet.

*(Ref #1) L. 132: specify inner diameter and material of inlet tubing*

The material of the main inlet is specified in L. 136. Additionally, the materials, diameters and length of all tubing, including those running from the instruments to the main inlet are now specified in the flow diagram added to figure 2 (See figure G and its caption).

We have also added the inner diameter of the main inlet line to the text.
L. 133, (3/8" OD, 5/16" ID tubing)

**Spectral corrections**

> **(Ref #1)** *L. 243: I encourage the authors to show the figure Appendix A3 in the main manuscript next to the Figure 6 since these figures visualize the methodology explained in this manuscript. I therefore think A3 is significantly more relevant in the main article compared to e.g. the time lag figure*

> **(Ref #1)** *Figure 6: Please add a line at the LPF frequency to visualize in which range you are fitting and please add the unscaled wH₂O co-spectrum for comparison*

We agree that figure A3 is relevant for the method and, as suggested, in the revised version it has been combined with figure 6 (see Figure I). We have also indicated the frequency range for H₂O. Adding an unscaled wH₂O cospectrum is impossible given the vertical range of the figures (+100x larger for wH₂O). Putting it on a secondary axis would result in exactly the black (or rescaled) line in the graph.

[Figure]

*Figure I (Figure 6 in manuscript): Cospectral flux comparison of δ13C with CO₂ (from EC) in panel A, and of δ18O (of H₂O) with H₂O (from EC) in panel B. All cospectra were calculated with respect to w'. The CO₂ and H₂O spectra were re-scaled to match the isotopic exchange spectra at timescales longer then 100 seconds. The spectra are based on a 30-minute data interval taken during the measurement period starting at 14:00 on July 25th 2021.*

Additionally, Figure A3 was removed and in text references to both Figure 6 and A3 were adjusted, including panel label references.

> **(Ref #1)** *L. 255: the phrase "instead of using the measured d18O covariance" is confusing here since you ARE using the measured d18O covariance for the fitting procedure. Reword please.*

To correct for this loss in the high frequency δ18O flux signal we use the H₂O covariance, re-scaled using the measured low frequency contributions from the δ18O flux covariance.

> **(Ref #1)** *L. 265: How was the fitting done? By minimizing what function?*

An example of a fit is shown in Figure J of this document.
We have clarified this in the revised text as follows:
L. 265: "The condition should thus be adjusted from ≡ to ̸=. Fitting equation 6 is done using a zero-intercept linear regression that relates the spectral densities of 18O and H₂O, where the slope coefficient equates to SF. Now, …"

*(Ref #1)* L. 258: Considering that the atmospheric state influences which are the dominating eddy sizes one could think that a varying LPF frequency is beneficial to increase the "fittable" part of the spectra whenever the atmospheric conditions are favourable for the system. From a very practical approach, how many datapoints were available for fitting with this LPF and an integration time of 30min?

*(Ref #2)* L259: Can this be shown, even if only in the appendix? (is SF really insensitive to LPF )

We agree that atmospheric conditions can affect the strength of the signal in the LPF region. Still, we feel that the artifact causing signal lag, and not the atmospheric conditions, should dictate the position of the LPF. When refining the approach in the future, a variable LPF could be added, but it should be dependent on the artifact strength. For now, we choose to present the concept in the simplest possible way and thus with a constant LPF.

In the process of responding to the referee comments, we re-examined our spectral fitting approach, in particular the number of datapoints included in the fitting procedure. We discovered that the number of low frequency datapoints in the $H_2O$ spectra was increased because of an imposed 0-padding by the FFT function we used. This can be seen in the old figure 6, where 2.5e-4 Hz (4000s) waves are present in a 30 minute cospectrum (1800s). We have changed the FFT calculation settings to generate raw cospectra, where limited smoothing is used to make the high frequency spectral range visible in the figures. This visually affects the spectra shown in the manuscript, which get coarser at low frequencies. Also, it causes nuanced differences in the derived spectral scaling and correction factors, as can be seen in Figures E & F (figures 8 and 9). Nevertheless, the patterns which could be seen in the figures are not affected, and all the described results and conclusions are identical.

In the final spectral analysis, 19 Datapoints are available for the fit using the 0.012 Hz LPS. Fewer might be used dependent on sign differences between the respective spectral densities (eq. 6).

We investigated alternative low pass filters and they retain 15 or 25 datapoints instead of 19. As shown in figure J below, the slope of the fit is not significantly different when one or several datapoints are excluded from the fit. We reworded the sentence referred to by referee #2 to be in line with this observation as we agree that 'insensitive' suggested too much certainty.

[Figure]

*Figure J (Not included in manuscript): Example fit of the cospectral densities of δ18O and $H_2O$, both with respect to w', at timescales longer then 0.012 Hz.*

We have applied the following changes to the text:
We chose a LPF of 0.012 Hz based on visual inspection of various co spectra, leaving 19 datapoints for making a fit. We note that that the addition or subtraction of some datapoints near the filter boundary doesn't change the fit much, suggesting limited sensitivity to the exact value of the LPF.

> *(Ref #1) L. 236 ff: I encourage the authors to separately outline the different spectral correction methods used previously in the literature that they cite here and how their approach is different.*
>
> *(Ref #1) L. 387: I encourage the authors to expand on the discussion between the two different correction methods, present detailed analysis and discuss causes for the differences.*

The theory section refers to the current state-of-the-art correction approach used by various authors (i.e. Oikawa et al., 2017; Wahl et al., 2021; Wehr et al., 2013) in a brief but in our view complete way. We clarify that they all use a loss in $H_2O$ signal to correct for minor isotopologue & delta fluxes and explain how that correction factor can be calculated in equation 5. The key point is that they apply the same correction to all isotopologues, and that delta fluxes are increased by the same loss factor as found for the net flux. Directly after this description we explain how our spectral scaling method works, which extra problems it solves and how it therefore generates larger correction factors. We believe that this is sufficient to contrast the method we present with the approaches applied previously in the literature. While we agree that an extensive CF analysis could be valuable, this is not within the scope of this first case study.

> *(Ref #1) L. 273: How did the CF between d18O and dD for $H_2O$ compare? Were there strong differences? Also, judging from Fig 8, panel 3, the CF for d18O in $CO_2$ is often >1.*
> *Show analysis of CF for both $CO_2$ and $H_2O$ systems*
>
> *(Ref #1) Also L. 345-348: Which formula/method was used to calculate OPGA scaling? Could you compare the performance of the different methods and analyze differences and explain them with meteorological conditions – the bias seems to be quite significant à maybe do this in discussion*
>
> *(Ref #1) Fig 8 - There is significantly more noise visible on the signal in panel 3 compared to 2, and scaling factor and thus CF seem to be >1.*

Generally, CF's were approximately 1.1x larger for dD than for d18O during midday, with considerable variance and variability over the day.

Equation 5 in section 4.3.2 shows how the OPGA scaling was derived. A detailed analysis on the differences between the spectral scaling method and the OPGA method under varying meteorological conditions could be valuable but was outside the scope of this study. Our main message is that previously used scaling methods apply the same spectral loss to all isotopologue fluxes, whereas our first order correction using spectral scaling results in strongly different correction factors for the net fluxes and delta-fluxes.

Yes, panel C has more noise in the re-scaled fluxes compared to panel B. An important cause of this is that delta fluxes are near zero, as we mention in the text. In addition, the instability (low frequency drift) of the d18O measurements with the LS-$CO_2$ affects the low frequency contribution of the 18O flux signal, which is used for scaling. The combination of small 'real' flux signals, and known instrument instability are the major reasons for the increased scatter in panel C. The revised figure 8, reproduced in the qu*ality control and error estimation* section of this rebuttal suggests that CF-d18O is regularly larger then 1, which may indicate that some spectral correction is required for this species. Alternatively, the significant low-frequency instrument instabilities can generate an artificial correlation with w' at low frequency, where the cospectra are less likely to average to zero. This would also cause CF's to be larger than 1. This second option, which would be a result of limiting instrument performance, prevents us from confidently stating that the correction method works well for d18O-$CO_2$.

We have applied the following changes to the text:
While the OPGA correction is significant, panels B and C reveal that the magnitude of the spectral scaling correction on the δ-fluxes is even larger (Sect. 6) which is in line with the loss of high frequency contributions to the δ-flux shown in Fig. 6. The CFs for δD were about 10% larger than the CFs of δ18O, which matches with the larger lag times for δD shown in Sect. 5.1.

Additionally, we have implemented the following changes to the text to explicitly mention the influence of instrument instability:

18O δ-fluxes vary around zero, indicating small and variable atmospheric gradients caused by small differences between atmospheric and source compositions. Instrument instability also affects $F\delta$-$\delta18O$ and given the small 'real' flux signal, the instability likely dominates the noisy spectrally scaled $F\delta$-$\delta18O$ signal. However, CF's are regularly larger then 1, which is likely a low frequency drift artifact but could also be an indication of some missed high frequency flux signal. In line with a small δ-flux, our daytime measurements show no notable difference between the isotopic composition of the vertical flux and the atmosphere"

> (Ref #1) L. 302: So you did spectrally correct the CO₂ isotope fluxes? This sentence is in disagreement with L. 309 and L. 337? Please clarify.

> (Ref #1) panel 2 and 3: since the re-scaled data points are your final data product, consider swapping the markers for raw and re-scaled, since the light gray crosses are very subtle

We do show the spectrally corrected CO₂ isotope fluxes in figure 8, However, the CO₂ laser spectrometer suffered from low frequency drift in the isotopic signals which causes uncertainties in the fluxes. As these uncertainties get inflated using the spectral correction method, we use the raw fluxes. This is in line with the current descriptions in lines 309 and 337.

To prevent confusion, we have added the following clarification to the text:
L. 307: the raw δ13C and δ18O δ-fluxes are comparable to the spectrally corrected variants. This indicates and supports that we can resolve the δ-fluxes for CO₂ without corrections (Sect. 4.3.2). The instability of the CO₂ analyser on longer timescales (Sect. 4.1) causes increased uncertainty in the spectrally scaled δ-flux signals compared to the raw δ-flux signals. Therefore, we show the spectrally corrected δ-fluxes in Fig. 8 in grey for completeness but proceed in the further analysis with the raw δ-flux signals.

**Best Practices**

> (Ref #1) L. 458: I would argue that a time lag correlation is no new concept, please remove Conclusions section: The whole section is missing impact and the highlighting of why this method is superior to other methods. Further, an uncertainty statement and a best practice recommendation should be given for any reader that would like to implement a similar technique in future field studies.

> (Ref #1) L. 360: The paper would benefit from a clear best-practice recommendation from the authors concerning the system set-up for future field campaigns to allow for more widespread implementation.

We added a *best practice* section with recommendations for setting up isotope flux measurement systems based on the findings communicated in this manuscript. In addition, some adjustments were made to the conclusions highlighting the impact of the lag times we found and correction method we developed.

7. Best practices

Based on our experience measuring high frequency iso-fluxes using the eddy covariance technique, we provide best practice recommendations for setting up isotope flux experiments

- Laser instruments should be installed in weatherproof environments, which are temperature stabilized to increase instrument stability. In harsh environmental conditions, climatization for the inlet and laser spectrometer pumps needs to be considered. Finally, the combined power consumption of the analysers, enclosures and pumps exceeds 4 kW for our setup, requiring reliable power supplies in remote places.
- Mole fraction calibrations are crucial, and more important than thorough span calibrations for isotope flux measurements. Mole fraction dependencies need to be corrected for to prevent artificial iso-flux contributions. The range for which the mole fraction dependencies are performed should match range in of mole fractions in ambient air.
- Isotopic compositions should be measured at sub second time scales and the sample flow rate must be sufficiently fast to keep the residence time of air in the sample cell short enough such that the air in the cell is replaced in between measurements.

- To maximize the high frequency response of iso-flux measurements and enable effective time lag corrections, we recommend placing the air inlet in proximity (~20 cm) of the anemometer centre, minimizing the length of the inlet line, and minimizing air sample mixing in the inlet line by enforcing short residence times and turbulent flow conditions.
- Liquid water formation through condensation on hydroscopic materials leads to isotopic exchange which severely affects the results. We recommend using inlet tubing with a smooth inner surface and active heating to minimize potential condensation and exchange effects. In addition, hydrophilic materials in the inlet system, such as our hydrophilic inlet filter, should not be used.

**8. Conclusions**

We have presented a methodological approach for measuring iso-fluxes during the LIAISE 2021 field campaign. The measurements encompassed six days and were supported by comprehensive auxiliary data which will support future modelling studies of biosphere-atmosphere exchange, integrating the associated isotope effects. Our analysis has introduced two key new concepts in terms of data processing. First, we have identified and quantified time lags between mole fractions and δ-values, have shown how to correct for those, and have interpreted these lag times as a marker for isotopic inlet line attenuation and water condensation artefacts. We have documented large time shifts between the $H_2O$ isotopologues, which has not been reported previously. We have demonstrated that when a time shift is detected, δ-flux spectra most probably lack high frequency contributions compared to net exchange spectra. This cannot be taken into account by previously used data processing approaches, which posit that signal loss is identical for net fluxes and δ-fluxes. The second new concept is that of spectral scaling, which allows the asymmetric signal loss to be corrected for. The attenuation of the isotopic signal in the inlet line was likely caused by a small liquid water reservoir on the inlet filter. Finally, we have illustrated how this new spectral scaling technique would impact flux partitioning.

**Uncategorised comments**
* * *
***(Ref #1)*** *L. 37&38: Please describe what "framework" you are referring to and it is not clear at this point which parameters are needed for measuring the "turbulent iso-flux" (what does "iso-flux" mean here)*

The scope of this technical manuscript is the description our iso-flux measurement setup, as well as unexpected artefacts that we encountered during the deployment, and the data analysis procedures that we developed to correct for these artefacts. This manuscript provides a thorough technical description of our setup for future implementation of isoflux measurements, with the goal to partition the gross surface fluxes into their individual components. The described procedures should be useful for other scientists setting up similar measurements. In the revised version we have more clearly indicated the scope of the present manuscript. Also, we have added some more detail on the 'framework' and the 'iso-flux'.
* * *
***(Ref #1)*** *L. 80: In section 2.2 or in the Introduction you should add more literature on the different methods used for measuring iso-fluxes and discuss their limitations and give reasons why you chose your set-up. Literature missing: Yakir & Wang 1996, Keeling 1958, Welp 2008, Wei 2015, and specifically Good et al. 2012 and 2014.*

Thanks for the suggested articles.
Currently, the introduction describes how measurements of isofluxes have developed historically, including gradient based measurements. The EC approach provides the only direct measurement of isotope fluxes. While the papers of Yakir &Wang, Keeling, Welp and Wei are relevant, they do not target iso-flux measurements, but rather source signature measurements using Keeling plots or a gradient method to partition fluxes. These studies are mentioned in the introduction section of the revised version as alternative approaches.
* * *
> Good et at. (2012) do apply an EC isoflux technique in their analysis, deriving both isofluxes and source signatures. They found relatively larger uncertainties for the source compositions derived with EC compared to other methods. We note that they applied the traditional method of correcting iso-flux spectra, which we improved upon in our manuscript

We have implemented the following changes to the text:

L. 37; Doing so allows us to better understand, and consequently model, the drivers of each flux component, including non-linear short term (diurnal, sub-diurnal) effects (Vilà-Guerau de Arellano et al., 2023). Keeling (1958) introduced the well-known "Keeling plot" budgeting approach where atmospheric isotopic composition measurements can be used to infer bulk source isotopic compositions. The technique can be used for flux partitioning as well, when combined with analysis and/or modelling of specific source isotopic compositions (Good et al., 2012; Yakir & Wang, 1996). The limitations of Keeling's approach are a lack of insight into the turbulent processes underlying the exchange and an ill-defined footprint.
More recent approaches use micrometeorological measurements to derive the isotopic composition of the exchanging gas flux (iso-flux) and attempt to use it for partitioning. In that case, a mathematical framework relating ecosystem gas exchange fluxes to isotopic compositions needs to be used. Oikawa et al. (2017) present such a framework and clarify that accurate iso-flux measurements are key to reliably partition fluxes.

L. 39; Two decades ago, isotopic compositions measured with laser spectrometers became precise enough to allow for gradient based iso-flux methods (Griffis et al., 2004, 2007, Welp et al. 2008, Wei et al. 2015).

L. 40; Some years later, it was shown that high sample throughput and precision could be achieved to perform direct flux measurements by combining stable isotope measurements with Eddy Covariance (EC) (Good et al., 2012; Griffis, 2013; Sturm et al., 2012).

L. 45; Still, there is much to be learned about δ-fluxes, the challenges in measuring them and how to correct sub-optimal data (Oikawa et al., 2017). Motivated by the goal of deriving flux partitioning using a method that incorporates turbulence and has a field scale footprint, we focus on implementing measurements and corrections of turbulent iso-fluxes derived using EC. In this manuscript, we describe"

> *(Ref #2) L33-34: The short mentioning of global NEE partitioning together with the terms "the same principal" (princip\*le\*?) raises the expectation that the same methodology can be used for global and local isotope-based partitioning. However, on a local scale we are talking about partitioning measured fluxes (eddy covariance) with isofluxes, while on a global scale partitioning has to be inferred from concentrations, isotope concentrations and additional assumptions. Depending on length issues and how important it is, I suggest to either change the wording such that no too tight identity between the two methods is expected, or alternatively explain the global method in a bit more detail, and then clearly mark where you start focusing on the local one relevant for your study.*

Thanks for mentioning this. We rephrased the section to reflect this distinc tion more accurately.

We included the following changes in the text:
On the global scale, measurements of the δ18O isotopic compositions have been used to separate annual NEE into GPP and Reco (Prentice et al., 2001). On smaller spatiotemporal scales (hourly, local), a different isotopic budget approach, based on eddy covariance can be used to split NEE into GPP and Reco, and Evapotranspiration (ET) into Evaporation (E) and Transpiration (T) (Lee et al., 2009; Vilà-Guerau de Arellano et al., 2020).

> *(Ref #1) L 323-327: Have you analyzed a sample of the irrigation water for its isotopic composition?*

We have not analysed a sample of irrigation water for its isotopic composition. We agree that such measurements, in combination with precipitation isotopic compositions could have helped to understand the source water isotopic composition better.

> **(Ref #1)** L 236: How do the results from the $H_2O$ isotopes support this hypothesis? If the difference in Irrigation vs Precipitation isotopic composition is indeed the reason, you would expect to see a prominent difference in the ET isotopic composition after the Precipitation event compared to other days.

We believe this comment refers to line 326.

Yes, the d18O-$H_2O$ isotopic signals should be impacted if d18O-$CO_2$ signals are changed due to oxygen isotope exchange between $CO_2$ and $H_2O$ within plants. In Fig. 9 we can see that in the hours after the rain event, $H_2O$ iso-fluxes are only slightly reduced. Also, the source isotopic compositions are slightly more depleted, i.e. more similar to the atmospheric compositions compared to other days. As precipitation water is expected to be more enriched compared to irrigation water, the more depleted isotopic composition in the leaves would be due to increased relative humidity preventing leaf water enrichment rather than a changed ground water source.

Importantly, the alfalfa crop is known for having deep roots making it likely that the evaporated water is an "older" mixture of irrigation and precipitation water, preventing a quick response to precipitation water (also note that an irrigation event preceded the rain by 2 days). We suggest that the $H_2O$ and $CO_2$ isotopic signals can both be explained by a relatively depleted water source in the deep soil, still resulting in upward delta fluxes for $H_2O$ (be it less strong than then it would have been with only precipitation as source water), but causing equilibrated $CO_2$ to not be enriched enough to generate positive $CO_2$-18O delta fluxes, as was observed in previous studies (e.g. Clog et al., 2015).

We have implemented the following changes to the text:

L. 352; "The precipitation water - which is more enriched compared to the irrigation water – does not have a notable impact on the delta flux or source composition on short timescales. This suggests that the alfalfa, which has deep roots, takes up water from deeper soil layers where the water sources are mixed or buffered. In line with the positive δ-fluxes, the isotopic composition of the vertical flux (δhXF) starts at near atmospheric values in the early morning,"

> **(Ref #1)** L. 370-373: I would suggest to either show the results of these phase spectra or remove the sentences relating to it.
>
> **(Ref #2)** L373: replace "made" by e.g. "analyzed", "showed", "plotted" or "considered" (the spectra are not made by the scientist but by nature and/or the instrument). Is this something only mentioned here or actually shown in the results, as the next sentence suggests? It contributes to the somnewhat "anecdotal" character of the manuscript.

Based on the referee suggestion we deleted the sentences relating to the phase spectra from the text.

> **(Ref #1)** L 405: In other words, you assume that turbulence is not fractionating.

We do not think that the message can be simplified to the statement that turbulence is not fractionating. We suggest that all individual sources within the eddy footprint are represented over all eddy scales and that no mismatch should arise between the cospectral shapes of a net exchange flux and a related delta flux. This observation is essential for our spectral correction method and, we believe, cannot be reduced to the statement that turbulence does not fractionate.

> **(Ref #1)** *L. 417-425: This is an interesting thought but I would assume the performance of this method is highly dependent on atmospheric stability and the nature of the turbulent transport (i.e. better suitable for unstable atmospheric stratification). Please elaborate on the limitations of the presented method concerning different atmospheric and general field site conditions. If the atmospheric stability requires shorter flux integration periods (very stable atmospheres might require integration times of a few minutes only) and using the constant LPF, there could be cases where essentially no data points are available for the fitting procedure of your presented method. Additionally: Could you somehow manipulate an existing timeseries to represent a timelag and smoothing impacted timeseries and therefore show the feasibility of your method?, i.e. no signal loss even if you only use frequencies up to the LPF. Section 6.1: I find the presentation on the quadrant analyses very interesting and would encourage an additional figure (maybe included in Figure 10) with a complementary time window (e.g. night time) to see the differences in the iso flux behavior and the added information of such an analysis technique.*

Stable atmospheric conditions
We agree that there is a limit on how slow the low frequency isotopic composition measurements could be, and that uncertainties in the fit increase during stable conditions when the largest eddy scales are not present.

We have implemented the following changes:
L. 421; "and setting up the analyser further away from the EC station. Note that in stable atmospheric conditions, the larger eddy scales do not contribute to the flux. Still, ample flux signal is needed in the limited fitting window to make a reliable correction, and this poses a limit on how slow the low frequency isotopic composition measurements can be. Even when low frequency flux contributions are present, there will be uncertainty in the determined $\delta$-flux due to the error in fitting the spectra (Sec. 4.4)."

Synthetic dataset
We agree that testing the method on a synthetic dataset would be an appropriate next step to show that the method works as expected. In our study, we partially tested this already using the LS-CO$_2$ data. Fig A3 (6A in new version) shows how well a rescaled CO$_2$ spectrum matches the measured 13C spectrum. Figure 8 shows the close correspondence between the rescaled and measured iso-fluxes, suggesting no substantial loss in information eventhough the LS-CO$_2$ instrument was unstable at long time scales, affecting the fits.

Quadrant analysis
Thank you for your interest in the quadrant analysis. We present this in the outlook section and will investigate the high frequency turbulent variations in much more detail in a future publication, including a more in depth look at isotopic quadrant analyses. Therefore, we decided not to add more examples in the present manuscript.

> **(Ref #2)** *L86-90: This part is obscure and at least near the end also not completely correct. Eq. (2) says nothing about gradients, it relates a covariance to a flux. A relation between any of both with the vertical gradient is not shown in the Eq. (and not in the Fig. alone either); it is subject to a constant micrometeorological variation of the proportionality "constant", and occasionally even counter-gradient fluxes do exist where the scale of transport-relevant eddies is larger than the vertical distance between local profile extrema (e.g. in or near canopies or with strong entrainment). Re-check which role (if any), knowledge of the gradients actually has in your methodology and where (my guess would be mostly if not everywhere) it is more appropriate to speak about fluxes or covariances. Even then gradients might help to "teach" your rationale in an idealized way (such as in Figure 1), but the reader should be able to easily separate these from cases where gradients are actually relevant to the derivation of your methodology (if any). Concerning δ-fluxes vs. isofluxes, try to find a more systematic way to relate both to each other and to the ultimately wanted partitioned fluxes. If the δ-flux is the key part of the iso-flux as mentioned here, and the latter is needed for the partitioned flux, then which Equation term including Fdelta is the iso-flux? Or alternatively, if your chosen methodology circumvents any need to explicitly address the iso-flux, write this more clearly.*

Gradients to visualize idealized exchange
It seems that in the previous manuscript we did not state clearly enough that we use the gradient figures primarily to conceptually visualize how the isotopic fractionation effects at the land surface affect the atmosphere and subsequently cause delta-fluxes of a given sign in idealized situations. This is better described in Sec 2.1 of the revised version. We found these figures very helpful in our own internal discussions and think they can be a useful tool to illustrate the effects. We agree that exceptions like counter gradient fluxes occur, but this does not prevent the use of these figures for illustrating the general ideas. We have adjusted the text to clarify that the method or measurements of the iso-fluxes do not use gradients, but that gradients will ultimately drive the iso-fluxes.

We included the following changes to the text:
Section title 2.1; "Isotope fractionation effects associated…"

L. 75; These idealized gradients are displayed in Figure 1 to conceptually visualize the processes. Note that no gradient-based methods were used.

Fig 1 caption; "view of the idealized gradients…"

In the revised version we have deleted the second reference to the gradients in in section 2.2, where delta fluxes are described, because this is indeed physically questionably.

Unclear iso-flux – δ-flux relationship
The scope of this paper is limited to measurements and corrections of δ-fluxes, which we have made more explicit in the introduction of the revised manuscript. We have deleted the link with iso-fluxes to keep the text focussed and avoid the introduction of equations that are not used in the analysis.

> **(Ref #2)** L115: As a first approximation literature typically assumes that ET it is always is limited by at least one of both, energy (in your case radiation) or water supply. The fact that both, radiation and soil water, were ample, does not automatically imply none of them was limiting. This is only the case if supplying even more could not increase ET more. Especially for radiation this does not necessarily need to be true in your case. It is acknowledged that third factors may limit ET instead, making it insensitive to changes in both water and radiation supply - such as e.g. turbulent removal or evaporated vapour, similar as discussed in Lobos-Roco et al. 2021 (Atm. Chem. Phys., https://doi.org/10.5194/acp-21-9125-2021). However, if you have such a non self-evident hypothesis it should be clarified in the manuscript, and whether it is supported by measurements or models for your site or merely an assumption. In most settings, more radiation tends to increase convection, such that ultimately it remains the limnitiung factor.

Thanks for the comment. We intended to communicate that the vegetation was not water stressed (according to leaf level stomala conductance measurements), and that incoming radiation during the campaign was as large as it gets during the year. We adjusted the text to specify more directly the environmental conditions, including radiation fluxes, together with the observation that the vegetation was not water stressed.

L. 115; "…"

L. 121; "largely clear sky conditions and some cirrus, 32∘C mean peak temperatures, and 650 Wm−2 mean peak net radiation. Despite the high temperature and radiation levels, the alfalfa was not water stressed according to leaf-level measurements of stomatal conductance".

> **(Ref #2)** L141: "in principle" here means something like "in absence of the physical response issues we partly deal with in this study"? Note that it is good practice to still subject even open-path systems with 20 s-1 logging frequency to a correction for spectral loss - see also comment on L361. This may be negligible in the face of much larger loses of the isotope analyzers, and thus not necessary to apply in your study, partly also since the open-path system is an IRGASON with virtually no measurement path offset between w' and the concentrations (which however still suffer from some loss due to path length averaging). However, it should be avoided to accidently leave the impression that the mentioned logging frequencies alone should be sufficient, with the best available measurement system, to do without any spectral loss correction.

We have implemented the following changes in the revised text:
"… which allows for eddies of the smaller turbulent scales (cm) to be distinguished, and therefore most of the turbulent energy spectrum to be resolved (Moene and Van Dam, 2014)."
The comment on spectrally correcting the OPGA was dealt with in the EC section above.

> **(Ref #2)** L208: Much the same method is sometimes used to get rid of physical delays in EC data causing part of the spectral response issue, e.g. between sonic anemometer and gas analyzer (wind drift from one to the other, tube delays, other physical response time issues differeing between two instruments). What makes you think here that the result is "the most probably offset in the clocks"? If by design it includes the abovementioned effects, clarify whether this is a wanted (hopefully yes) or unwanted effect in this case.

Our aim was indeed to both synchronize clocks and get rid of inlet delays and wind effects as we describe in line 200. We agree that the wording "offset in the clocks" suggests that only data logger time delays were corrected for, while we intended to say that such a time shift would best time align the two signals irrespective of the cause.

We have implemented the following changes:
The maximum value of r indicates the time shift of optimal correlation and thus the most probable offset between the measured time series.

> **(Ref #2)** L285, Fig. 8, Fig. 9 and others: The terms for the different instruments (e.g. OPGA, Aerodyne, Picarro, laser spectrometer) vary across the manuscript. Try to minimize variability to what is needed to stress the relevant physical properties of the instrument in the respective section / figure and, more important, to contrast natural pairs of terms (e.g. not the company name of one instrument with the measurement principle of the other instrument as in Fig. 8 and 9).

Thanks for the comment. In the revised version we have minimized use of instrument manufacturer names and refer to $CO_2$ and $H_2O$ laser spectrometers (or LS-$H_2O$, LS-$CO_2$), which contrasts with the OPGA term we use. See for example the updated Figure K.

We have applied the following changes to the text:
- Figures 5,6, 8, and 9 were updated (example in Figure k below).
- We updated the text and figure in Appendix A (Now A3) to no longer mention the manufacturer
- We changed various occurrences in the text to refer to LS-$H_2O$ and LS-$CO_2$.

[Figure]

*Figure K (Figure 5 in manuscript): Comparison of the cospectra of w' and q' from both the closed path laser spectrometers and the OPGA of the Eddy Covariance station. Panel A represents the $CO_2$-w cospectra, and panel B the $H_2O$-w cospectra. The spectra are based on a 30-minute data interval taken during the measurement period starting at 12:00 on July 25th 2021.*
* * *
**(Ref #2)** *L323-325: "pose" is strange wording in this context - maybe "assume"? Which part of the reasoning is supported by the Gat and Yakir references how? In neither of them could find reference to the effect of altitude on drop formation, maybe because Gat is too long to be cited without a page reference. The bibliographic information on Gat at L496 is probably insufficient, check journal guidelines. Did you consider that the higher altitude at which droplets hit the surface in mountains does not necessarily imply a much higehr altitude of their formation?*

We have replaced "pose" by "assume".
We referred to Gat et al. (2001)as a source for understanding the claimed altitude dependence of the isotopic composition of precipitation. In that document, pages 53-55 are about the isotopic altitude effect. Also, the vertical isotopic gradients within cloud described on page 28 relate to the same phenomenon. We now indicate the relevant page numbers in the text.
Yakir et al. (1994) describes the relation between $H_2O$ source water and atmospheric $CO_2$ isotopic compositions after oxygen 18 exchange. We repositioned the references so they both relate only to the section directly preceding the reference.
We also adjusted the reference to include all bibliographic information.

We made the following changes to the text:
L. 324; "We assume that the water fed to the vegetation"

L. 326; This water will be more depleted compared to rainwater because of the increased altitude at which the precipitation formed (Gat et al., 2001, p. 53-55). When this isotopically depleted water is used by the plants and exchanges isotopes with $CO_2$ in the mesophyll, this will lead to relatively 18O-$CO_2$ depleted δ-fluxes during daytime (Yakir et al., 1994).

Updated reference; Gat, J. R., Mook, W. G., and Meijer, H. A. J.: Environmental isotopes in the hydrological cycle, volume II, Atmospheric water, Tech. rep., International Hydrological Programme (IHP-V), Paris, https://www.hydrology.nl/images/docs/ihp/Mook_II.pdf, 2001.
* * *
**(Ref #2)** *L357: The opening statement of the discussion is a bit bold given that the paper does not seem to try to actually show partitioned fluxes (e.g., transpiration and evaporation).*

We partially agree with this comment. We do believe that the isotope fluxes give insight into the processes underlying the net exchange fluxes, even when no partitioned fluxes are shown. We adjusted the text to

include this nuance. Note that we also better clarified our goal of investigation isotopic exchange fluxes and not partitioning in the introduction (responds to Ref. #1, L 37&38 comment).

We have shown in our analysis that isotope flux measurements can add process information to state-of-the-art $H_2O$ or $CO_2$ net ecosystem exchanges flux measurements and atmospheric isotopic composition measurements, that can be used in future studies to partition fluxes.

> **(Ref #2)** L393: They do not affect the eddies, but the way in which they are seen in the data. Replacing "affect" by "concern" or "apply to" might also work.

Agreed, changed in text.

> **(Ref #2)** L437: "The tool...": whole sentence strange, both language and (unclear) meaning

Thanks for pointing this out. As this is already partially mentioned in the preceding sentence, we decided to extend that remark and delete the unclear sentence.

Here, the co-varying perturbations constituting the fluxes are plotted. This allows for dynamics and background patterns of the exchange to be investigated, such as the contributions of specific types of eddies. .

> **(Ref #2)** L426f: Try to make the outlook a bit less a collection of "claims" what your own work group intends in the future, and a bit more an open discussion of what the community including yourself can do as logical next steps based on your findings in this study (or their synthesis with those of others). Figure 10 and the discussion around it are interesting, however somewhat poorly embedded into what is already done (e.g. in the $CO_2$ vs. $H_2O$ quadrant space). Though not sure, I wonder whether the outlook is actually the best (or even a proper) place to present such data new to the reader - what about either an own results or appendix subsection if there is enough material, or removing / keeping to a minimum for future studies?

Scientific outlook
Motivated by this comment, we added a paragraph to the start of the outlook section to suggest future research topics based on the results presented. This includes some good suggestions made by the referees regarding virtual data and error analysis.

The isotope-dependent, loss of high frequency information in the co-spectra reported here is an essential feature that needs to be addressed in future, isotope-based flux partitioning. Ideally, a field experiment designed to determine the full isotopic δ-flux cospectra needs to be performed that applies the best practices we outline in Sect. 7. A high frequency data set that does not suffer from the artifacts we encountered could then be used to corroborate our proposed spectral correction method by down-sampling to a lower frequency data set. In addition, a lower measurement frequency limit can be established which ensures that adequate low frequency δ-flux contributions are available to make a fit, under a variety of meteorological conditions.
We also recommend the exploration of our spectral correction method for isotope analysers that are not necessarily designed for fast turbulence measurements, for instance the laser spectrometers with high flow rates used to measure vertical δ-gradients (Griffis et al., 2007; Wei et al., 2015). By adding sonic anemometers close to the inlets, spectral corrected δ-fluxes can be derived using the EC method and compared with the δ-fluxes derived from the vertical profiles. Good agreement would give confidence in the spectral scaling method. Further investigation into the lag times between delta-values and mole fractions would be necessary if the isotopic memory effects presented here cannot be excluded by avoiding the use of hydroscopic materials.

Besides flux partitioning at 30-minute intervals, there are other open questions in land-atmosphere exchange that require high frequency iso-flux measurements…"

Quadrant analysis

In deciding on the structure of the current manuscript we have considered both alternatives for the quadrant analysis that you suggest; leaving them out entirely or integrating them into the results. Still, we believe the outlook, where we mention future research opportunities using higher frequency isotope data, is most fitting.

**Comments on figures:**

> *(Ref #1) Fig. 5: Consider introducing subplot labels instead of "left" and "right".*

We introduced subplot labels A and B including references to the new labels. See Figure K (Figure 5 in manuscript).

> *(Ref #1) Fig. 8: Panel labels are missing,*

We incorporated this comment. Please see Figures E and F (8 and 9 in manuscript respectively).

> *(Ref #1) -Top panel: Could you show agreement between Aerodyne and OPGA as scatter plot and give correlation coefficient? What software did you use to calculate the net fluxes shown in this panel?*

Here we share a scatter plot between the fluxes derived with the laser spectrometer and the open path gas analyser (Figure L). The slope is 0.91 suggesting that fluxes were relatively well resolved.  $R^2$ is 0.98, indicating that variability in the fluxes was well captured. We would argue that the same conclusions can be drawn from Figure F (Fig 9 in manuscript), panel A, where the diurnal cycle of the fluxes is shown, and the temporal evolution can be directly compared with delta fluxes and source compositions. Instead of adding a scatter plot with identical data, we chose to add these coefficients to the text.

[Figure]

*Figure L (Not included in manuscript): Fit between the $CO_2$ flux measurements from the OPGA and the LS-$CO_2$ for the entire measurement campaign.*

Software and net flux calculations are described in the "Eddy Covariance" section of the rebuttal, and are now also included in section 3.2 of the manuscript.

We applied the following changes to the text:

The total $CO_2$ flux signal derived from the closed cell $CO_2$ laser spectrometer matches the OPGA well. A linear regression between both results in a slope of 0.91, and an R2 of 0.98.

> **(Ref #1)** *Fig 9: Top panel: a scatter plot between OPGA and Picarro LvE would be nice to see.*

We show the requested scatter plot in figure M. As we did for Figure L, we have only included the fit coefficients in the text, as the same data is already presented in panel A of Figure 9.

[Figure]

*Figure M (Not included in manuscript): Fit between the $H_2O$ flux measurements from the OPGA and the LS-$H_2O$ for the entire measurement campaign.*

We incorporated the following changes in the text:
L. 344; The LvE signal also reveals large differences between the gas exchange measured by the closed path instrument and the OPGA. A linear regression between both variables results in a slope of 0.59 and an R2 of 0.94.

> **(Ref #1)** *Top panel: Do you have an explanation for the "W" shape of the OPGA signal when looking at one day from noon-noon?*

This is an interesting observation indeed, but no, we do not have an explanation for the w-shape. A negative, dew related moisture flux would be more logical. We re-checked to data processing to confirm that no absolute value of the flux was used accidentally, which could make a dew event seem like an evaporation event. We think it is most likely that the evaporation fluxes are related to the complex local circulations related to irrigation related temperature gradients, sea breeze, and katabatic winds. One hypothesis could be that the colder sea-breeze, which started in the late afternoon and ends in the night, can cause unstable conditions over the warmer surface, allowing for evaporation flux to take place at night.

> *Figure A1: instead of showing the fitting lines color coded it would be better to show the fitting parameters as timeseries to see drift and/or variability. Details on how the data points that are the basis for these fits were generated are missing. (accepted variability, length of calibration pulses,...)*

The figure is intended to show a lack of variability in the mole fraction calibration lines (linear and squared dependency) over a longer measurement period with instrument drift (offset). While drift over time could also be plotted in the way the reviewer suggests, this is not the message we want to convey. We have added further details on the calibration procedure to the figure caption.

L. 375; Even though the absolute isotopic composition varied strongly over hourly timescales, mole fraction dependencies remained relatively constant during the 9-day period. The calibrations shown were made using a system that traps a calibration sample in the instruments sample cell. After flushing the sample cell with calibration gas 4 times, the final gas sample was trapped and measured for 5 minutes. Data from the last 3 minutes was averaged and used to fit calibration lines. The 3-minute plateaus were all stable below σ-$CO_2$ of 1 µmol mol−1. This process was repeated at 4 $CO_2$ concentration levels to derive one of the plotted calibration lines.

> *(Ref #2)* Figure 9: The panel letters B to E referred to in the caption are missing in the figure, panel A is not referred to in the caption.

This has been incorporated. See Figure F (Figure 9 in the manuscript).

Caption; 30-minute $H_2O$ exchange fluxes over the 6-day measurement period in Spain near Mollerussa. Panel A shows the evolution of LvE measured by the LS-$H_2O$ and OPGA. Panel B and C display the δ-flux derived using Eq. (2), including the 1σ error from the spectral scaling method (Sect. 4.4) "

**Technical corrections:**

> *(Ref #1)* L. 10: rewrite ".. analyse the measured values of central ∂-flux variable". What does central mean here?
>
> *(Ref #2)* L10: Something missing in "of central δ-flux variable"

We reformulated the sentence as follows;
L 10; We present a systematic procedure to scrutinise and analyse measurements of the δ-flux variable, which plays a central role in flux partitioning.

> *(Ref #1)* L. 44: Delete the Griffis 2013 citation here. Eq. 1.: Usually, the R is defined with the isotopologue ratio instead of the isotope ratio (Mook 2000) which is also what the laser spectrometer instruments measure.

We have deleted the reference to Griffis 2013. Also, we adjusted Eq. 1 to refer to molecules instead of elements. Accordingly, equation 4 was adjusted to exclude the 1/n term.

> *(Ref #1)* L. 66: Evaporation should not be used as exemplary process to explain "equilibrium fractionation" since it is NOT an equilibrium process alone and kinetic fractionation contributes to total fractionation. Rewrite that paragraph.

In this section we highlight some of the major processes causing isotopic fractionation in land-atmosphere exchange. We agree that transpiration, and even most types of natural evaporation are a combination of both equilibrium kinetic effects. However, we intended to only describe the effect of the phase transition, and the text is updated accordingly.

L. 66; One example is the phase transition of liquid water to vapor, where light isotopologues evaporate preferentially compared to heavy isotopologues.

> *(Ref #1)* L. 350: what is meant by "evaporative fractionation related to the transpiration" -reword

We reformulated the sentence as follows:
L 350: "The persistent preferential evaporation of light isotopologues from the small water volume (Sect. 2.1)."

| *(Ref #1)* L. 19: ET and NEE need to be introduced first | Incorporated |
|---|---|

| | |
|---|---|
| *(Ref #1) L. 44 and other places: The "Wahl et al." reference is missing details. Update reference to the published version* https://doi.org/10.1029/2020JD034400 | Updated reference |
| *(Ref #1) L. 119: 26th* | Implemented |
| *(Ref #1) L. 127: mention sampling frequency of Irgason* | Implemented |
| *(Ref #1) L. 131: correct "inst"* | Implemented |
| *(Ref #1) L. 135: correct grammar* | Implemented |
| *(Ref #1) L. 138: delete "m"* | Implemented |
| *(Ref #1) L. 314: insert what delta values you are giving here? D18O? of $CO_2$ or $H_2O$?* | Implemented (although specified in 313) |
| *(Ref #1) L. 316: Specify everywhere in the script whether you talk about d18O of $CO_2$ or $H_2O$.* | Partially implemented, often clear from context |
| *Fig. 9 caption: The panel labels are mixed up.* | Implemented |
| *(Ref #1) L. 381: Exchange "A possible cause" with "The cause"* | implemented |
| *Figure A3: Which time period is presented?, use "re-scaled $CO_2$" for consistency inlegend* | implemented |
| *(Ref #2) L43: group\*s\** | implemented |
| *(Ref #2) L226: exten\*t\** | implemented |
| *(Ref #2) L341 deform\*s* | implemented |
| *(Ref #2) L354: effect\** | implemented |
| *(Ref #2) L391:-392; cause\*, loss \*of\* covariance* | implemented |
| *(Ref #2) L409: While \*the\*? Principle* | implemented |
| *(Ref #2) L436: analys\*i\*s* | implemented |
| *(Ref #2) L455: presented \*a\*? methodological approach* | implemented |

**Changes from authors:**

- Subtitle changes:
    - 4.3.1 Mole fraction correction → Net flux correction
    - 4.3.2. delta-value correction → δ-flux correction

- More accurately specified atmospheric isotopic compositions in table 1.

- Included error estimations in table 1.

- Minor grammatical corrections.

**References**

Clog, M., Stolper, D., & Eiler, J. M. (2015). Kinetics of CO2(g)-H2O(1) isotopic exchange, including mass 47 isotopologues. *Chemical Geology*, *395*, 1–10. https://doi.org/10.1016/j.chemgeo.2014.11.023

Fan, S.-M., Wofsy, S. C., Bakwin, P. S., Jacob, D. J., & Fitzjarrald, D. R. (1990). Atmosphere-biosphere exchange of CO2 and O3 in the central Amazon forest. *Journal of Geophysical Research*, *95*(D10). https://doi.org/10.1029/jd095id10p16851

Finkelstein, P. L., & Sims, P. F. (2001). Sampling error in eddy correlation flux measurements. *Journal of Geophysical Research Atmospheres*, *106*(D4), 3503–3509. https://doi.org/10.1029/2000JD900731

Foken, T. (2008). The energy balance closure problem: An overview. *Ecological Applications*, *18*(6), 1351–1367. https://doi.org/10.1890/06-0922.1

Fratini, G., & Mauder, M. (2014). Towards a consistent eddy-covariance processing: An intercomparison of EddyPro and TK3. *Atmospheric Measurement Techniques*, *7*(7), 2273–2281. https://doi.org/10.5194/amt-7-2273-2014

Gat, J. R., Mook, W. G., & Meijer, H. A. J. (2001). Environmental isotopes in the hydrological cycle, volume II, Atmospheric water. In *Technical Documents in Hydrology I No* (Vol. 39). https://www.hydrology.nl/images/docs/ihp/Mook_II.pdf

Good, S. P., Soderberg, K., Wang, L., & Caylor, K. K. (2012). Uncertainties in the assessment of the isotopic composition of surface fluxes: A direct comparison of techniques using laser-based water vapor isotope analyzers. *Journal of Geophysical Research Atmospheres*, *117*(15). https://doi.org/10.1029/2011JD017168

Griffis, T. J. (2013). Tracing the flow of carbon dioxide and water vapor between the biosphere and atmosphere: A review of optical isotope techniques and their application. *Agricultural and Forest Meteorology*, *174–175*, 85–109. https://doi.org/10.1016/j.agrformet.2013.02.009

Griffis, T. J., Zhang, J., Baker, J. M., Kljun, N., & Billmark, K. (2007). Determining carbon isotope signatures from micrometeorological measurements: Implications for studying biosphere-atmosphere exchange processes. *Boundary-Layer Meteorology*, *123*(2), 295–316. https://doi.org/10.1007/s10546-006-9143-8

Keeling, C. D. (1958). The concentration and isotopic abundances of atmospheric carbon dioxide in rural areas. In *Geochimica et Cosmochimica Acta* (Vol. 13).

Kljun, N., Calanca, P., Rotach, M. W., & Schmid, H. P. (2004). A simple parameterisation for flux footprint predictions. *Boundary-Layer Meteorology*, *112*, 503–523.

Kohn, M. J. (2010). *Carbon isotope compositions of terrestrial C3 plants as indicators of (paleo)ecology and (paleo)climate*. *107*(46). https://doi.org/10.1073/pnas.1004933107/-/DCSupplemental

Moore, C. J. (1986). Frequency response corrections for eddy correlation systems. *Boundary-Layer Meteorology* , *37*, 17–35.

Oikawa, P. Y., Sturtevant, C., Knox, S. H., Verfaillie, J., Huang, Y. W., & Baldocchi, D. D. (2017). Revisiting the partitioning of net ecosystem exchange of CO2 into photosynthesis and respiration with simultaneous flux measurements of 13CO2 and CO2, soil respiration and a biophysical model, CANVEG. *Agricultural and Forest Meteorology*, *234–235*, 149–163. https://doi.org/10.1016/j.agrformet.2016.12.016

Sturm, P., Eugster, W., & Knohl, A. (2012). Eddy covariance measurements of $CO_2$ isotopologues with a quantum cascade laser absorption spectrometer. *Agricultural and Forest Meteorology*, *152*(1), 73–82. https://doi.org/10.1016/j.agrformet.2011.09.007

Vickers, D., & Mahrt, L. (1997). Quality Control and Flux Sampling Problems for Tower and Aircraft Data. *Journal of Atmospheric and Oceanic Technology*, *14*, 512–526.

Wahl, S., Steen-Larsen, H. C., Reuder, J., & Hörhold, M. (2021). Quantifying the Stable Water Isotopologue Exchange Between the Snow Surface and Lower Atmosphere by Direct Flux Measurements. *Journal of Geophysical Research: Atmospheres*, *126*(13). https://doi.org/10.1029/2020JD034400

Wehr, R., Munger, J. W., Nelson, D. D., McManus, J. B., Zahniser, M. S., Wofsy, S. C., & Saleska, S. R. (2013). Long-term eddy covariance measurements of the isotopic composition of the ecosystem-atmosphere exchange of $CO_2$ in a temperate forest. *Agricultural and Forest Meteorology*, *181*, 69–84. https://doi.org/10.1016/j.agrformet.2013.07.002

Wei, Z., Yoshimura, K., Okazaki, A., Kim, W., Liu, Z., & Yokoi, M. (2015). Partitioning of evapotranspiration using high-frequency water vapor isotopic measurement over a rice paddy field. *Water Resources Research*, *51*(5), 3716–3729. https://doi.org/10.1002/2014WR016737

Weng, Y., Touzeau, A., & Sodemann, H. (2020). Correcting the impact of the isotope composition on the mixing ratio dependency of water vapour isotope measurements with cavity ring-down spectrometers. *Atmospheric Measurement Techniques*, *13*(6), 3167–3190. https://doi.org/10.5194/amt-13-3167-2020

Wilczak, J. M., Oncley, S. P., & Stage, S. A. (2001). Sonic anemometer tilt correction algorithms. *Boundary-Layer Meteorology*, *99*, 127–150.

Yakir, D., Berry, J. A., Giles, L., & Osmond, C. B. (1994). Isotopic heterogeneity of water in transpiring leaves: identification of the component that controls the d18O of atmospheric O2 and CO2. *Plant, Cell and Environment*, *17*, 73–80.

Yakir, D., & Wang, X.-F. (1996). Fluxes of CO2 and water between terrestrial vegetation and the atmosphere estimated from isotope measurements. *Nature*, *380*, 515–517.

---

## Referee Report (RR1)

Review for „Data treatment and corrections for estimating H2O and CO2 isotope fluxes from high-frequency observations" by Robbert P.J. Moonen et al.

**General comments:**

The authors have successfully included the feedback from the first review round and the manuscript has improved in clarity. I also thank the authors for the extensive replies to my comments, which resolved most of the questions I had. Some general questions remain which I will outline below.
In general, I think the manuscript is a valuable contribution to the field of measuring stable isotope fluxes and isofluxes and will help in designing future field campaigns and post-processing procedures.

**Specific comments:**

**Mole-fraction calibration for H2O:**
Thank you for outlining the calibration protocol in Table A1 which resolved many questions I had. I admit though that I still don't fully understand what was done and why a leak would explain the cross dependency between mole fraction dependence and standard water isotopic composition considering that the leak/ambient isotopic composition before and after the campaign when the calibrations were done was probably not identical in itself. I also still think that a visual representation of the *calibration coefficients* in Figure 3 is not very helpful and should at least be accompanied by the final values that you used in a table format (possibly together with the CO2 coefficients in Table 2) and a graph showing the fitted quadratic curves to the datapoints from the calibration (in the Appendix). Could you also outline what mole fraction level you used as reference for the correction function? A priori I would assume that a fit to "3 humidity levels with 6 and 4 datapoints each" (did I understand correctly?) in the range 7000-24000ppmv is not very robust and might be the reason for differences in *calibration coefficients* before and after the campaign rather than a leak. Similarly, you are attributing the differences in span calibration (0.4‰ for 18O is rather high) to drift alone but this could also just be due to noise in your calibration pulses and should be phrased more cautiously in my opinion. As you say though, the accuracy of your measurements is arguably less important than the precision. That being said, I wonder if the mole-fraction calibration is indeed as crucial as you say since the fluctuations within the 30min averaging window will be small, even if the mole fraction sensitivity of your instrument seems very high if I read off the coefficient values correctly from Figure 3.
And lastly, I don't understand the uncertainties of the standard waters you give in Table 1 and A1 in terms of significant figures/decimal places and origin of these values. Does the standard error come from a separate analysis of these standard waters?

**CF factor analysis**
I still think it would be interesting to see or discuss the CF factor **values**. The statement of Line 415 that the authors were willing to include (comparison between CF for dD and d18O) is very limited and does not discuss the "difficulty" of correction values that are VERY high for either isotope species. I think this aspect of correcting a measured signal by a very high factor (at least for H2O isotopes) should be discussed in the section 6 as a general problem of measuring isotope or delta fluxes. It could be discussed also in the context of the

intriguing thought of precision vs. sampling frequency in new instrument development (L. 489-492).

Additionally, I think it is worth to include a discussion of the CF factors for CO2 that are ≈ 1 and thus in line with previous studies which I think is supporting your new correction technique. I therefore recommend to include a short CO2 CF factor discussion in section 6.

**Data availability statement:**
I do support the general community effort of making data publicly available with a doi upon publication in line with the FAIR principles:
https://www.atmospheric-measurement-techniques.net/policies/data_policy.html

https://www.atmospheric-measurement-techniques.net/policies/data_policy.html#data_availability

**Technical corrections:**

L. 16-18: It sounds as if you have used your results to validate models in this manuscript. Maybe rephrase.
L. 21: missing space after CO2
L. 56 & 60: "δ-flux" term used before its introduced/defined/explained in section 2.2
L. 76: change "isotope fractionation" to "isotopic fractionation"
L. 129: correct "we included figure is included"
L. 154: correct "and an scroll additional scroll pump"
L. 164: repetition of "which"
L. 190: something missing: "the calibration of the isotopic against a reference standard"
Section 4.2, L. 229: I think you should include a short explanation here why you chose to time shift 10minutes instead of 30 minutes.
L. 287: repetition of "that"
L. 301, 452, 456, 457: I think the term "isotopic *exchange* in the inlet line" is misleading. I suggest to use "isotopic retention" or, as you use in Line. 562, "inlet line attenuation" instead.
L. 321, L. 329: How much of the data (30min windows) did you eventually filter/exclude from the analysis? This is interesting for the reader to know.
Figure 7 caption, L. 342, L. 346, Figure A3: replace "R2" with "r"
L. 343: I don't see why a linear interpolation is the "likely pattern" during the night given that the variability in periods of similar length during day time is much more variable than linear. Since you exclude night time periods from your analysis (L. 353) I recommend to not show linear interpolations in Figure 7 and remove the "likely pattern" statement.
L. 383: delete "="
L. 493: Replace "still" with "however"
L. 495: include here a statement on the choice of LPF frequency
L. 547: delete "in"
Section 7: I would also include best practice recommendations not only for the technical set-up but also for the post-processing procedure since this is what you are newly proposing in this manuscript (LPF, averaging time, clock synchronization…)
L. 567: Consider moving the sentence "The attenuation of the isotopic signal…" to Line 564 after "reported previously."

---

## Author Response (AR2)

**Rebuttal 2**

*Data treatment and corrections for estimating $H_2O$ and $CO_2$ isotope fluxes from high frequency observations.*

Robbert P.J. Moonen, Getachew A. Adnew, Oscar K. Hartogensis, Jordi Vilà -Guerau de Arellano, David J. Bonell Fontas, and Thomas Röckmann

We again thank the referee and the editor for their valuable comments and encouraging the resubmission. Our replies are given in blue, whereas changes to the final manuscript are given in red.

**General comments**

**Referee #1**

The authors have successfully included the feedback from the first review round and the manuscript has improved in clarity. I also thank the authors for the extensive replies to my comments, which resolved most of the questions I had. Some general questions remain which I will outline below.
In general, I think the manuscript is a valuable contribution to the field of measuring stable isotope fluxes and isofluxes and will help in designing future field campaigns and postprocessing procedures.

**Specific comments**

> **Mole-fraction calibration for H2O**
> *Thank you for outlining the calibration protocol in Table A1 which resolved many questions I had. I admit though that I still don't fully understand what was done and why a leak would explain the cross dependency between mole fraction dependence and standard water isotopic composition considering that the leak/ambient isotopic composition before and after the campaign when the calibrations were done was probably not identical in itself. I also still think that a visual representation of the calibration coefficients in Figure 3 is not very helpful and should at least be accompanied by the final values that you used in a table format (possibly together with the CO2 coefficients in Table 2) and a graph showing the fitted quadratic curves to the datapoints from the calibration (in the Appendix). Could you also outline what mole fraction level you used as reference for the correction function? A priori I would assume that a fit to "3 humidity levels with 6 and 4 datapoints each" (did I understand correctly?) in the range 7000-24000ppmv is not very robust and might be the reason for differences in calibration coefficients before and after the campaign rather than a leak. Similarly, you are attributing the differences in span calibration (0.4‰ for 18O is rather high) to drift alone but this could also just be due to noise in your calibration pulses and should be phrased more cautiously in my opinion. As you say though, the accuracy of your measurements is arguably less important than the precision. That being said, I wonder if the mole-fraction calibration is indeed as crucial as you say since the fluctuations within the 30min averaging window will be small, even if the mole fraction sensitivity of your instrument seems very high if I read off the coefficient values correctly from Figure 3. And lastly, I don't understand the uncertainties of the standard waters you give in Table 1 and A1 in terms of significant figures/decimal places and origin of these values. Does the standard error come from a separate analysis of these standard waters?*

We thought it important to specify our mole fraction calibration approach and correction procedure as we observed large differences in the 30-minute isofluxes derived with or without applying mole fraction dependence corrections. This difference arises from artificial fluctuations in the delta-value signal which correlate with w' as the underlying mole fraction fluctuations correlate with w'. As the reviewer indicates, our instrument does have a large mole fraction dependence, resulting in a large artificial flux when not using dependency corrections. We agree that a rough dependency estimate would already result in reasonable isofluxes, and that the correction for a possible leak is only a small improvement.
As the reviewer observes, atmospheric isotopic compositions were likely different before and after the campaign. However, natural atmospheric variations are much smaller compared to the differences

between the standards we used (Table 1). The approximate atmospheric background composition is also reflected in the rough 3/1 ratio between the two standards to reflect the atmospheric isotopic composition.

We created the figure below, which we expect will clear up the mole fraction calibrations and leak correction we applied. It is an example where the calibration coefficients are fitted to the references measured after the campaign. Note that the measurements are offset to be approximately zero on the y-axis to be able to display both standards in one figure. The solid green and blue lines indicate the fitted dependencies of each standard. For dD, both lines depart from linear behaviour at low mole fractions. The GL ice standard, which is naturally depleted, tends towards higher dD values at low mole fractions, while the NL tap water standards, which his naturally enriched (see Table 1 in the manuscript), is moved towards lower dD values at low mole fractions. To show that this effect is likely caused by a leak of ambient air, we simulated a 0.3% leak in the inlet during the calibrations. We observe that at low mole fractions, the impact of the simulated leak on the delta values is larger due to the low water content in the calibration gas stream. More importantly, the leak corrected data points and calibration lines are consistent with the Weighted Avg mole fraction dependence we derived, which indirectly corroborates that our correction approach makes sense. Note that the effect of a leak in d18O-H2O is much less pronounced due to the larger mole fraction dependence in d18O, combined with a smaller absolute difference between the delta values of the calibration standards and ambient air.

We double checked if the same leak could have affected the span calibration measurements. In that case, a scale contraction should be observed. In addition, this scale contraction should be larger during the calibrations after the campaign, as a larger leak was present there (see Figure 3). We observe neither of these effects in our calibration data. We recall that a leak of ambient air during a span calibration affects the entire calibration scale, except at the isotopic composition of said ambient air, which is exactly what we measured during the campaign.

[Figure]

The error bars of the measurements plotted in the figure indicate the standard error of the mean during each 20min measurement plateau, after averaging the high frequency data in 1-minute bins. While our calibration approach is simple, the error bars indicate the method is robust. In line with that, repeat measurements were mostly indistinguishable from the initial measurement.

The standard errors of the standards in Table 1 were also derived as the standard error of the mean during the crosscalibration procedure. However, we did not bin the data to 1-minute averages in the previous version of the manuscript, causing the sqrt(n) term to explode due to the frequency of the measurements and thereby generating unrealistically small uncertainties. In addition, we realize that the deuterium error given in table 1 was 10x too small due to the accidental addition of a decimal 0, causing it to be extra unrealistic. We now applied 1-minute binning which we assume to result in independent samples and adjusted the errors accordingly.

Finally, Table 2 (below) was updated to include the weighted average mole fraction calibration coefficients.

**Table 2.** The dependence of isotopic composition measurements ($\delta$-values) on the mole fraction of the respective molecule, $H_2O$ or $CO_2$, expressed in $\mu mol\,mol^{-1}$. The $CO_2$ dependencies were derived after the campaign while the $H_2O$ dependencies represent the weighted average coefficients during the campaign (see Fig. 3).

| Mole fraction dependence | $\delta^{18}O$ - $CO_2$ | $\delta^{13}C$ - $CO_2$ | $\delta D$ - $H_2O$ | $\delta^{18}O$ - $H_2O$ |
|---|---|---|---|---|
| Linear | -1.27e$^{-2}$ | -3.64e$^{-2}$ | -7.71e$^{-5}$ | -8.54e$^{-5}$ |
| Quadratic | 1.80e$^{-6}$ | 1.16e$^{-5}$ | -1.78e$^{-9}$ | -2.60e$^{-9}$ |

Besides adding the figure to the appendix, updating table 2, and including in-text references, we adjusted the following:

L. 202; Differences in the span calibrations performed before and after the campaign were +/- 0.4‰ for δ18O and +/- 0.3‰ for δD at atmospheric isotopic compositions. We suspect that instrument drift is the cause given the small uncertainty in the (re)calibrations (0.02‰ for both d13C and d18O). When ignoring drift inducing events like instrument rebooting and transportation, the interpolated drift during the measurement period is still below 0.1‰ for both species.

Adjusted errors in table 1 and appendix table A1:

**Table 1.** The isotopic compositions of the $H_2O$ calibration standards including the standard error of the 1-min binned data during cross-calibrations of the references, and the average atmospheric isotopic composition during the measurement period including its approximate range

| | $\delta^{18}O$ | $\delta D$ |
|---|---|---|
| NL tap water | -6.98 ± 0.02‰ | -47.12 ± 0.04‰ |
| GL icecore | -30.80 ± 0.02‰ | -240.86 ± 0.04‰ |
| LIAISE atm | -13.4 ± 1‰ | -94 ± 8‰ |

L. 213; We suggest that the similarity in the Weighted Avg (yellow) calibration coefficients "Before" and "After" the campaign in Fig. 3 is no coincidence, but a feature of an ambient air leak of variable magnitude. In Appendix A2, we give an example where the calibration coefficients in Figure 3 are fitted to the measurements. We also show simulations where we assume a small (counter) leak of ambient air (of 0.3%), which is able to explain the observed mole fraction dependencies. An instrument related cross dependency of the isotopic composition on the mole fraction dependence, as described by Weng et al. (2020), is not expected to average out like this. Ultimately, the dependencies were eliminated using 15000 umol mol-1 as the reference H2O mole fraction.
* * *
*CF factor analysis*

*I still think it would be interesting to see or discuss the CF factor values. The statement of Line 415 that the authors were willing to include (comparison between CF for dD and d18O) is very limited and does not discuss the "difficulty" of correction values that are VERY high for either isotope species. I think this aspect of correcting a measured signal by a very high factor (at least for H2O isotopes) should be discussed in the section 6 as a general problem of measuring isotope or delta fluxes. It could be discussed also in the context of the intriguing thought of precision vs. sampling frequency in new instrument development (L. 489-492).*

*Additionally, I think it is worth to include a discussion of the CF factors for CO2 that are ≈ 1 and thus in line with previous studies which I think is supporting your new correction technique. I therefore recommend to include a short CO2 CF factor discussion in section 6.*
* * *
Thanks for these useful suggestions which we implemented in the text. The second point was already partially mentioned from L. 479 to L. 483, but that we extended this argument.

We made the following changes to the text;

L. 489; However, isotopic ecosystem flux partitioning is impacted by our findings.

Be aware that in contrast to most correction approaches, where the order of magnitude of the correction is 10%, the spectral scaling approach leads to correction factors of the order of 100%. This means that most of the signal in the d-flux does not originate from the actual measurements of the delta-flux, but from the correction method, which is not desirable. Consequently, the errors of the corrected delta fluxes (see Sec. 4.4) can best be based on the uncertainty in fit of the correction, and not on the uncertainty of the measured delta-flux. As long as errors are properly quantified by propagating this fit error to the flux, we do believe that using the spectral correction approach is valuable for deriving delta-fluxes of the correct magnitude.

The implications of the spectral scaling principle are broader than using it to find adequate corrections.

L. 479; To prove this hypothesis we can investigate the cospectra of $\delta$-fluxes and mole fraction fluxes measured with a setup in which all turbulent scales are well represented. In Fig. 6 we show that the cospectral density for the $\delta_{13}C$ and $CO_2$ observations is generally very similar, supporting the validity of our hypothesis. While comparing the corrected and non-corrected d-fluxes of both $\delta_{13}C$ and $\delta_{18}O$ in Fig. 8 we see the same pattern. The corrected d-fluxes are mostly similar to the non-corrected d-fluxes, even though all high frequency fluctuations were eliminated. Note that the noise and large uncertainties that are present in the corrected delta fluxes can be partially attributed to the instrument drift in the LS-CO2 on long timescales and a relatively low signal to noise ratio in the d18O-fluxes. <s>While the principle seems to hold, the $\delta_{13}C$ signal in Fig. 6 is impacted by instrument drift on long timescales and a relatively low signal to noise ratio</s>. More precise experimental measurements of the net ecosystem exchange and net isotopic exchange of trace gasses affected by various fractionation processes should increase confidence in this hypothesis.

> *Data availability statement:*
> *I do support the general community effort of making data publicly available with a doi upon publication in line with the FAIR principles:*
>
> *https://www.atmospheric-measurement-techniques.net/policies/data_policy.html*
> *https://www.atmospheric-measurementtechniques.net/policies/data_policy.html#data_availability*

We now made the 30-minute flux data available online using *figshare*, using the following DOI: 10.6084/m9.figshare.23828514. We added the DOI to the data availability statement, including our contact details if other researchers want to use our high frequency data.

**Technical corrections**

| | |
|---|---|
| *L. 16-18: It sounds as if you have used your results to validate models in this manuscript. Maybe rephrase.*

 Only after such corrections and verifications are made, ecosystem scale fluxes can be partitioned using isotopic fluxes as constraints, which in turn allows for conceptual land-atmosphere exchange models to be validated. | Incorporated |
| *L. 21: missing space after CO2* | Incorporated |
| *L. 56 & 60: "$\delta$-flux" term used before its introduced/defined/explained in section 2.2* | incorporated |
| *L. 76: change "isotope fractionation" to "isotopic fractionation"* | Incorporated |
| *L. 129: correct "we included figure is included"* | incorporated |
| *L. 154: correct "and an scroll additional scroll pump"* | incorporated |
| *L. 164: repetition of "which"* | incorporated |
| *L. 190: something missing: "the calibration of the isotopic against a reference standard"* | incorporated |

| | |
|---|---|
| *Section 4.2, L. 229: I think you should include a short explanation here why you chose to time shift 10minutes instead of 30 minutes.*

After the two time series are coarsely aligned using known clock offsets, the longer time series was divided into short data intervals.
We choose to use 10-minute intervals instead of the 30-minute intervals used for flux calculations to increase sensitivity to data series which drift fast and irregularly with respect to each other. One of the data series is subsequently cropped by a minute on each side which makes it possible to shift it in time with respect to the other data series within this (two) minute window. | incorporated |
| *L. 287: repetition of "that"* | incorporated |
| *L. 301, 452, 456, 457: I think the term "isotopic exchange in the inlet line" is misleading. I suggest to use "isotopic retention" or, as you use in Line. 562, "inlet line attenuation" instead.* | incorporated |
| *L. 321, L. 329: How much of the data (30min windows) did you eventually filter/exclude from the analysis? This is interesting for the reader to know.*

After these filtering steps, 81% of the 30-min intervals of co2 isotope fluxes, and 80% of the 30-min intervals of h2o isotope fluxes were still complete. | incorporated |
| *Figure 7 caption, L. 342, L. 346, Figure A3: replace "R2" with "r"* | incorporated |
| *L. 343: I don't see why a linear interpolation is the "likely pattern" during the night given that the variability in periods of similar length during day time is much more variable than linear. Since you exclude night time periods from your analysis (L. 353) I recommend to not show linear interpolations in Figure 7 and remove the "likely pattern" statement.*

Updated Fig 7 to remove night-time interpolation shown in dashed green, after which the likely pattern remark about the dashed green line is correct. | incorporated |
| *L. 383: delete "="* | incorporated |
| *L. 493: Replace "still" with "however"* | incorporated |
| *L. 495: include here a statement on the choice of LPF frequency* | incorporated |
| *L. 547: delete "in"* | incorporated |
| *Section 7: I would also include best practice recommendations not only for the technical set- up but also for the post-processing procedure since this is what you are newly proposing in this manuscript (LPF, averaging time, clock synchronization...)*

We extended the best practice recommendations to include post-processing procedures:

Additionally, we provide the following best practice recommendations for post-processing the isotope flux measurements.

■ Apply outlier filtering to the detrended scalar data before flux calculations, especially for the measured isotopic compositions. This prevents errors in the flux estimates due to fluctuations that were not induced by turbulence. We used an Inter-Quartile Range (IQR) filter for outlier filtering, as described in Sec. 4.4.
■ Time alignment of the high frequency isotope analyser(s) and the high frequency anemometer measurements is a prerequisite for calculating reliable iso-fluxes. The alignment strategy we used is described in Sec. 4.2 and uses the mole fraction signals of the target molecule, measured by both the OPGA and high frequency isotope analyser, as a reference. It corrects for both the effects of instrument clock drift and inlet delays. | incorporated |

| | |
|---|---|
| ■ Test if there is a lag between the mole fraction and δ-value signals of the isotope analyser, especially when measuring $H_2O$ isotope fluxes. This can also be done using the alignment strategy described in Sec. 4.2. If the lag is not zero, isotopic inlet attenuation likely occurred, and exchange spectra are probably affected.
■ Compare the cospectra of the net exchange flux with the cospectra of the isotope exchange flux to identify if high frequency isotope flux signal was missed, as is shown in Fig. 5. If flux signal was missed, the spectral scaling method can be used to correct for the missing high frequency signal.
■ Quantify the uncertainty inherent to the correction method in case a major contribution to the δ-fluxes comes from the spectral scaling correction. In Sec. 4.4 we indicate how this uncertainty can be derived and propagates to the δ fluxes and isotopic source compositions. | |
| *L. 567: Consider moving the sentence "The attenuation of the isotopic signal..." to Line 564 after "reported previously."* | incorporated |

**Changes from authors**

In figure 3 the units of linear and quadratic terms were swapped in the old manuscript. This has now been corrected.